# CALM-PDE: Continuous and Adaptive Convolutions for Latent Space Modeling of Time-dependent PDEs

**Jan Hagnberger**[*]    **Daniel Musekamp**[1]    **Mathias Niepert**[1,2]

Machine Learning and Simulation Lab, Institute for Artificial Intelligence, University of Stuttgart
[1]International Max Planck Research School for Intelligent Systems (IMPRS-IS)
[2]Stuttgart Center for Simulation Science (SimTech)

## Abstract

Solving time-dependent Partial Differential Equations (PDEs) using a densely discretized spatial domain is a fundamental problem in various scientific and engineering disciplines, including modeling climate phenomena and fluid dynamics. However, performing these computations directly in the physical space often incurs significant computational costs. To address this issue, several neural surrogate models have been developed that operate in a compressed latent space to solve the PDE. While these approaches reduce computational complexity, they often use Transformer-based attention mechanisms to handle irregularly sampled domains, resulting in increased memory consumption. In contrast, convolutional neural networks allow memory-efficient encoding and decoding but are limited to regular discretizations. Motivated by these considerations, we propose CALM-PDE, a model class that efficiently solves arbitrarily discretized PDEs in a compressed latent space. We introduce a novel continuous convolution-based encoder-decoder architecture that uses an epsilon-neighborhood-constrained kernel and learns to apply the convolution operator to adaptive and optimized query points. We demonstrate the effectiveness of CALM-PDE on a diverse set of PDEs with both regularly and irregularly sampled spatial domains. CALM-PDE is competitive with or outperforms existing baseline methods while offering significant improvements in memory and inference time efficiency compared to Transformer-based methods.

## 1 Introduction

Many scientific problems, such as climate modeling and fluid mechanics, rely on simulating physical systems, often involving solving spatio-temporal Partial Differential Equations (PDEs). In recent years, Machine Learning (ML) models have been successfully used to approximate the solution of PDEs (Lu et al., 2019; Li et al., 2020; Brandstetter et al., 2022; Chen & Wu, 2024), offering several advantages over classical numerical PDE solvers. For instance, ML models offer a data-driven approach that is applicable even if the underlying physics is (partially) unknown, can generate solutions more efficiently (Li et al., 2021b; Tompson et al., 2017), and are inherently differentiable by design, which is often not the case for numerical solvers (Takamoto et al., 2022).

Practical applications usually require a densely discretized spatial domain $\Omega$, leading to more than 1M spatial points per timestep. For example, ML-based weather forecasting models typically operate on a spatial domain of $720 \times 1440$ points or pixels (Pathak et al., 2022). Learning the solution of

---

[*]Corresponding author: j.hagnberger@gmail.com
The source code of CALM-PDE is available on GitHub: **https://github.com/jhagnberger/calm-pde**

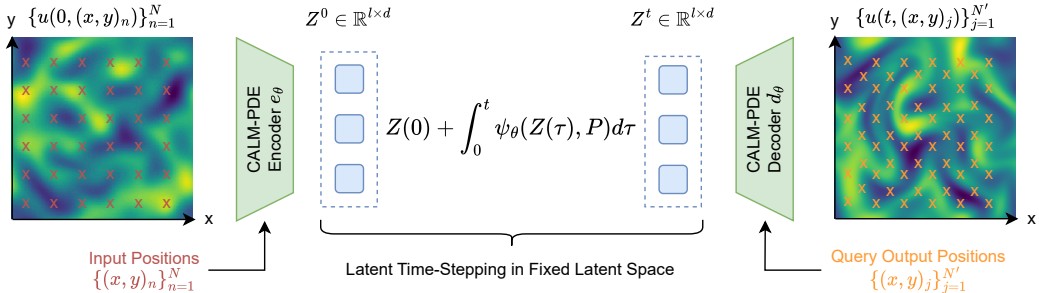

Figure 1: CALM-PDE encodes the arbitrarily discretized PDE solution into a fixed latent space $\mathbb{R}^{l \times d}$, computes the dynamics in the latent space, and decodes the solution for the given query points.

a time-dependent PDE directly in the physical domain $\Omega$ rather than in a compressed latent space can result in high memory and computational costs. Consequently, several architectures for reduced-order PDE-solving have been introduced (Wu et al., 2022; Alkin et al., 2024; Serrano et al., 2024; Wang & Wang, 2024). These models adopt the encode-process-decode paradigm (Sanchez-Gonzalez et al., 2020), wherein the physical domain is encoded into a compact latent space, its dynamics are evolved via a processor, and the output is decoded back to the original domain. Many real-world applications involve geometries with irregularly sampled spatial domains, necessitating discretization-agnostic models to effectively process such data. Unfortunately, existing approaches mainly utilize Convolutional Neural Networks (CNNs), which necessitate a regular spatial discretization (Wu et al., 2022), or memory-intensive attention-based mechanisms (Vaswani et al., 2017) to enable the encoding and decoding of arbitrarily discretized spatial domains (Alkin et al., 2024; Serrano et al., 2024; Wang & Wang, 2024).

Motivated by these considerations, we introduce CALM-PDE (**C**ontinuous and **A**daptive Convolutions for **L**atent Space **M**odeling of Time-dependent **PDE**s), an architecture featuring a novel encoder and decoder for reduced-order modeling of arbitrarily discretized PDEs. CALM-PDE compresses the spatial domain into a fixed latent representation with an encoder that builds on parametric continuous convolutional neural networks (Wang et al., 2018) and adaptively learns where to apply the convolution operator (learnable query points). This enables CALM-PDE to selectively sample more densely in important regions of the spatial domain, such as near complex solid boundaries, while allocating fewer points to smoother regions. Furthermore, we incorporate a locality inductive bias into the kernel function, consisting of an epsilon neighborhood and a distance-weighting, to enhance computational efficiency and to facilitate the learning of local patterns. An autoregressive model computes the temporal evolution completely in the latent space (latent time-stepping), which can be interpreted as Neural Ordinary Differential Equation (NODE; Chen et al. (2018)). The discretization-agnostic decoder, which builds on similar layers as the encoder, enables querying the output solution at arbitrary spatial locations. An overview of the CALM-PDE framework is shown in Figure 1.

We demonstrate the effectiveness of our approach through a broad set of experiments, solving Initial Value Problems (IVPs) in fluid dynamics with regularly and irregularly sampled spatial domains.

Our main contributions are summarized as follows:

- We propose a model featuring a novel encoder and decoder that solves arbitrarily discretized PDEs in a fixed latent space.

- A novel encoder-decoder approach based on continuous convolutions that learns where to apply the convolution operation (query points) to effectively sample the spatial domain.

- A kernel function with a locality inductive bias (epsilon neighborhood and distance weighting) to enhance efficiency and encourage learning local patterns, and a modulation that allows query points with similar positions to consider different spatial features.

## 2   Problem Definition

In the following section, we formally introduce the problem of solving PDEs. We refer to Appendix B for information about the dataset and training objective to train neural surrogates for PDE-solving.

**Partial Differential Equations.**   Similar to Brandstetter et al. (2022), we consider time-dependent PDEs over the time dimension $t \in [0, T]$ and multiple spatial dimensions $\boldsymbol{\omega} = (x, y, z, \ldots)^\top \in \Omega \subseteq \mathbb{R}^{N_d}$ where $N_d$ denotes the spatial dimension of the PDE. Thus, a PDE is defined as

$$
\begin{aligned}
\partial_t \boldsymbol{u} &= F(t, \boldsymbol{\omega}, \boldsymbol{u}, \partial_{\boldsymbol{\omega}} \boldsymbol{u}, \partial_{\boldsymbol{\omega}\boldsymbol{\omega}} \boldsymbol{u}, \ldots), & (t, \boldsymbol{\omega}) &\in [0, T] \times \Omega \\
\boldsymbol{u}(0, \boldsymbol{\omega}) &= \boldsymbol{u}^0(\boldsymbol{\omega}) = \boldsymbol{u}^0, & \boldsymbol{\omega} &\in \Omega \\
\mathcal{B}[\boldsymbol{u}](t, \boldsymbol{\omega}) &= 0, & (t, \boldsymbol{\omega}) &\in [0, T] \times \partial\Omega
\end{aligned}
\tag{1}
$$

where $\boldsymbol{u} : [0, T] \times \Omega \to \mathbb{R}^{N_c}$ represents the solution function of the PDE that satisfies the Initial Condition (IC) $\boldsymbol{u}(0, \boldsymbol{\omega})$ for $t = 0$ and the boundary conditions $\mathcal{B}[\boldsymbol{u}](t, \boldsymbol{\omega})$ if $\boldsymbol{\omega}$ is on the boundary $\partial\Omega$ of the domain $\Omega$. $N_c$ denotes the number of output channels or field variables of the PDE. Solving a PDE involves computing (an approximation of) the function $u$ that satisfies Equation (1).

## 3   Background and Preliminaries

We briefly introduce deep parametric continuous convolutional neural networks (Wang et al., 2018). For comparison, discrete convolution and discrete CNNs are explained in Appendix C.

**Deep Parametric Continuous Convolutional Neural Networks.**   Let $f, k : \mathbb{R} \to \mathbb{R}$ be two real-valued functions and $a \in \mathbb{R}$, then $(f * k)(a) := \int_{-\infty}^{\infty} f(\alpha) \cdot k(a - \alpha) d\alpha$ is the convolution of $f$ and $k$. Wang et al. (2018) propose to approximate the function $k$, which is the learnable kernel, with a Multi-Layer Perceptron (MLP) and the integral with Monte Carlo integration, which yields

$$
(f * k)(a) = \int_{-\infty}^{\infty} f(\alpha) \cdot k(a - \alpha) d\alpha \approx \frac{1}{N} \sum_{n=1}^{N} f(\alpha_n) \cdot k(a - \alpha_n)
\tag{2}
$$

where $N$ input points $\alpha_n$ are sampled from the domain. The kernel function $k$ is constructed using an MLP $k_{\boldsymbol{\theta}}(a - \alpha)$ which spans the entire domain and is parametrized by a finite vector $\boldsymbol{\theta}$ containing the weights and biases. Similar to discrete convolutional layers, a deep parametric continuous convolutional layer consists of multiple filter kernels for $N_i$ input channels, which leads to

$$
(f * k)_o(a_j) = \sum_{i=1}^{N_i} \sum_{n=1}^{N} f_i(\alpha_n) \cdot k_{i,o}(a_j - \alpha_n)
\tag{3}
$$

where $o$ denotes the $o^{th}$ output channel. The output is computed for all output or query points $a_j$ and input points $\alpha_n$ with the function value $f(\alpha_n)$. We denote the set of output or query points as $A = \{a_j\}_{j=1}^{N'}$ and the set of input points as $\mathcal{A} = \{\alpha_n\}_{n=1}^{N}$. The number of output points $A$ does not necessarily have to be the same as the input points $\mathcal{A}$, which allows the continuous convolutional layer to reduce or compress information (cf., downsampling) or to increase the number of points (cf. upsampling). Similar to discrete CNNs, the receptive field for each query point can be limited to $M$ points such that

$$
(f * k)_o(a_j) = \sum_{i=1}^{N_i} \sum_{m \in \texttt{RF}(a_j)} f_i(\alpha_m) \cdot k_{i,o}(a_j - \alpha_m)
\tag{4}
$$

where $\texttt{RF}(a_j)$ outputs the indices for the $M$ points that lie in the receptive field of a query point $a_j$. The receptive field can be constructed by only considering the K-nearest neighbors of the point $a_j$ or by considering the points in an epsilon neighborhood. Setting $M := N$ means that the receptive field is not limited. Appendix D shows a visualization of continuous convolution. To generalize the layer from 1D to d-D, only the input dimension of the kernel function has to be adapted.

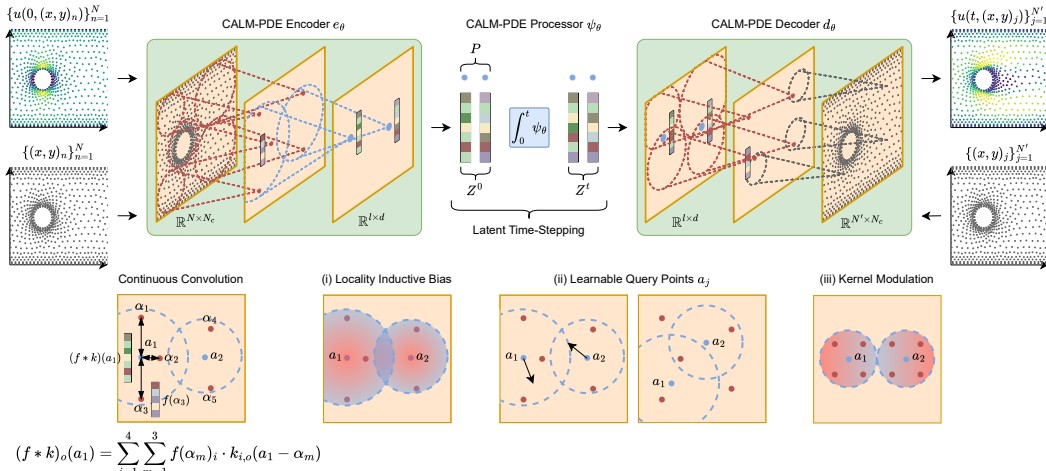

$$(f * k)_o(a_1) = \sum_{i=1}^{4} \sum_{m=1}^{3} f(\alpha_m)_i \cdot k_{i,o}(a_1 - \alpha_m)$$

Figure 2: Encode-process-decode architecture of CALM-PDE. The encoder reduces the spatial dimension and increases the channel dimension. It is based on multiple CALM layers, which perform continuous convolution on learnable query points constrained to an epsilon neighborhood.

# 4 Method

First, we introduce the CALM layer, which enhances continuous convolutional layers and acts as a building block for the architecture. Thereafter, we describe the encoder, processor for latent time-stepping, and decoder of the CALM-PDE model. Finally, we elaborate on the training strategy.

## 4.1 Learning Continuous Convolutions with Adaptive Query Points

We follow the formulation of a deep parametric continuous convolutional layer in Equation (4) introduced by Wang et al. (2018) and propose the following improvements for encoding and decoding PDE solutions, which yield the CALM layer used for the encoder and decoder.

**Parametrization of Kernel Function.** We use a 2-layer MLP that takes $a - \alpha$ as input to parameterize the continuous kernel function $k_{i,o}(a - \alpha)$. The translation $a - \alpha$ is encoded with Random Fourier Features (RFF; Li et al. (2021a)) to allow the kernel function to be less smooth (i.e., a small change in the input translation can lead to large output changes in the weight) which could be beneficial for high-frequency content where slightly different translation vectors should result in a large difference. Thus, the parametrized kernel function is given as $k_{i,o}(a - \alpha) = \text{MLP}\big(\text{RFF}(a - \alpha)\big)_{i,o}$ where $i$ is the $i^{th}$ input channel and $o$ the $o^{th}$ output channel.

**Epsilon Neighborhood and Distance Weighting.** We limit the receptive field by considering only the input points $\alpha_n$ within an epsilon neighborhood of the query point $a_j$, similar to Wang et al. (2018). The epsilon is dynamically computed by taking the $p^{th}$ percentile of the Euclidean distances from $a_j$ to all input points $\alpha_i$, similar to the mechanism proposed by Chen & Wu (2024). Thus, the hyperparameter $p$ controls the size of the epsilon neighborhood or receptive field in a relative fashion. Consequently, the size of the epsilon neighborhood varies depending on how densely the spatial points are sampled. In densely sampled regions, the epsilon neighborhood is smaller compared to sparsely sampled regions, with a larger epsilon neighborhood. We denote the receptive field based on this mechanism as $\text{RF}(a_j) = \{\alpha_n \in \mathcal{A} \mid \|a_j - \alpha_n\| \leq \epsilon(a_j)\}$ where $\epsilon(a_j)$ is the $p^{th}$ percentile of the Euclidean distances from $a_j$ to all input points $\alpha_n \in \mathcal{A}$. Furthermore, we introduce a distance weighting to the kernel function $k$. We use softmax with temperature $T$, similar to a Gaussian kernel, to represent the distances normalized to $[0, 1]$ within the epsilon neighborhood, which yields

$$k_{i,o}(a - \alpha) = \text{MLP}\big(\text{RFF}(a - \alpha)\big)_{i,o} \frac{\exp\left(-\frac{\|a-\alpha\|^2 - \min(a)}{T(\max(a) - \min(a))}\right)}{\sum\limits_{\alpha_j \in \text{RF}(a)} \exp\left(-\frac{\|a-\alpha_j\|^2 - \min(a)}{T(\max(a) - \min(a))}\right)} \tag{5}$$

for the kernel function with $\min(a) = \min_{\alpha_j \in \text{RF}(a)} \|a - \alpha_j\|^2$. This emphasizes input points closer to the query point within the epsilon neighborhood. The `softmax` also serves as a normalization factor for the Monte Carlo integration. The combination of the limited receptive field and distance weighting is a locality inductive bias, which is helpful because local patterns are often more important in PDE-solving (Chen & Wu, 2024). Part (i) of Figure 2 shows the receptive field and the distance weighting within the receptive field.

**Learnable Query Points.** In continuous convolution, the input and output points do not have to be the same. This means that downsampling and upsampling are inherently supported, in contrast to CNNs, where strided convolution and variants are needed for downsampling. This allows us to reduce the number of points for encoding and to increase the number of points for decoding. We propose to make the query points for both encoding and decoding learnable. This allows the model to sample more densely in regions with important characteristics (e.g., a region that contains turbulence or an obstacle). The query positions are initially uniformly sampled from the domain $\Omega$. We assume that $\Omega$ is normalized in a range $[0,1)$ which yields $A = \{a_j\}_{j=1}^K = \{(x_j, y_j)\}_{j=1}^K$ with $x_j, y_j \sim \mathcal{U}(0,1)$ for the initialization of the query points in 2D. Alternatively, the query points can be initialized by sampling $K$ query points from the underlying mesh. We call this initialization method of the query points as "mesh prior". During the training, the query points are moved with a variant of stochastic gradient descent such as Adam (Kingma & Ba, 2017) as $(x_j, y_j)^\top \leftarrow (x_j, y_j)^\top - \eta \frac{\partial \mathcal{L}_\theta}{\partial (x_j, y_j)^\top} \quad \forall j$ where $\eta$ denotes the learning rate and $\mathcal{L}_\theta$ the cost function. The kernel function provides feedback in which direction to move, and the `softmax` function in Equation (5) avoids hard cuts caused by the learnable query points and epsilon neighborhood by weighting the input points on the boundary of the epsilon neighborhood lower. The learnable query points are illustrated in part (ii) of Figure 2.

**Kernel Modulation.** We allow each query point to have a customized filter kernel (i.e., different kernel weights for the same or similar translation vectors). This way, there can be multiple query points at the same location, each paying attention to different features. This could be helpful for simulations that contain a geometry or obstacles (e.g., many query points close to the obstacle, but each query learns different characteristics). The kernel modulation is done by scaling and shifting the intermediate representations (Perez et al., 2018) of the MLP in Equation (5). Each query point has a location as well as scaling and shifting parameters. We denote the modulated kernel function as

$$k_{i,o}(a - \alpha) = \left( \boldsymbol{W_2} \cdot \sigma\Big( \big( \boldsymbol{W_1} \cdot \text{RFF}(a - \alpha) + \boldsymbol{b_1} \big) \odot \boldsymbol{\gamma}_a + \boldsymbol{\beta}_a \Big) + \boldsymbol{b_2} \right)_{i,o} \tag{6}$$

where we exclude the distance weighting for the sake of simplicity. $\boldsymbol{W}$ denotes a weight matrix with a suitable shape, $\sigma$ is a non-linearity, and $\boldsymbol{\gamma}_a, \boldsymbol{\beta}_a$ are the scale and shift for query point $a$, respectively. Part (iii) of Figure 2 demonstrates that the kernel modulation enables query points, despite having identical receptive fields and translation vectors of input points, to output different kernel weights.

**Periodic Boundaries.** PDE solutions often involve periodic boundary conditions. Thus, we adapt the translation vector $a - \alpha$ to account for periodic boundaries for the kernel function if the solution has a periodic boundary.

**CALM Layer.** A non-linearity follows the continuous convolution operation, which completes the CALM layer. Hence, the CALM layer with the previously defined kernel function $k$ is given as

$$l(f, \mathcal{A}, A) = \{(f * k)(a_j)\}_{j=1}^{N'}, \quad (f * k)_o(a_j) = \sigma\left( \sum_{i=1}^{N_i} \sum_{m \in \text{RF}(a_j)} f_i(\alpha_m) \cdot k_{i,o}(a_j - \alpha_m) + b_o \right) \tag{7}$$

where $f$ and $\mathcal{A}$ refers to the sampled input and $A$ to the query points. The query points are learnable, except for the final decoding layer, where the query points correspond to the queried spatial points. $\sigma$ corresponds to a suitable non-linearity such as ReLU or GELU, and $b_o$ is the bias for the $o^{th}$ channel.

## 4.2 Neural Architecture for Discretization-Agnostic Reduced-Order PDE-Solving

The architecture of the CALM-PDE model follows an encode-process-decode paradigm (Sanchez-Gonzalez et al., 2020). Figure 2 shows an overview of the CALM-PDE architecture.

**Encoder.** We stack multiple CALM layers together for the encoder. Each layer increases the number of channels (the number of output channels is larger than the number of input channels, i.e., $N_o \gg N_i$) and reduces the number of (spatial) points (the number of query points is much smaller than the number of input points, i.e., $|A| \ll |\mathcal{A}|$). This design is similar to CNN-based encoders, which reduce the spatial size of the feature maps and increase the number of feature maps to encourage hierarchical feature learning. The output of the encoder $e_\theta$ is a set of latent tokens $Z^t$ for time $t$ and their corresponding positions $P$. We denote the encoder as

$$e_\theta(\{u^t(\omega_n)\}_{n=1}^N, \{\omega_n\}_{n=1}^N) = \left( \underbrace{l_k\big(\ldots l_1(\{u^t(\omega_n)\}_{n=1}^N, \{\omega_n\}_{n=1}^N, A_1), A_{k-1}, A_k\big)}_{:=Z^t \in \mathbb{R}^{l \times d}}, \underbrace{A_k}_{:=P} \right) \quad (8)$$

where $k$ denotes the number of layers, $l$ is a CALM layer, $\{u^t(\omega_n)\}_{n=1}^N$ describes the solution function at time $t$ evaluated at the locations $\{\omega_n\}_{n=1}^N$. $Z^t$ and $P$ are the final representations of the input solution. Note that the positions $P$ are learned, fixed after the training, and independent of $u^t$.

**Processor for Latent Time-Stepping.** The model evolves the dynamics within the latent space through an autoregressive prediction. In particular, the processor $\psi_\theta$ predicts the difference from the latent tokens $Z^t$ at time $t$ to the future latent tokens $Z^{t+\Delta t} = \Delta t \cdot \psi_\theta(Z^t, P) + Z^t$. To obtain the solution for a timestep $t + n \cdot \Delta t$, the processor is applied $n$ times iteratively to completely evolve the dynamics in the latent space (latent time-stepping). We opt for a Transformer model (Vaswani et al., 2017) since attention allows the processor to capture global information from local tokens, and latent tokens can dynamically interact with each other. For instance, the processor can learn the dynamics of a moving vortex in a CFD simulation by adapting the latent tokens on the path of the moving vortex and pushing it from one position to another. We include the positional information of the latent tokens into self-attention using the Euclidean distance, similar to Chen & Wu (2024). We refer to Appendix E.2 for details. Using the processor to predict the residual and scaling it with the temporal resolution $\Delta t$ can be understood as solving a NODE parametrized by $\psi_\theta$, which is given as $Z(t + \Delta t) = Z(t) + \int_t^{t+\Delta t} \psi_\theta(Z(\tau), P)d\tau$ with an explicit Euler solver. Thus, the processor implicitly defines a vector field that could be integrated with solvers more advanced than the explicit Euler method, such as a Runge-Kutta solver.

**Decoder.** The decoder $d_\theta$ is also based on multiple CALM layers that reduce the number of channels and increase the number of spatial points. The decoder takes the latent tokens $Z^t$, positions $P$, and a set of query points $\{\omega_j\}_{j=1}^{N'}$ as input and outputs the solution in the physical space for the queried points. The decoder with $k$ layers is given as

$$d_\theta(Z^t, P, \{\omega_j\}_{j=1}^{N'}) = l_k\left( l_{k-1}\big(\ldots l_1(Z^t, P, A_1), A_{k-2}, A_{k-1}\big), A_{k-1}, \{\omega_j\}_{j=1}^{N'} \right)$$
$$= \{u^t(\omega_j)\}_{j=1}^{N'} \quad (9)$$

which maps from the latent tokens $Z^t$ and their position $P$ to the physical domain for the given set of queried spatial points $\{\omega_j\}_{j=1}^{N'}$.

### 4.3 Training Procedure

We opt for an end-to-end training procedure because we found it to be more stable compared to a two-stage training procedure that first trains the encoder-decoder using a self-reconstruction loss and, after that, trains the processor to learn the latent dynamics (Yin et al., 2023; Serrano et al., 2024). The end-to-end training consists of a curriculum strategy that slowly increases the trajectory length during training (Li et al., 2023a) and a randomized starting points strategy (Brandstetter et al., 2022; Hagnberger et al., 2024) that randomly samples ICs along the trajectory. Besides future timesteps, the model predicts the IC, which we incorporate as self-reconstruction into the loss function.

## 5 Related Work

We give an overview of surrogate models, introduce continuous convolutional neural networks, and present reduced-order PDE-solving models. We refer to Appendix F for details on related work.

**PDE Solving with Neural Surrogate Models.** Neural surrogate models are increasingly used for approximating the solutions of PDEs (Lu et al., 2019; Li et al., 2021b, 2020; Cao, 2021; Li et al., 2023a; Hagnberger et al., 2024). Neural operators (Kovachki et al., 2023) constitute an important category, and prevalent neural operator models include the Graph Neural Operator (GNO; Li et al. (2020)) based on message passing (Gilmer et al., 2017) and the Fourier Neural Operator (FNO; Li et al. (2021b)), which leverages Fourier transforms, alongside its various variants. Transformers are also increasingly employed for surrogate modeling, typically using the attention mechanism on the spatial dimension of the PDE (Cao, 2021; Li et al., 2023a; Hao et al., 2023; Li et al., 2023b). Another distinct group of models, neural fields, is particularly well-suited for solving PDEs due to their ability to represent spatial functions. These models are gaining traction in PDE-solving applications (Yin et al., 2023; Chen et al., 2023; Serrano et al., 2023; Hagnberger et al., 2024; Knigge et al., 2024).

**Point Clouds and Continuous Convolution.** Solving fluid dynamics problems can be interpreted as a dense prediction problem for point clouds. Models that apply convolution to point clouds either use a discrete filter and extend it to a continuous domain via interpolation or binning (Hua et al., 2018) or parameterize the filter with a continuous function (Wang et al., 2018; Xu et al., 2018; Wu et al., 2019). These models were mainly introduced for the segmentation of point clouds. Ummenhofer et al. (2019) adapt continuous convolution to simulate Lagrangian fluids, where the model predicts the movement of the particles. In contrast to our method, their model parametrizes the filters in a discrete fashion and the weights are interpolated to extend it to a continuous domain, while our approach follows the method proposed by Wang et al. (2018) and uses an MLP to parametrize the filters. Winchenbach & Thuerey (2024) define the kernel function using separable basis functions and employ the Fourier series as the basis with even and odd symmetry for particle-based simulations.

**Neural Networks for Reduced-Order PDE-Solving.** Neural networks for reduced-order PDE-solving usually compress the spatial domain into a smaller space and solve the dynamics in that smaller space. LE-PDE (Wu et al., 2022) employs CNNs for encoding and decoding and an MLP to compute the dynamics. PIT (Chen & Wu, 2024) compresses the spatial domain into a predefined latent grid with position attention that computes attention weights based on the Euclidean distance. UPT (Alkin et al., 2024) supports simulations in the Eulerian and Lagrangian representations and employs a hierarchical structure to encode the input. The encoder uses message passing to aggregate information in super nodes, and a Transformer and Perceiver (Jaegle et al., 2021) to further distill the information into latent tokens. A Transformer computes the dynamics in the latent space, and the Perceiver-based decoder decodes the processed latent tokens. AROMA (Serrano et al., 2024) utilizes cross-attention and learnable query tokens to distill information from the spatial input into query tokens. It uses denoising diffusion to map from one latent representation to the subsequent latent representation by denoising the latent tokens. LNO (Wang & Wang, 2024) encodes the spatial input into a latent space by computing cross-attention between the input positions and learnable query positions in a high-dimensional space. Since the query positions in LNO are learnable, they simplify the attention computation, which makes the query positions inaccessible. They use a Transformer to compute the next timestep in the latent space, and a decoder decodes the latent tokens back to the physical space.

## 6 Experiments and Evaluation

We focus on solving IVPs and evaluate the performance of CALM-PDE on a set of various PDEs in fluid dynamics. The experiments are designed to answer the following Research Questions (RQ):
● **RQ1:** How effective is CALM-PDE compared to the state-of-the-art methods for regularly sampled spatial points? ● **RQ2:** How well does CALM-PDE perform on irregularly sampled points? ● **RQ3:** Does solving the dynamics in a compressed latent space yield a lower memory consumption and provide a speedup? ● **RQ4:** Where does the model place the query points and do they learn local or global information? ● **RQ5:** Which components contribute to the model's performance?

### 6.1 Datasets

For regularly sampled spatial domains, we conduct experiments on the 1D Burgers' equation dataset from PDEBench (Takamoto et al., 2022), the 2D Navier-Stokes equation datasets introduced by Li et al. (2021b), and the 3D compressible Navier-Stokes dataset of Takamoto et al. (2022). The 2D

Table 1: Rel. L2 errors of models trained and tested on regular meshes. Values in parentheses indicate the percentage deviation to CALM-PDE and underlined values indicate the second-best errors.

| Model | Relative L2 Error ($\downarrow$) | | | |
|---|---|---|---|---|
| | 1D Burgers' $\nu = 1e^{-3}$ | 2D Navier-Stokes $\nu = 1e^{-4}$ | 2D Navier-Stokes $\nu = 1e^{-5}$ | 3D Navier-Stokes $\eta = \zeta = 1e^{-8}$ |
| FNO | 0.0358 (+46%) | 0.0811 (+169%) | 0.0912 (-12%) | 0.6898 (+2%) |
| F-FNO | 0.0362 (+47%) | 0.0863 (+187%) | 0.0844 (-18%) | **0.6466** (-4%) |
| OFormer | 0.0575 (+134%) | 0.0380 (+26%) | 0.1938 (+88%) | 0.6719 (-1%) |
| PIT | 0.1209 (+391%) | 0.0467 (+55%) | 0.1633 (+58%) | 0.7423 (+10%) |
| LNO | 0.0309 (+26%) | 0.0384 (+28%) | **0.0789** (-24%) | 0.7063 (+4%) |
| AROMA | 0.0937 (+281%) | 0.1061 (+252%) | 0.1931 (+87%) | 1.3328 (+97%) |
| CALM-PDE | **0.0246** | **0.0301** | 0.1033 | 0.6761 |

Euler equation with an airfoil geometry and the incompressible Navier-Stokes equation with cylinder geometries datasets from Pfaff et al. (2020) are used to evaluate the models on irregularly sampled domains. We refer to Appendix H for additional details on the datasets.

## 6.2 Baseline Models

We compare the CALM-PDE model against FNO (Li et al., 2021b) and Geo-FNO (Li et al., 2023c) for irregularly sampled meshes, F-FNO (Tran et al., 2023), OFormer (Li et al., 2023a), PIT (Chen & Wu, 2024), LNO (Wang & Wang, 2024), and AROMA (Serrano et al., 2024). We train the autoregressive models (FNO, F-FNO, Geo-FNO, and PIT) with a curriculum strategy that slowly increases the trajectory length, which improves the error compared to training with a full autoregressive rollout (see Appendix I). OFormer uses a similar strategy, LNO a one-step training, and AROMA a denoising diffusion training to learn the temporal dynamics of the PDE as proposed by the authors.

## 6.3 Results

We report the mean values and percentage deviations of the relative L2 error of multiple runs with different initializations. We opt for the relative L2 error as an evaluation metric because it weights channels with small and large magnitudes equally and does not ignore time-dependent decay effects like in the 1D Burgers' equation. We refer to Appendix N.1 for the full results with standard deviations and Appendix N.2 for qualitative results.

**RQ1.** We train and evaluate the models on regularly sampled spatial domains. Regularly sampled spatial domains refer to meshes with equidistant nodes or points. We select datasets that span the entire spatial dimension from 1D to 3D. Table 1 shows the relative L2 errors of the baselines and CALM-PDE model. On 2 out of 4 benchmark problems, CALM-PDE outperforms all baselines. On one problem, CALM-PDE achieves the third-lowest errors and on the remaining benchmark problem, CALM-PDE is outperformed by F-FNO and LNO, and outperforms PIT and OFormer.

**RQ2.** CALM-PDE is also designed to support irregularly sampled spatial domains. Irregularly sampled domains are characterized by a mesh with varying distances between the nodes. Geometries mainly cause this irregularity because the mesh is usually denser in regions of interest, such as the tip of an airfoil, where a higher accuracy is required. We train and evaluate the models on two datasets, namely the airfoil and cylinder datasets. The first dataset simulates the airflow around a static airfoil geometry. The geometry can be considered static because it does not change between different training and test samples. In the second dataset, the water flow in a channel with a cylinder as an obstacle is simulated. In contrast to the airfoil geometry, the cylinder geometry changes, which means that each training and test sample has a different cylinder geometry with a different diameter and position. Table 2 shows the relative L2 errors for both benchmark problems. On the airfoil dataset, AROMA achieves the lowest error overall, while our model outperforms Transformer-based baselines such as OFormer and LNO, and on the cylinder dataset, CALM-PDE delivers the second-best performance, closely trailing AROMA. Thus, CALM-PDE not only supports irregularly sampled domains but also generalizes across different geometries.

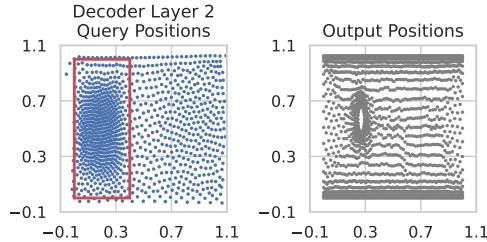

Figure 3: Learned query positions of CALM-PDE. The model samples more query points in the region where the cylinders are located (red rectangle) by moving query points to this region.

| Model | Relative L2 Error (↓) | |
|---|---|---|
| | 2D Airfoil | 2D Cylinder |
| Geo-FNO | 0.0388 (-25%) | 0.1383 (+17%) |
| F-FNO | 0.1081 (+110%) | 0.1490 (+26%) |
| OFormer | 0.0520 (+1%) | 0.2264 (+91%) |
| PIT | 0.0894 (+74%) | 0.1400 (+18%) |
| LNO | 0.0582 (+13%) | 0.1654 (+39%) |
| AROMA | **0.0372** (-28%) | **0.1139** (-4%) |
| CALM-PDE | 0.0515 | 0.1186 |

Table 2: Relative L2 errors of models trained and tested on the irregular meshes with geometries in the fluid flow. The values in parentheses indicate the percentage deviation to CALM-PDE.

**RQ3.** We measure the inference times on an NVIDIA A100 GPU to evaluate the efficiency of CALM-PDE. Figure 4 shows the inference times measured on the 2D Navier-Stokes dataset with a resolution of $64 \times 64$ and 200 trajectories. We increase the trajectory length to measure the scaling behavior. CALM-PDE is significantly faster than OFormer and LNO and achieves a competitive inference time to that of FNO and PIT, which are fast baselines due to the use of the Fast Fourier Transform (FFT) and position attention, respectively. Table 3 shows the time per epoch and memory consumption for the forward and backward pass during training on 2D Navier-Stokes with a batch size of 32. CALM-PDE outperforms OFormer and LNO in terms of time and memory consumption.

**RQ4.** Further, we investigate the learned positions of the query points and the represented information (local or global information). The results show that the model places more query points in regions that can intuitively be considered important and that the tokens primarily learn local information. Figure 3 illustrates the learned query positions for the decoder on the cylinder dataset, showing that the model samples more query points in areas where the cylinders are located. We refer to Appendix L for more details. Thus, learnable query points allow the model a higher information density in important regions and allow learning and discovering unknown important regions.

**RQ5.** Finally, we analyze the impact of the model's components in the ablation study presented in Appendix M. The results indicate that learnable query points reduce the error more effectively than both fixed and randomly sampled query points, as well as fixed query points sampled from the underlying mesh. Additionally, the kernel modulation improves the error for regularly and irregularly sampled spatial domains. Furthermore, the distance weighting, which is part of the locality inductive bias, also helps the model to further reduce the error and stabilize the training.

## 7 Limitations

CALM-PDE compresses the input into a smaller latent space, which implies that information such as fine-grained details is lost. This could be an issue for PDEs such as Kolmogorov flow that have fine details that must be accurately captured. Another limitation, that also applies to other neural PDE solvers, are the required computational resources for real-world applications with large spatial domains with millions of spatial points. Compared to the Transformer-based methods, CALM-PDE is more efficient but not efficient enough for practical applications. For the 3D Navier-Stokes equation with 21 timesteps and a spatial resolution of $64 \times 64 \times 64$, which corresponds to 262k points, CALM-PDE requires one A100 80GB GPU for the training, while LNO requires four A100 80GB GPUs with a batch size of 4, respectively. However, practical applications would require spatial resolutions larger than $64 \times 64 \times 64$.

## 8 Conclusion and Future Work

With CALM-PDE, we propose an efficient framework for reduced-order modeling of arbitrarily discretized PDEs. The experiments demonstrate that CALM-PDE achieves low errors on regularly

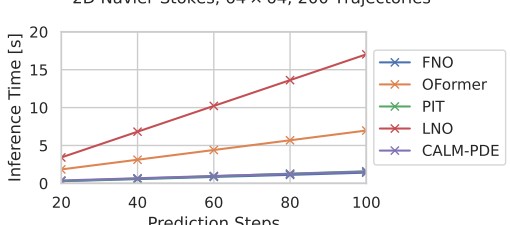

**2D Navier-Stokes, 64 × 64, 200 Trajectories**

| Model | 2D Navier-Stokes | |
|---|---|---|
| | Time/Batch [ms] | Memory [MB] |
| FNO | $126.17^{\pm 4.38}$ | **5453** |
| OFormer | $786.36^{\pm 8.67}$ | 43807 |
| PIT | $\mathbf{124.23}^{\pm 9.16}$ | 16968 |
| LNO | $1196.57^{\pm 0.57}$ | 61191 |
| CALM-PDE | $138.25^{\pm 4.73}$ | 14147 |

Figure 4: Comparison of inference times on 2D Navier-Stokes with 200 trajectories. Prediction steps are increased to evaluate the scaling. FNO, PIT, and CALM-PDE achieve similar times.

Table 3: Time and memory consumption needed for the forward and backward pass during training on the 2D Navier-Stokes dataset for a batch size of 32 on an NVIDIA A100 GPU.

and irregularly discretized PDEs, as well as that the model can generalize to different geometries. While it does not consistently surpass all baselines, it delivers competitive errors across various problems. This makes it a viable alternative to established approaches such as FNOs and transformer-based architectures and demonstrates that convolution can also be applied to irregular meshes. Beyond PDEs, CALM layers have broader applicability, including potential use in fields like chemistry. For future work, we aim to improve CALM-PDE for problems with high frequencies, which are important in real-world applications. Additionally, future work could investigate "dynamic query points" that depend on the IC and geometry or move during the rollout to track important regions. Another interesting direction for future research is improving the model for a two-stage training procedure.

## Acknowledgments and Disclosure of Funding

Funded by Deutsche Forschungsgemeinschaft (DFG, German Research Foundation) under Germany's Excellence Strategy - EXC 2075 – 390740016. We acknowledge the support of the Stuttgart Center for Simulation Science (SimTech). The authors thank the International Max Planck Research School for Intelligent Systems (IMPRS-IS) for supporting Daniel Musekamp and Mathias Niepert. Additionally, we acknowledge the support of the German Federal Ministry of Research, Technology and Space (BMFTR) as part of InnoPhase (funding code: 02NUK078). Lastly, we acknowledge the support of the European Laboratory for Learning and Intelligent Systems (ELLIS) Unit Stuttgart.

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

# APPENDIX FOR CALM-PDE

# A Impact Statement

Advanced and efficient neural surrogate models significantly lower the expenses of running otherwise cost-intensive simulations, such as those used in weather forecasting and fluid mechanics. This improvement helps to speed up simulations and to reduce energy consumption, costs, and $CO_2$ emissions. However, a potential drawback is the risk of misuse by bad actors, as fluid dynamic simulations play a role in designing military equipment.

# B Details on the Problem Definition

**Discretized Dataset.** We use discretized data generated by numerical solvers to train and test the surrogate models. The temporal domain $[0, T]$ is discretized into $N_t$ timesteps yielding a sequence $(u^0, u^{t_1}, \ldots, u^T)$ with $N_t$ elements which describes the evolution of the PDE. $\Delta t := t_{i+1} - t_i$ denotes the temporal step size or resolution. Similarly, the spatial domain $\Omega$ is also discretized into a finite set of N points $\{\omega_n\}_{n=1}^N$ by discretizing each spatial dimension which yields a discretized representation $\{u^t(\omega_n)\}_{n=1}^N$ of the function. Figure 5 visualizes the discretization process. A dataset $\mathcal{D} = \{(\boldsymbol{X_1}, \boldsymbol{Y_1}), \ldots, (\boldsymbol{X_{N_s}}, \boldsymbol{Y_{N_s}})\}$ for each PDE consists of $N_s$ samples. $\boldsymbol{X_j} = \{u^0(\omega_n)\}_{n=1}^N$ denotes the IC and $\boldsymbol{Y_j} = (\{u^{t_1}(\omega_n)\}_{n=1}^N, \ldots, \{u^T(\omega_n)\}_{n=1}^N)$ denotes the target sequence of timesteps.

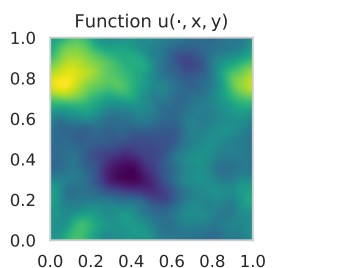
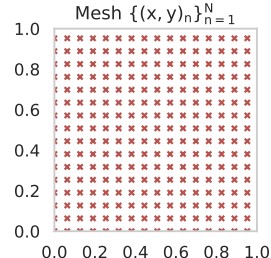
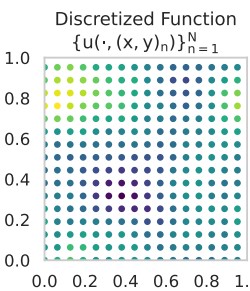

Figure 5: The continuous solution function $u(\cdot, x, y)$ has to be discretized along the spatial dimension with a suitable mesh or grid into a discrete representation $\{u(\cdot, (x, y)_n)\}_{n=1}^N$ of the function.

**Training Objective.** The training objective aims to optimize the parameter vector $\boldsymbol{\theta}$ that contains all weights and biases of the model $f_{\boldsymbol{\theta}}$ that best approximates the true function $u$ by minimizing the empirical risk over the dataset $\mathcal{D}$ as

$$\underset{\boldsymbol{\theta} \in \Theta}{\arg\min} \, \mathcal{L}_{\boldsymbol{\theta}} = \underset{\boldsymbol{\theta} \in \Theta}{\arg\min} \, \frac{1}{N_s} \sum_{s=1}^{N_s} L\Big(f_{\boldsymbol{\theta}}\big(\{t_i\}_{i=1}^{N_t}, \{\omega_j\}_{j=1}^N \mid \boldsymbol{X_s}, \{\omega_n\}_{n=1}^N\big), \boldsymbol{Y_s}\Big) \tag{10}$$

where $\mathcal{L}_{\boldsymbol{\theta}}$ denotes the overall cost function and $L$ denotes a suitable loss function such as the Mean Squared Error (MSE) or relative L2 norm. $f_{\boldsymbol{\theta}}\big(\{t_i\}_{i=1}^{N_t}, \{\omega_j\}_{j=1}^N \mid \boldsymbol{X_s}, \{\omega_n\}_{n=1}^N\big)$ represents the predicted trajectory of the neural network queried with the set of times $\{t_i\}_{i=1}^{N_t}$ and the set of spatial points $\{\omega_j\}_{j=1}^N$, given the initial condition $\boldsymbol{X_s}$ evaluated at the points $\{\omega_n\}_{n=1}^N$.

# C Discrete Convolution and Cross-Correlation

We briefly explain discrete convolution and elaborate on how discrete convolution is used in CNNs.

**Discrete Convolution and Cross-Correlation.** In the discrete case (i.e., the functions $f$ and $k$ are only defined for integers), the convolution operator simplifies to

$$(f * k)[a] = \sum_{\alpha=-\infty}^{\infty} f[a - \alpha] \cdot k[\alpha] \tag{11}$$

with $a \in \mathbb{Z}$. The points usually have equal spacing because discrete convolution operates on sequences or discrete signals with regularly indexed data points. Similarly to the continuous case, $k[\alpha]$ has to be reflected to $k[-\alpha]$ to compute cross-correlation.

**Convolutional Neural Networks.** CNNs implement cross-correlation where $f$ is the finite and discrete input signal (i.e., the input features) with a length of $N$, and $k$ represents a finite and learnable kernel with the length $M$. This leads to

$$(f * k)[a] \stackrel{\text{reflect } k}{=} (f \star k)[a] = \sum_{\alpha=1}^{M} f[a + \alpha - 1] \cdot k[\alpha] \tag{12}$$

with $a \in [1, N - M + 1]$. As mentioned previously, cross-correlation is equivalent to convolution with a reflected kernel. Since the weights of the kernel are learned during training, it does not matter whether the layer implements cross-correlation or convolution. Usually, the input contains multiple input features or channels (i.e., is a vector) and multiple convoluted output features or channels are desired. $N_i$ and $N_o$ denote the number of input and output channels respectively which yields

$$(f * k)_o[a] = \sum_{i=1}^{N_i} \sum_{\alpha=1}^{M} f_i[a + \alpha - 1] \cdot k_{i,o}[\alpha] \tag{13}$$

where $o$ corresponds to the the $o^{th}$ output feature or channel and $i$ to the $i^{th}$ input channel and $k[\alpha] \in \mathbb{R}^{N_i \times N_o}$ contains the filter kernels. $k[\alpha]$ is usually implemented using a finite, multi-dimensional array. The convolution layer as introduced above can be extended from 1D to d-D.

# D   Deep Parametric Continuous Convolutional Neural Networks

Figure 6 visualizes continuous convolution with and without a receptive field for 4 input points $\alpha_n$ and one query point $a_1$. As proposed by Wang et al. (2018), the kernel function $k$ is parametrized by an MLP. However, continuous convolution can be optimized further for the encoding and decoding of PDE solutions. For instance, CALM layers learn the position of the query points $a_j$ to effectively sample the spatial domain, use a locality inductive bias including an explicit weighting with the Euclidean distance to emphasize the input points closer to the query point and an epsilon neighborhood to improve efficiency, and each query point modulates the MLP to support different kernels for query points that have similar translation vectors.

$$(f * k)_o(a_1) = \sum_{i=1}^{4} \sum_{n=1}^{4} f(\alpha_n)_i \cdot k_{i,o}(a_1 - \alpha_n) \qquad\qquad (f * k)_o(a_1) = \sum_{i=1}^{4} \sum_{m \in \mathbf{RF}(a_1)} f(\alpha_m)_i \cdot k_{i,o}(a_1 - \alpha_m)$$

(a) Without receptive field                    (b) With receptive field (blue circle)

Figure 6: Continuous convolution without receptive field (a) and with a receptive field (b) that limits the number of considered input points. The kernel function $k$ takes as input the translation vector $a - \alpha$ and outputs a weight for an input channel $i$ and output channel $o$. An MLP parametrizes the kernel function.

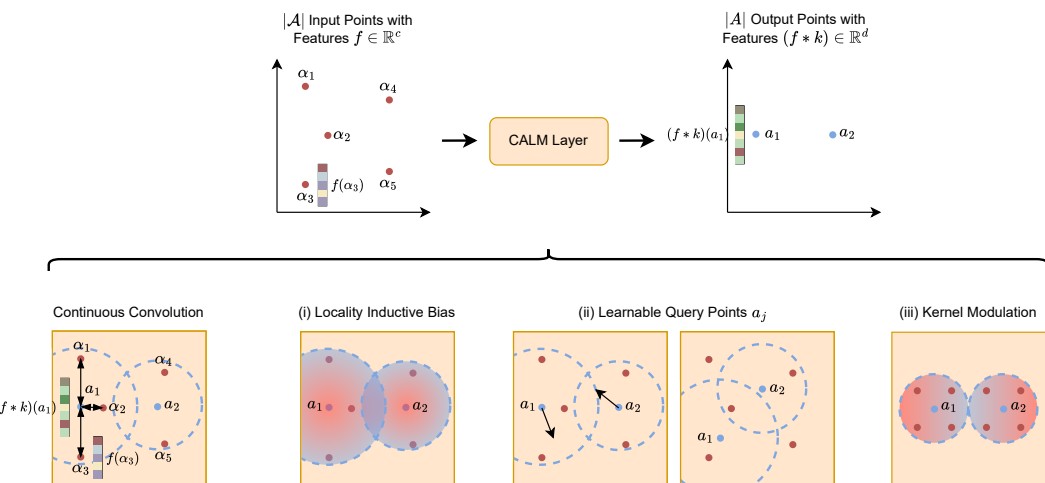

Figure 7: A CALM layer takes a point cloud with $|\mathcal{A}|$ points as input and outputs a new point cloud with $|A|$ points and a new feature dimension $d$. CALM layers compute continuous convolution with a locality inductive bias to emphasize closer points, learnable query points (output positions), and a kernel modulation to allow different weights for different query points with similar translation vectors.

# E    Additional Details on the CALM-PDE Framework

## E.1    CALM Layer

**Overview.**    Figure 7 shows a CALM layer that performs continuous convolution with three distinct features: (i) locality inductive bias, consisting of a limited receptive field and distance weighting, to emphasize closer points, (ii) learnable query points to let the model learn where to compute convolution, and (iii) kernel modulation to allow different query points to have different weights for similar translation vectors. Depending on the purpose (e.g., encoder or final layer of the decoder), the query points can be learned or provided as an external input query. The kernel function $k$ is given as

$$k_{i,o}(a-\alpha) = \underbrace{\left(\boldsymbol{W_2} \cdot \sigma\Big(\big(\boldsymbol{W_1} \cdot \mathrm{RFF}(a-\alpha) + \boldsymbol{b_1}\big) \odot \boldsymbol{\gamma}_a + \boldsymbol{\beta}_a\Big) + \boldsymbol{b_2}\right)_{i,o}}_{\text{Modulated MLP}} \underbrace{\frac{\exp\big(-\frac{\|a-\alpha\|^2-\min(a)}{T(\max(a)-\min(a))}\big)}{\sum\limits_{\alpha_j\in\mathrm{RF}(a)}\exp\big(-\frac{\|a-\alpha_j\|^2-\min(a)}{T(\max(a)-\min(a))}\big)}}_{\text{Distance Weighting}}$$

(14)

where $\boldsymbol{\gamma}_a$ and $\boldsymbol{\beta}_a$ denote the modulation parameters which depend on the query point $a$, and $\min(a) = \min_{\alpha_j\in\mathrm{RF}(a)}\|a-\alpha_j\|^2$ computes the minimum distance within the epsilon neighborhood. Similarly, $\max(a)$ computes the maximum distance within the epsilon neighborhood. The distance normalization ensures that small distances are amplified and the effect of large distances is reduced.

**Implementation of Kernel Function.**    In practice, the kernel function $k$ is parametrized by a 2-layer MLP that outputs a large vector in $\mathbb{R}^{N_i \cdot N_o}$ followed by a reshape operation to get a kernel matrix in $\mathbb{R}^{N_i \times N_o}$. $N_i$ denotes the number of input channels and $N_o$ is the number of output channels. The bias term $\boldsymbol{b_2}$ is initialized with Kaiming uniform initialization (He et al., 2015) where the number of input features corresponds to the number of input channels $N_i$ and not to the hidden dimension of the MLP. We opt for this initialization to ensure proper initialization of the weights used in the continuous convolution operation. This approach can also be understood as initializing a learnable weight $\bar{k}_{i,o}$, which is independent of the translation $a - \alpha$ and shared between all points, with Kaiming uniform initialization and using a 2-layer MLP without a bias in the last layer to learn an additive correction or residual depending on the input $a - \alpha$ to parameterize the continuous kernel function $k_{i,o}(a - \alpha)$.

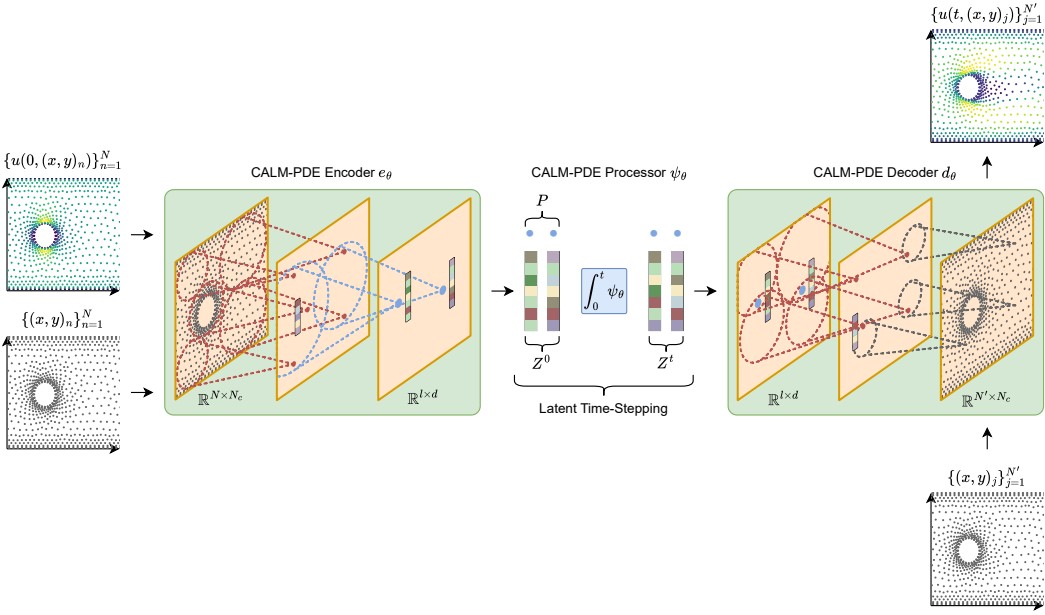

Figure 8: The CALM-PDE model utilizes multiple CALM layers with learnable query positions to encode arbitrarily discretized PDE solutions into a fixed latent space. A Transformer processes the compressed latent representation and outputs the latent representations of future timesteps which will be decoded by the decoder. The decoder also uses multiple CALM layers with learnable query positions, except for the last decoder layer which uses the mesh as an external input query.

**Receptive Field and Epsilon Neighborhood.** The receptive field of a query point $a_j$ is given as $\mathrm{RF}(a_j) = \{\alpha_n \in \mathcal{A} \mid \|a_j - \alpha_n\| \leq \epsilon(a_j)\}$ where $\epsilon(a_j)$ is the $p^{th}$ percentile of the Euclidean distances from query point $a_j$ to all input points $\alpha_n \in \mathcal{A}$ similar to Chen & Wu (2024). The hyperparameter $p$ controls the size of the receptive field. For instance, $p = 0.1$ means that the epsilon neighborhoods are constructed so that each query point $a_j$ aggregates at least 10% of all input points. Consequently, the epsilon neighborhood is smaller in densely sampled regions than in sparsely sampled regions which avoids convolution or aggregation with too many or few input points.

**Pointwise Operations.** Before and after applying the continuous convolution operation, a pointwise linear operation and a pointwise MLP are applied, respectively. The pointwise operations enable the model to combine the input or output channels of continuous convolution to a richer feature representation.

**Computational Complexity.** The complexity of a CALM layer is given as follows. Let $N$ be an arbitrary number of input points. The output of the layer are features for $N'$ output or query points, where $N'$ is constant for all layers except for the final decoder layer. The percentile, which determines the size of the receptive field, and the feature or channel dimension of the CALM layer are denoted as $p$ and $d$, respectively. This results in $\mathcal{O}(p \cdot N \cdot N' \cdot d)$ computations in total. In contrast, models such as AROMA (Serrano et al., 2024) or LNO (Wang & Wang, 2024) do not consider a receptive field, which requires $\mathcal{O}(N \cdot N' \cdot d)$ computations for $N'$ query tokens. Since $p$ is usually small (e.g., $p = 0.01$), the complexity is reduced by two orders of magnitude compared to models without a receptive field.

## E.2 Neural Architecture

In this section, we provide additional details on the neural architecture of CALM-PDE. Figure 8 shows the architecture of the model with the encoder, processor, and decoder.

**Processor for Latent Time-Stepping.** The processor network $\psi_\theta(Z^t, P)$ is a Transformer with combined attention that combines scaled dot-product attention (Vaswani et al., 2017) and position

attention (Chen & Wu, 2024). It takes the latent representation $Z^t$ for time $t$ and the positions $P$ as input and computes the change between the latent representations $Z^t$ and $Z^{t+1}$ as follows

$$
\begin{aligned}
H_1 &= \big(Z^t \| \mathrm{RFF}(\mathrm{P})\big) W^{\mathrm{In}} \\
H_2 &= \mathtt{CombinedAttn}(H_1, P) + H_1 \\
H_3 &= \mathtt{MLP}(H_2, P) + H_2 \\
H_4 &= \mathtt{CombinedAttn}(H_3, P) + H_3 \\
H_5 &= \mathtt{MLP}(H_4, P) + H_4 \\
\psi_\theta(Z^t, P) &= H_5 W^{\mathrm{Out}}
\end{aligned}
\tag{15}
$$

$$
Z^{t+1} = \Delta t \cdot \psi_\theta(Z^t, P) + Z^t
$$

$$
\mathtt{CombinedAttn}(\mathrm{H}, \mathrm{P}) = \mathtt{softmax}\big(\frac{1}{\sqrt{d}}\mathtt{LN}(HW^Q)\mathtt{LN}(HW^K) - d(P, P)\big)HW^V
$$

where $d(P, P)$ outputs a distance matrix with the pairwise distances of the positions $P$. $\mathtt{LN}$ denotes layer normalization (Ba et al., 2016) and $\Delta t$ is the temporal resolution. We only normalize the queries and keys to allow the propagation of the scaling of the features through the processor similar to Cao (2021). The processor is applied iteratively to compute the latent representation of timesteps further into the future. Due to the residual connection and the scaling of the predicted change with the temporal step size $\Delta t$, the prediction $Z^{t+1} = \Delta t \cdot \psi_\theta(Z^t, P) + Z^t$ can be interpreted as solving a neural ordinary differential equation parametrized by $\psi_\theta$ with an explicit Euler solver. The exact solution is given as

$$
Z(t + \Delta t) = Z(t) + \int_t^{t+\Delta t} \psi_\theta(Z(\tau), P)d\tau
\tag{16}
$$

which can be approximated with an explicit Euler solver as follows

$$
\begin{aligned}
Z(t + \Delta t) &= Z(t) + \int_t^{t+\Delta t} \psi_\theta(Z(\tau), P)d\tau \\
&\approx Z(t) + \Delta t \cdot \psi_\theta(Z(t), P) \\
&= Z^t + \Delta t \cdot \psi_\theta(Z^t, P)
\end{aligned}
\tag{17}
$$

where $\Delta t$ denotes the temporal step size.

## F   Continuation of Related Work

**Latent Neural Surrogates and Latent Time-Stepping.**   We introduce the term Latent Neural Surrogates (LNS) to refer to models that internally use a smaller spatial latent representation instead of the original spatial representation, such as Position-Induced Transformer (PIT; Chen & Wu (2024)) and Latent Neural Operator (LNO; Wang & Wang (2024)). We use the term Latent Time-Stepping (LTS) to describe models that solve the dynamics completely in the latent space without decoding it in every timestep. In contrast, models like PIT and LNO follow an autoregressive rollout in the physical space, meaning they decode the solution in every timestep and do not belong to the LTS category. Prevalent LTS models include LE-PDE (Wu et al., 2022), which employs a CNN for both encoding and decoding and an MLP as the latent processor. OFormer (Li et al., 2023a) utilizes a Transformer as encoder and decoder and a pointwise MLP on the spatial dimension as processor. UPT (Alkin et al., 2024) applies attention for encoding and decoding and a Transformer for latent time-stepping. AROMA (Serrano et al., 2024) uses cross-attention as encoder and decoder and a diffusion Transformer as latent processor. The proposed CALM-PDE model belongs to both model categories. It is an LNS as it uses the proposed CALM layers for the encoder and decoder to compress the solution and it is an LTS model since it solves the dynamics completely in the latent space with a Transformer.

**Discretization-Agnostic Architectures.**   Discretization-agnostic architectures for PDE-solving decouple the input and output spatial domains and allow solving arbitrarily discretized PDEs. Transformer-based models such as OFormer, UPT, AROMA, and LNO are discretization-agnostic.

Variants such as Geo-FNO (Li et al., 2023c) of the Fourier Neural Operator (Li et al., 2021b) are also discretization-agnostic and support arbitrarily discretized inputs. Neural fields are also suitable for discretization-agnostic models and have been used in DINo (Yin et al., 2023), CORAL (Serrano et al., 2023), and Equivariant Neural Fields (Knigge et al., 2024) for PDE-solving. CALM-PDE is based on continuous convolution and also supports the encoding and decoding of arbitrarily discretized and irregularly sampled solutions. Furthermore, CALM-PDE decouples the input from the output domain.

**Point Clouds.** Solving fluid mechanics problems in the Eulerian or Lagrangian representation can be interpreted as a dense prediction problem for point clouds. Models such as PointNet (Qi et al., 2017a) and PointNet++ (Qi et al., 2017b) have shown great success in the classification and semantic segmentation of point clouds. PointNet is based on pointwise transformations and a max pooling operation to extract a global representation of the point cloud. They concatenate the point features with the global representation to obtain a rich feature vector with local and global information for semantic segmentation. PointNet++ extends PointNet with a hierarchical structure to gradually combine regions into larger ones.

# G    Comparison to Related Models

We compare related models such as OFormer (Li et al., 2023a), PIT (Chen & Wu, 2024), UPT (Alkin et al., 2024), AROMA (Serrano et al., 2024), and LNO (Wang & Wang, 2024) with the proposed CALM-PDE model. We briefly introduce the models and elaborate on similarities and differences.

PIT (Chen & Wu, 2024) compresses the spatial domain into a smaller latent representation with position attention, computes the dynamics for one timestep, and decodes it back with position attention. They propose an attention mechanism called position attention that computes the attention weights of the input points and the latent points based on the Euclidean distance instead of the scaled-dot product (Vaswani et al., 2017). A predefined set of points is used in the latent space as a latent mesh. UPT (Alkin et al., 2024) supports simulations in the Eulerian and Lagrangian representations and employs a hierarchical structure to encode the input. The encoder uses message passing to aggregate information in super nodes, and a Transformer and Perceiver (Jaegle et al., 2021) to further distill the information into latent tokens. A Transformer computes the dynamics completely in the latent space (latent time-stepping), and the Perceiver-based decoder decodes the processed latent tokens. AROMA (Serrano et al., 2024) utilizes cross-attention and learnable query tokens to distill information from the spatial input into the query tokens. The decoding is also realized with cross-attention. The model is trained in a two-stage training. First, the encoder-decoder model is trained as a variational autoencoder (Kingma & Welling, 2022). After that, a denoising diffusion Transformer (Peebles & Xie, 2023) is trained to map from one latent representation to the subsequent latent representation by denoising the latent tokens. LNO (Wang & Wang, 2024) encodes the spatial input into a fixed latent space by computing cross-attention between the input positions and learnable query positions in a high-dimensional space. Thus, the attention weights depend only on the positions, similar to PIT and AROMA. Since the query positions in LNO are learnable, they simplify the attention computation, which makes the query positions inaccessible. They use a Transformer to compute the next timestep in the latent space, and a decoder decodes the latent tokens back to the physical space. The LNO model is trained with teacher forcing and used for inference in an autoregressive fashion, both in the physical space.

CALM-PDE learns query points to effectively sample the spatial domain, where each point has a specific position, giving the learned query points a tangible physical meaning. In contrast, AROMA and LNO also implicitly learn query points for cross-attention, but the query points are inaccessible and lack physical relevance, as they are defined in a high-dimensional space. The physical meaning of CALM-PDE's query points enables the incorporation of a mesh prior. For example, the query points can be initialized to be denser in predefined critical regions and sparse in less significant regions. This concept is similarly applied in PIT, where the latent space is structured on a predefined latent mesh. However, predefined query points are not always optimal, and the ability to learn query positions provides greater flexibility. This fundamental distinction sets CALM-PDE apart from PIT, which operates without learnable query points. The key distinctions from PIT include the use of continuous convolution, learnable query positions, a kernel MLP for computing convolution weights, and multiple filters. AROMA and CALM-PDE differ in the encoder and decoder (scaled dot-product

attention compared to continuous convolution), latent processor, and training procedure. UPT and CALM-PDE differ in the encoder and decoder architecture (message passing and attention compared to continuous convolution). Similarly, LNO and CALM-PDE differ in the use of attention compared to continuous convolution in the encoder and decoder. PIT, UPT, and CALM-PDE share the approach of restricting the receptive field to enforce locality and optimizing computational efficiency. AROMA and LNO do not restrict the receptive field of the query tokens. In contrast to the mentioned methods, CALM-PDE does not lift the input channels into a higher-dimensional space, which is required for the Transformer-based models. Due to multiple filters, our model can still extract enough features without the need to operate in such a high-dimensional space.

Table 4 provides an overview of similarities and differences between related work and the CALM-PDE framework. We compare the following properties:

- Supports Compression: Whether the model supports compressing the solution into a smaller latent space.

- Latent Time-Stepping: Whether the models compute the dynamics completely in the latent space.

- Hierarchical Encoder and Decoder: Whether the models use a hierarchical encoder and or decoder.

- Attention or Kernel Weights solely based on Positions: Whether the models compute attention or kernel weights solely based on the position of the sampled points.

- Large Number of Filters or Heads: Whether the models can support a large number of filters or heads. For attention-based models, the number of filters is limited by the dimension of the query, key, and values, while convolution-based models do not suffer from this limitation.

- Learnable Query Points: Whether the model has learnable query points or query tokens.

- Physical Meaning of Query Points: Whether the query points or tokens have a physical meaning, such as a position that provides information about where the model samples more densely.

- No Lifting or Embedding Required: Whether it is required to lift or embed the input channels (e.g., 1 to 5 channels) into a higher-dimensional space (e.g., 96 to 256) at the beginning, which results in a higher computational complexity.

Table 4: Overview of properties of related models and the proposed CALM-PDE model.

| Property | OFormer | PIT | UPT | AROMA | LNO | CALM-PDE |
|---|---|---|---|---|---|---|
| Supports Compression | ✗ | ✓ | ✓ | ✓ | ✓ | ✓ |
| Latent Time-Stepping | ✓ | ✗ | ✓ | ✓ | ✗ | ✓ |
| Hierarchical Encoder | ✗ | ✗ | ✓ | ✗ | ✗ | ✓ |
| Hierarchical Decoder | ✗ | ✗ | ✗ | ✗ | ✗ | ✓ |
| Attention or Kernel Weights solely based on Positions | ✗ | ✓ | ✗ | ✓ | ✓ | ✓ |
| Large Number of Filters or Heads | ✗ | ✗ | ✗ | ✗ | ✗ | ✓ |
| Learnable Query Points | ✗ | ✗ | ✓ | ✓ | ✓ | ✓ |
| Physical Meaning of Query Points | ✗ | ✓ | ✗ | ✗ | ✗ | ✓ |
| No Lifting or Embedding Required | ✗ | ✗ | ✗ | ✗ | ✗ | ✓ |

## H   Additional Details on the Datasets

We benchmark the baselines and CALM-PDE model on the following datasets with regularly and irregularly sampled spatial domains. We use the term regularly sampled spatial domain to refer to spatial points sampled from a uniform grid, in contrast to the term irregularly sampled domain, which refers to a grid or mesh with non-equidistant points. Meshes are mainly irregular due to geometries or obstacles in the fluid flow. Table 5 shows an overview of the used datasets.

Table 5: Overview of the used datasets. [†]Static geometry means that the geometry is the same for each sample. In contrast, changing geometry means that the geometry changes with each sample.

| PDE | Parameter | Timesteps | Spatial Resolution | Mesh | Geometry[†] | Channels $N_c$ |
|---|---|---|---|---|---|---|
| 1D Burgers' | $\nu = 1e^{-3}$ | 41 | 1024 | Regular | No | 1 |
| 2D Navier-Stokes | $\nu = 1e^{-4}$ | 20 | $64 \times 64$ | Regular | No | 1 |
| 2D Navier-Stokes | $\nu = 1e^{-5}$ | 20 | $64 \times 64$ | Regular | No | 1 |
| 3D Navier-Stokes | $\eta = \zeta = 1e^{-8}$ | 21 | $64 \times 64 \times 64$ | Regular | No | 5 |
| 2D Airfoil | N/A | 16 | 5233 points | Irregular | Yes, static | 4 |
| 2D Cylinder | N/A | 15 | 1885 points (avg) | Irregular | Yes, changing | 3 |

## H.1 Regularly Sampled Spatial Domains

**1D Burgers' Equation.** The 1D Burgers' equation models the non-linear behavior and diffusion process of 1D flows in fluid dynamics and is defined as

$$\partial_t u(t,x) + u(t,x)\partial_x u(t,x) = \frac{\nu}{\pi}\partial_{xx} u(t,x), \qquad t \in (0,2], x \in (-1,1] \tag{18}$$

where the parameter $\nu$ denotes the diffusion coefficient. We use the dataset provided by PDEBench (Takamoto et al., 2022) and select a diffusion coefficient $\nu = 1e^{-3}$ to encourage shocks. The equation is under periodic boundary conditions and the goal is to learn neural surrogates to approximate the function $u$. We use a spatial resolution of 1024 for $x \in (-1,1)$ and subsample the temporal resolution to 41 timesteps for $t \in (0,2]$. The dataset contains 10k trajectories, with the first 1k samples used for testing and the remaining 9k samples for training.

**2D Incompressible Navier-Stokes Equation.** The Navier-Stokes equations are important for Computational Fluid Dynamics (CFD) applications. We consider the 2D Navier-Stokes equation for a viscous, incompressible fluid in vorticity form on the unit torus which is given as

$$\begin{aligned}
\partial_t v(t,\boldsymbol{\omega}) + u(t,\boldsymbol{\omega}) \cdot \nabla v(t,\boldsymbol{\omega}) &= \nu \Delta v(t,\boldsymbol{\omega}) + f(\boldsymbol{\omega}), & \boldsymbol{\omega} \in (0,1)^2, t \in (0,T] \\
\nabla \cdot u(t,\boldsymbol{\omega}) &= 0, & \boldsymbol{\omega} \in (0,1)^2, t \in [0,T] \\
v(0,\boldsymbol{\omega}) &= v_0(\boldsymbol{\omega}), & \boldsymbol{\omega} \in (0,1)^2
\end{aligned} \tag{19}$$

where $v = \nabla \times u$ denotes the vorticity, $u$ the velocity, and $\nu \in \mathbb{R}_+$ the viscosity coefficient. $v_0$ is the initial vorticity and $f$ is the forcing term. The task is to learn a neural surrogate to approximate the function $v$. We use the dataset proposed by Li et al. (2021b) with periodic boundary conditions and consider the viscosities $\nu = 1e^{-4}$ and $\nu = 1e^{-5}$. We drop the first 10 timesteps due to less complex dynamics and consider a temporal horizon of $t \in (10,30]$ (20 timesteps) for $\nu = 1e^{-4}$ and use a temporal horizon of $t \in (0,20]$ (20 timesteps) for $\nu = 1e^{-5}$. The spatial domain is discretized into a grid of $64 \times 64$. We use the last 200 samples for testing and the remaining ones for training in both cases.

**3D Compressible Navier-Stokes Equation.** Additionally, we consider a compressible version of the Navier-Stokes equations (Equations 20a to 20c). The equations describe the flow of a fluid and are defined as

$$\partial_t \rho + \nabla \cdot (\rho \mathbf{v}) = 0, \tag{20a}$$

$$\rho(\partial_t \mathbf{v} + \mathbf{v} \cdot \nabla \mathbf{v}) = -\nabla p + \eta \triangle \mathbf{v} + (\zeta + \frac{\eta}{3})\nabla(\nabla \cdot \mathbf{v}), \tag{20b}$$

$$\partial_t(\epsilon + \frac{\rho v^2}{2}) + \nabla \cdot [(p + \epsilon + \frac{\rho v^2}{2})\mathbf{v} - \mathbf{v} \cdot \sigma'] = \mathbf{0}, \tag{20c}$$

$$\rho := \rho(t,\boldsymbol{\omega}), \mathbf{v} := \mathbf{v}(t,\boldsymbol{\omega}), p := p(t,\boldsymbol{\omega}), \qquad \boldsymbol{\omega} \in \Omega, t \in [0,T] \tag{20d}$$

where $\rho$ is the density, $\mathbf{v}$ the velocity, and $p$ the pressure of the fluid. $\epsilon$ denotes the internal energy and $\sigma'$ the viscous stress tensor. The parameters $\eta$ and $\zeta$ are the shear and bulk viscosity. Equation (20a) represents the conservation of mass, Equation (20b) is the equation of the conservation of momentum, and Equation (20c) is the energy conservation. The goal is to learn a neural surrogate approximating density, velocity, and pressure. We use the dataset from PDEBench (Takamoto et al., 2023) with

$\eta = \zeta = 1e^{-8}$, a spatial resolution of $64 \times 64 \times 64$, and the full temporal resolution of 21 timesteps. The equation in the dataset is under periodic boundary conditions. We use the first 10 samples for testing and the last 90 samples for training.

## H.2 Irregularly Sampled Spatial Domains

**2D Compressible Euler Equation with Airfoil Geometry.** The 2D Euler equation describes the flow of an inviscid fluid and corresponds to the Navier-Stokes equations with zero viscosity and no heat conduction. The 2D Euler equation for a compressible fluid is defined as

$$
\begin{aligned}
\partial_t \rho + \nabla \cdot (\rho \mathbf{v}) &= s_1, \\
\rho(\partial_t \mathbf{v} + \mathbf{v} \cdot \nabla \mathbf{v}) + \nabla p &= \mathbf{s}_2, \\
\partial_t(\epsilon + \frac{\rho v^2}{2}) + \nabla \cdot [(p + \epsilon + \frac{\rho v^2}{2})\mathbf{v}] &= s_3, \\
\rho := \rho(t, \boldsymbol{\omega}), \mathbf{v} := \mathbf{v}(t, \boldsymbol{\omega}), p := p(t, \boldsymbol{\omega}), \qquad \boldsymbol{\omega} &\in \Omega, t \in [0, T]
\end{aligned}
\tag{21}
$$

where $\rho$ is the density, $\mathbf{v}$ is the velocity, and $p$ the pressure of the fluid. $\epsilon$ denotes the internal energy and $s_1, \mathbf{s}_2, s_3$ are the source terms. The task is to learn a neural surrogate model that approximates the functions $\rho$, $\mathbf{v}$, and $p$. We use the dataset of Pfaff et al. (2020) which contains the geometry of an airfoil with a no-penetration condition imposed on the airfoil. The distances between the points in the meshes range from $2e^{-4}$ m to 3.5 m, which makes it a dataset with a highly irregular sampled spatial domain. We consider a subsampled temporal resolution of 16 timesteps for the experiments.

**2D Incompressible Navier-Stokes with Cylinder Geometries.** The 2D incompressible Navier-Stokes equation with a constant density is defined as

$$
\begin{aligned}
\partial_t \mathbf{v} &= 0, \\
\rho_0(\partial_t \mathbf{v} + \mathbf{v} \cdot \nabla \mathbf{v}) + \nabla p &= \mu \nabla^2 \mathbf{v}, \\
\mathbf{v} := \mathbf{v}(t, \boldsymbol{\omega}), p := p(t, \boldsymbol{\omega}), \qquad \boldsymbol{\omega} &\in \Omega, t \in [0, T]
\end{aligned}
\tag{22}
$$

where $\rho_0$ is the constant density, $\mathbf{v}$ the velocity, and $p$ the pressure. We use the dataset introduced by Pfaff et al. (2020) that models the flow of water in a channel with a cylinder as an obstacle in the fluid flow. With each sample, the diameter and position of the cylinder change. The neural surrogates are trained to predict the velocity and pressure. We subsample the temporal resolution to 15 timesteps for the experiments.

# I  Additional Details on the Baselines

**Fourier Neural Operator (FNO) and Geo-FNO.** The Fourier Neural Operator is a neural operator based on Fast Fourier Transforms (FFTs). We use the implementation of FNO from PDEBench (Takamoto et al., 2022) and use the model in an autoregressive fashion as proposed by Li et al. (2021b) as "FNO with RNN structure". However, we use a curriculum strategy that combines autoregressive training with teacher-forcing training by slowly increasing the rollout length of the autoregressive rollout (Li et al., 2023a) and by doing a teacher-forcing prediction with the remaining timesteps of the trajectory (Takamoto et al., 2023). The strategy improves the performance compared to full autoregressive training. We use the Geo-FNO (Li et al., 2023c), a variant of the FNO that supports irregularly sampled domains, for the irregularly sampled spatial domains experiments. The Geo-FNO also uses the curriculum training strategy for improved training.

**Factorized Fourier Neural Operator (F-FNO).** The Factorized Fourier Neural Operator (Tran et al., 2023) is an improved version of the Fourier Neural Operator that uses separable spectral convolution layers, improved residual connections, and pointwise MLPs. Separable spectral convolution layers factorize the Fourier transforms over the spatial dimension, which significantly decreases the number of model parameters. The improved residual connections, which are applied after the non-linearity, and the pointwise MLPs, akin to the pointwise MLPs in Transformers, further improve the performance. We use the original implementation of F-FNO and train the model in a similar fashion as FNO and Geo-FNO (i.e., autoregressive rollout with curriculum strategy).

**Operator Transformer (OFormer).** OFormer (Li et al., 2023a) leverages an attention-based encoder and decoder to decouple the input from the output spatial domain. The attention mechanism enables its direct application to irregularly sampled spatial domains. The model uses latent time-stepping to compute the dynamics of time-dependent PDEs. We follow the original implementation of OFormer and train it using a curriculum strategy that increases the trajectory length during training.

**Position-Induced Transformer (PIT).** PIT (Chen & Wu, 2024) uses an attention mechanism based on the Euclidean distances instead of the scaled dot-product. The model computes a latent representation that is based on a latent grid. The latent grid has to be fixed prior to the training and has the same dimension as the physical grid (e.g., 2D latent grid for 2D PDEs). For PDEs with regularly sampled domains, we follow the authors and use a uniform mesh with fewer points or resolution as latent grid which is a suitable choice. Similar to OFormer, PIT directly supports irregularly sampled domains. However, there are several suitable methods for generating the latent mesh for irregularly sampled domains (e.g., uniform mesh or sampling the latent mesh from the input mesh), and the authors did not experiment with irregularly sampled meshes. Thus, we generate the latent mesh for the airfoil dataset by randomly sampling from the training mesh and we use a uniform latent mesh for the cylinder dataset. Sampling a mesh for the cylinder dataset is not a good choice since there are huge differences in the meshes of different training samples and sampling from one mesh does not optimally cover the spatial domain. The authors propose to train the PIT model in an autoregressive fashion. We have conducted experiments training the model in an autoregressive fashion and noticed that the curriculum strategy, which we use to train the FNO and Geo-FNO, also significantly improved the performance of PIT. Thus, we train PIT with the same strategy to obtain lower errors.

**Latent Neural Operator (LNO).** LNO (Wang & Wang, 2024) compresses the input solution into a fixed latent representation with attention. The attention is solely computed using the input positions. A Transformer computes the next latent representation for the next timestep and a decoder maps the latent representation back to the physical space. Similar to OFormer and PIT, LNO directly supports irregularly sampled domains. The authors propose to train the model for one-step predictions (teacher forcing) and use it to do an autoregressive rollout. We follow their original setup and keep the training strategy.

**Attentive Reduced Order Model with Attention (AROMA).** AROMA (Serrano et al., 2024) is a reduced-order model that compresses the input solution into a fixed latent representation by applying cross-attention in a Perceiver-like fashion (Jaegle et al., 2021). A Transformer computes the latent representation for the next timestep with denoising diffusion, and a decoder maps the latent representation back to the physical space. Similar to LNO, AROMA directly supports irregularly sampled spatial domains. The authors use a two-stage training procedure that consists of an autoencoder training that trains the encoder-decoder model with a self-reconstruction loss and a dynamics training that trains the denoising diffusion model to predict the latent representation of the next timestep. We use their original implementation and keep the training strategy.

**Full Autoregressive Training vs Curriculum Strategy.** The errors of the autoregressive models (FNO, Geo-FNO, and PIT) can be reduced by using a curriculum strategy, that slowly increases the rollout length (Li et al., 2023a) and performs teacher-forcing training with the remaining timesteps (Takamoto et al., 2023), compared to fully autoregressive training. Table 6 shows that the curriculum strategy performs significantly better compared to fully autoregressive training. Thus, we train the autoregressive models with the curriculum strategy.

# J  Additional Details on the Experiments

## J.1  Hardware

We conduct all experiments on NVIDIA A100 SXM4 GPUs. The training of CALM-PDE takes 2 to 6 hours on an A100 GPU, depending on the dataset. In contrast, it takes up to 2 days to train the baselines on multiple A100 GPUs.

Table 6: Rel. L2 errors of FNO and PIT trained in a fully autoregressive fashion and with curriculum learning. The values in parentheses indicate the percentage deviation to the autoregressive model

| Model | Training Type | Relative L2 Error ($\downarrow$) 2D Navier-Stokes $\nu = 1e^{-4}$ |
|-------|---------------|---------------------------------------------------|
| FNO | Autoregressive | $0.1335^{\pm 0.0006}$ |
|     | Curriculum | $\mathbf{0.0811}^{\pm 0.0004}$ (-39%) |
| PIT | Autoregressive | $0.1038^{\pm 0.0052}$ |
|     | Curriculum | $\mathbf{0.0467}^{\pm 0.0068}$ (-55%) |

## J.2 Loss Function

We train the models with a relative L2 loss (Li et al., 2021b, 2023a,b; Chen & Wu, 2024), which offers the advantage of treating channels with small magnitudes on par with those having large magnitudes, ensuring balanced weighting, while also providing feedback when the PDE solution decays (i.e., the magnitudes decay) over time. Let $\boldsymbol{Y} \in \mathbb{R}^{N_t \times N \times N_c}$ be the ground truth and $\hat{\boldsymbol{Y}} \in \mathbb{R}^{N_t \times N \times c}$ the model's prediction. $N_t$ is the number of timesteps or trajectory length, $N$ the number of spatial points (e.g., $4096 = 64 \cdot 64$ for a resolution of $64 \times 64$ for 2D), and $N_c$ the number of channels. Then, the loss is defined as

$$L(\hat{\boldsymbol{Y}}, \boldsymbol{Y}) = \frac{1}{N_t \cdot N_c} \sum_{t}^{N_t} \sum_{c}^{N_c} \frac{\left\| \hat{\boldsymbol{Y}}_{t,\cdot,c} - \boldsymbol{Y}_{t,\cdot,c} \right\|}{\left\| \boldsymbol{Y}_{t,\cdot,c} \right\|}, \tag{23}$$

where $\|\cdot\|$ denotes the L2 norm.

## J.3 Evaluation Metric

We use the relative L2 error as defined previously as an evaluation metric. In addition to the advantages mentioned in the previous section, the relative L2 error can be interpreted as a percentage error.

## J.4 Hyperparameters

**Fourier Neural Operator (FNO) and Geo-FNO.** Table 7 shows the hyperparameters for FNO and Geo-FNO used in the experiments. We adopt comparable hyperparameters to those in Li et al. (2021b) for the 1D Burgers' equation and the 2D Navier-Stokes equations. For the 3D Navier-Stokes equations, we follow the hyperparameters outlined in Takamoto et al. (2022), with the exception of reducing the batch size from 5 to 2. For Geo-FNO, we utilize the hyperparameters proposed by Li et al. (2023c) which prove to be effective.

Table 7: Hyperparameters for FNO and Geo-FNO used in the experiments.

| Parameter | PDE | | | | |
|-----------|-----|-----|-----|-----|-----|
|           | 1D Burgers' | 2D Navier-Stokes | 3D Navier-Stokes | 2D Airfoil | 2D Cylinder |
| Width | 64 | 24 | 20 | 24 | 24 |
| Modes | 16 | 14 | 12 | 14 | 14 |
| Layers | 4 | 4 | 4 | 6 | 6 |
| Learning Rate | | | 1e-3 | | |
| Batch Size | 32 | 64 | 2 | 8 | 8 |
| Epochs | | | 500 | | |
| Parameters | 549,569 | 1,812,161 | 22,123,753 | 2,327,710 | 2,327,669 |

**Factorized Fourier Neural Operator (F-FNO).** Table 8 summarizes the hyperparameters used for F-FNO in our experiments. For the 1D Burgers' equation, we adopt hyperparameters similar to those in Li et al. (2021b), which prove to be effective. In the case of the 2D Navier-Stokes equations, we follow the configuration from Tran et al. (2023), but reduce the number of modes and the width to align with FNO, a necessary adjustment due to GPU memory limitations encountered during training.

The 3D Navier-Stokes configuration also mirrors FNO's hyperparameters. For the airfoil and cylinder datasets, we adopt the hyperparameters proposed by Li et al. (2023c).

Table 8: Hyperparameters for F-FNO used in the experiments.

| Parameter | PDE | | | | |
| --- | --- | --- | --- | --- | --- |
| | 1D Burgers' | 2D Navier-Stokes | 3D Navier-Stokes | 2D Airfoil | 2D Cylinder |
| Width | 64 | 24 | 20 | 24 | 24 |
| Modes | 16 | 14 | 12 | 14 | 14 |
| Layers | 4 | 4 | 4 | 6 | 6 |
| Learning Rate | | | 1e-3 | | |
| Batch Size | 32 | 64 | 2 | 8 | 8 |
| Epochs | | | 500 | | |
| Parameters | 665,281 | 151,361 | 131,913 | 625,270 | 625,229 |

**OFormer.** Table 9 shows the hyperparameters for OFormer used in the experiments. We employ similar hyperparameters for the 1D Burgers' equation and adopt the same parameters for the 2D Navier-Stokes equations as presented in Li et al. (2023a). For the 3D Navier-Stokes equations, we reduce the embedding dimensions to ensure the model fits on an A100 GPU. Increasing the model for 3D Navier-Stokes is infeasible due to excessive GPU memory usage and memory constraints in the experiments. For the 2D airfoil dataset, we utilize the hyperparameters specified in Li et al. (2023a) and adopt them for the 2D cylinder dataset.

Table 9: Hyperparameters for OFormer used in the experiments. [†]Increasing the model is infeasible due to memory constraints in the experiments.

| Parameter | PDE | | | | |
| --- | --- | --- | --- | --- | --- |
| | 1D Burgers' | 2D Navier-Stokes | 3D Navier-Stokes[†] | 2D Airfoil | 2D Cylinder |
| Encoder Embedding Dimension | 96 | 96 | 64 | 128 | 128 |
| Encoder Out Dimension | 96 | 192 | 128 | 128 | 128 |
| Encoder Layers | 4 | 5 | 5 | 4 | 4 |
| Decoder Embedding Dimension | 96 | 384 | 256 | 128 | 128 |
| Propagator Layers | 3 | 1 | 1 | 1 | 1 |
| Learning Rate | 8e-4 | 5e-4 | 5e-4 | 6e-4 | 6e-4 |
| Batch Size | 64 | 16 | 2 | 8 | 8 |
| Iterations | 128k | 128k | 22k | 48k | 48k |
| Parameters | 660,719 | 1,850,497 | 825,157 | 1,367,812 | 1,370,371 |

**Position-Induced Transformer (PIT).** Table 10 shows the hyperparameters for PIT used in the experiments. We adopt the hyperparameters introduced in Chen & Wu (2024) for the 1D Burgers' equation, with the modification of increasing the batch size from 8 to 64. For the 2D Navier-Stokes equations, we retain the same hyperparameters as specified in Chen & Wu (2024). To train the 3D Navier-Stokes model on an A100 GPU, we reduce the latent mesh, quantiles, and the number of heads. Increasing the model for 3D Navier-Stokes is impractical due to GPU memory limitations encountered during the experiments. For the 2D airfoil and 2D cylinder datasets, we apply the hyperparameters used for the 2D Navier-Stokes equations which prove to be effective. The latent mesh for the 2D airfoil dataset is sampled from the training mesh (cf. mesh prior initialization of CALM-PDE) while the model employs a uniform latent mesh for the 2D cylinder dataset.

**Latent Neural Operator (LNO).** Table 11 shows the hyperparameters for LNO used in the experiments. For the 1D Burgers' equation, the listed hyperparameters prove to be effective. Compared to the hyperparameters introduced in Wang & Wang (2024) for the 2D Navier-Stokes equations, we decrease the embedding dimension, number of modes, and number of projector layers to better match the other models in terms of the number of parameters and required compute. For 3D Navier-Stokes, the embedding dimension is reduced further to train the model on an A100 GPU. Increasing the model is infeasible due to memory constraints in the experiments. We adopt the hyperparameters from 2D Navier-Stokes for the 2D airfoil dataset which prove to be effective. For the 2D cylinder dataset, we use similar hyperparameters as for the 2D cylinder dataset but reduce the embedding dimension from 192 to 176.

Table 10: Hyperparameters for PIT used in the experiments. [†]Increasing the model is infeasible due to memory constraints in the experiments.

| Parameter | PDE | | | | |
|---|---|---|---|---|---|
| | 1D Burgers' | 2D Navier-Stokes | 3D Navier-Stokes[†] | 2D Airfoil | 2D Cylinder |
| Embedding Dimension | 64 | 256 | 256 | 256 | 256 |
| Heads | 2 | 2 | 1 | 2 | 2 |
| Layers | 4 | 4 | 4 | 4 | 4 |
| Latent Mesh | 1024 | $16 \times 16$ | $8 \times 8 \times 8$ | 256 | $16 \times 16$ |
| Encoder Quantile | 0.01 | 0.02 | 0.002 | 0.02 | 0.02 |
| Decoder Quantile | 0.08 | 0.02 | 0.01 | 0.02 | 0.02 |
| Learning Rate | 6e-4 | 1e-3 | 5e-4 | 6e-4 | 6e-4 |
| Batch Size | 64 | 20 | 1 | 16 | 16 |
| Epochs | 500 | 500 | 500 | 500 | 500 |
| Parameters | 78,861 | 1,249,805 | 923,659 | 1,251,088 | 1,252,383 |

Table 11: Hyperparameters for LNO used in the experiments. [†]Increasing the model is infeasible due to memory constraints in the experiments.

| Parameter | PDE | | | | |
|---|---|---|---|---|---|
| | 1D Burgers' | 2D Navier-Stokes | 3D Navier-Stokes[†] | 2D Airfoil | 2D Cylinder |
| Embedding Dimension | 96 | 192 | 128 | 192 | 176 |
| Modes | | | 128 | | |
| Projector Layers | 3 | 2 | 2 | 2 | 2 |
| Propagator Layers | 4 | 8 | 8 | 8 | 8 |
| Learning Rate | | | 1e-3 | | |
| Batch Size | 64 | 32 | 2 | 8 | 8 |
| Epochs | 500 | 250 | 250 | 500 | 500 |
| Parameters | 461,121 | 2,847,105 | 1,276,805 | 2,848,260 | 2,397,267 |

**AROMA.** Table 12 summarizes the hyperparameters used for AROMA in our experiments. We adopt the hyperparameters reported in Serrano et al. (2024), but reduce the number of epochs to finish each training within 24 hours.

Table 12: Hyperparameters for AROMA used in the experiments.

| | Parameter | PDE | | | | |
|---|---|---|---|---|---|---|
| | | 1D Burgers' | 2D Navier-Stokes | 3D Navier-Stokes | 2D Airfoil | 2D Cylinder |
| Encoder-Decoder | Hidden Dimension | | | 128 | | |
| | Number of Latents | 32 | 256 | 256 | 64 | 64 |
| | Latent Dimension | 8 | 16 | 16 | 16 | 16 |
| | Depth | | | 3 | | |
| | Encode Geometry | ✗ | ✗ | ✗ | ✓ | ✓ |
| | Learning Rate | | | 1e-3 | | |
| | Batch Size | 128 | 10 | 4 | 32 | 64 |
| | Epochs | 1500 | 1000 | 500 | 5000 | 10000 |
| Dynamics Model | Hidden Dimension | | | 128 | | |
| | Depth | | | 4 | | |
| | Minimum Noise | 1e-2 | 1e-3 | 1e-3 | 1e-6 | 1e-3 |
| | Denoising Steps | | | 3 | | |
| | Learning Rate | | | 1e-3 | | |
| | Batch Size | 512 | 64 | 4 | 32 | 128 |
| | Epochs | | | 1000 | | |
| | Parameters | 665,281 | 1,845,505 | 1,852,773 | 1,950,148 | 1,949,891 |

**CALM-PDE.** Table 13 shows the hyperparameters for CALM-PDE used in the experiments. The notation [16, 32, 64] for the channels means that the first layer has 16 channels, the 2nd 32 channels, and the 3rd 64 channels. Similarly, for different hyperparameters in the table. The dimensionality of the latent space is defined by the number of latent variables or query points and their channel dimension. We determine the dimensionality of the latent space based on the principle that the number of query points, receptive field, and channel dimension must cover the entire spatial domain. Similar to discrete CNNs, a small receptive field with a large stride (e.g., only a few query points) misses information, while a large receptive field with a small stride could have an averaging effect. With this principle, we obtain the number of query points (e.g., 1024, 256, 16 for the encoder layers),

which we use for all 2D experiments. Thus, a fixed number of query points works across different problems.

Table 13: Hyperparameters for CALM-PDE used in the experiments.

| | Parameter | 1D Burgers' | 2D Navier-Stokes | 3D Navier-Stokes | 2D Airfoil | 2D Cylinder |
|---|---|---|---|---|---|---|
| | | | | PDE | | |
| Encoder | Layers | | | 3 | | |
| | Channels | [16, 32, 64] | [32, 64, 128] | [64, 128, 256] | [32, 64, 128] | [32, 64, 128] |
| | Query Points | [256, 64, 8] | [1024, 256, 16] | [1024, 256, 64] | [1024, 256, 16] | [1024, 256, 16] |
| | Percentile | [0.05, 0.1, 0.5] | [0.01, 0.01, 0.2] | [0.001, 0.01, 0.2] | [0.01, 0.01, 0.5] | [0.01, 0.01, 0.5] |
| | Temperature | [1.0, 1.0, 1.0] | [1.0, 1.0, 0.1] | [1.0, 1.0, 0.1] | [1.0, 1.0, 0.1] | [1.0, 1.0, 0.1] |
| Decoder | Layers | | | 3 | | |
| | Channels | [32, 16, 1] | [64, 32, 1] | [128, 64, 5] | [64, 32, 4] | [64, 32, 3] |
| | Query Points | [64, 256, -] | [256, 1024, -] | [256, 1024, -] | [256, 1024, -] | [256, 1024, -] |
| | Percentile | [1.0, 0.5, 0.1] | [0.2, 0.01, 0.01] | [0.2, 0.01, 0.001] | [0.5, 0.01, 0.01] | [0.5, 0.01, 0.01] |
| | Temperature | [1.0, 1.0, 1.0] | [0.1, 1.0, 1.0] | [0.1, 1.0, 1.0] | [0.1, 1.0, 1.0] | [0.1, 1.0, 1.0] |
| | Periodic Boundary | ✓ | ✓ | ✓ | ✗ | ✗ |
| | Mesh Prior | ✗ | ✗ | ✗ | ✓ | ✗ |
| | Processor Layers | | | 2 | | |
| | Random Starting Points | ✓ | ✓ | ✓ | ✓ | ✗ |
| | Learning Rate | 1e-3 | 1e-3 | 1e-4 | 6e-4 | 1e-3 |
| | Batch Size | 64 | 32 | 4 | 16 | 8 |
| | Epochs | 500 | 500 | 250 | 500 | 500 |
| | Parameters | 568,331 | 2,230,627 | 8,367,267 | 2,237,240 | 2,239,403 |

# K   Hyperparameter Study

This section provides additional information about the hyperparameters of CALM-PDE and their robustness. In particular, we show the similarities of the hyperparameters of CALM-PDE and discrete CNNs, and demonstrate the robustness of the softmax temperature.

**Comparison to discrete CNNs.**   CALM-PDE introduces the softmax temperature, receptive field radius (percentile), channel dimension, and number of query points as hyperparameters. The receptive field radius (percentiles) is equivalent to the kernel size in discrete CNNs, and the number of query points to downsampling (e.g., stride and pooling) in CNNs. The channel dimension is equivalent to the channel dimension in CNNs. Alternatively, the number of query points can be compared to the number of latent query tokens used in other models, such as AROMA (Serrano et al., 2024) or LNO (Wang & Wang, 2024). Consequently, CALM-PDE maintains a conventional hyperparameter footprint without introducing excessive complexity.

**Hyperparameter Selection and Robustness.**   Selecting hyperparameters for CALM-PDE does not require an extensive hyperparameter search. As shown in Table 13, we use the same temperatures, receptive field sizes (percentiles), and number of query points for all 2D experiments, which demonstrates that the hyperparameters are robust and work across different problems.

**Hyperparameter Study of Softmax Temperature.**   The softmax temperature is a new hyperparameter that controls the distance weighting within the receptive field. We also use the same temperatures across all experiments, demonstrating its robustness. Additionally, we perform a hyperparameter study by sweeping the temperature from a high value in model (a) to a low value in model (f). As shown in Table 14, temperatures in the range of $T \in [0.1, 5.0]$ result in only minor changes in test error, confirming the robustness of the model to this parameter. All experiments are conducted on the 2D Navier-Stokes dataset with $\nu = 1e^{-4}$.

Table 14: Relative L2 errors of CALM-PDE models with different temperatures trained and tested on the 2D Navier-Stokes dataset.

| | Model Configuration | Relative L2 Error ($\downarrow$) |
|---|---|---|
| | Temperature T in Encoder and Decoder | 2D Navier-Stokes $\nu = 1e^{-4}$ |
| (a) | 10 | 0.0398 |
| (b) | 5 | 0.0336 |
| (c) | 2 | 0.0316 |
| (d) | 1 | 0.0304 |
| (e) | 0.1 | 0.0341 |
| (f) | 0.01 | 0.1640 |

## L    Model Analysis

**Visualization of Learned Positions.**    We visualize the learned positions of the query points of CALM-PDE. We conduct the experiment on models trained on the 2D Navier-Stokes equation, 2D Euler equation airfoil, and 2D Navier-Stokes cylinder datasets. Figure 9 shows the latent positions for a regular mesh where CALM-PDE samples more regularly. Figure 10 shows the input and output positions as well as learned query positions for the airfoil dataset. The results show that CALM-PDE samples more densely in important regions such as the boundary of the airfoil (see query points of decoder layer 2). A similar, but less emphasized effect can be observed for the encoder layer 1. Encoder layer 2 and decoder layer 1 do not show such an effect. Figure 11 shows the positions for the cylinder dataset. Similar to the airfoil dataset, CALM-PDE samples more densely in the regions where the cylinders are located. This effect can be observed in encoder layer 1 and decoder layer 2. Thus, the learned positions correspond to intuitively important regions and learnable query points help the model to have a higher information density in these important regions.

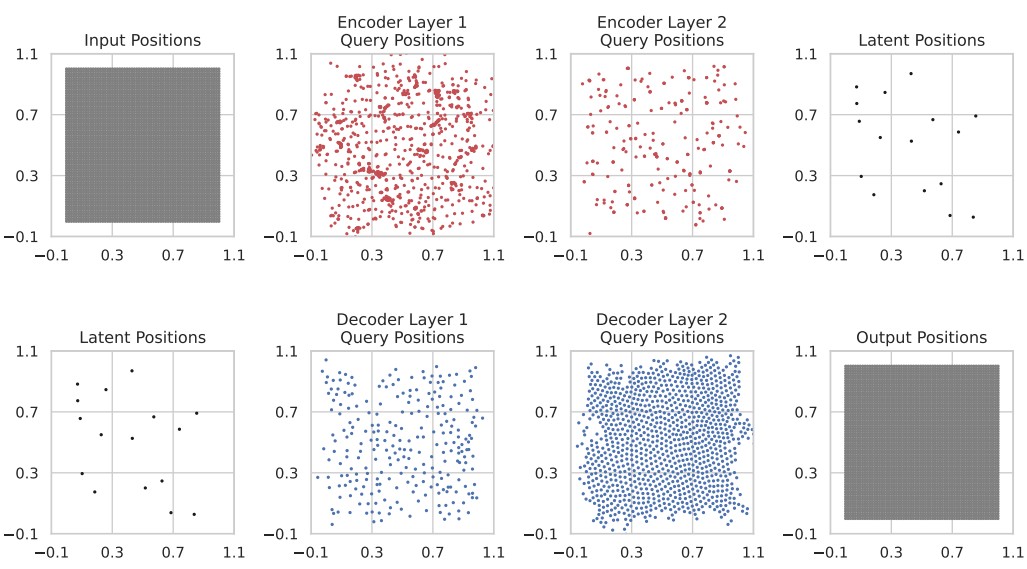

Figure 9: Input, output, and latent positions of CALM-PDE trained on the 2D Navier-Stokes dataset. CALM-PDE samples more regularly from the domain since the input and output positions are also regularly sampled.

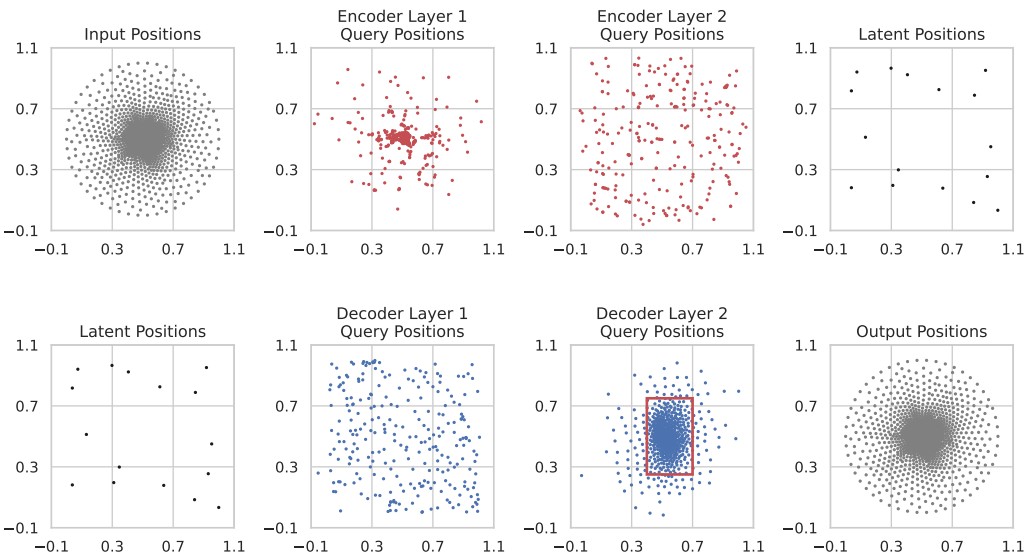

Figure 10: Input, output, and latent positions of CALM-PDE trained on the 2D Euler equation airfoil dataset. CALM-PDE samples more densely at the boundary of the airfoil (indicated by the red rectangle) by moving more query points to these areas.

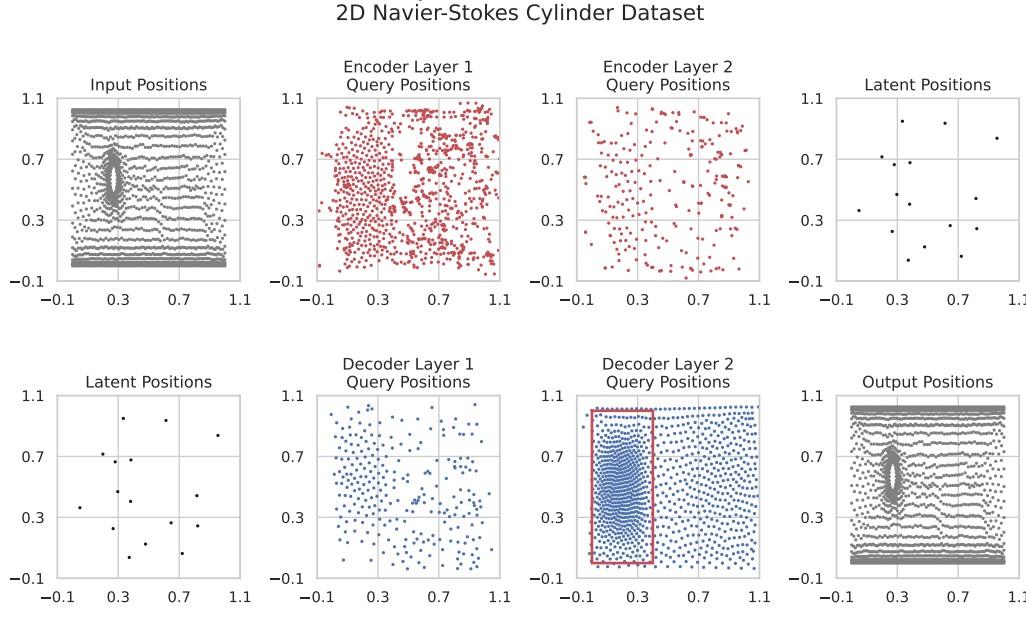

Figure 11: Input, output, and latent positions of CALM-PDE trained on the 2D incompressible Navier-Stokes cylinder dataset. CALM-PDE samples more densely in regions where the cylinders are located (indicated by the red rectangle) by moving more query points to these areas.

**Information Content of Query Points.**   Next, we investigate the information content of the query points. We are mainly interested in whether query points represent local or global information. Local

information means that each query point represents the solution of the nearby surroundings while global information means that a single query point influences the global spatial domain. Due to the constrained receptive field, we assume that each token only represents local information. We add Gaussian noise to a token and investigate how the solution is affected to validate the hypothesis. Figure 12 shows that only the surrounding of the token is affected which matches our hypothesis that each token represents local information.

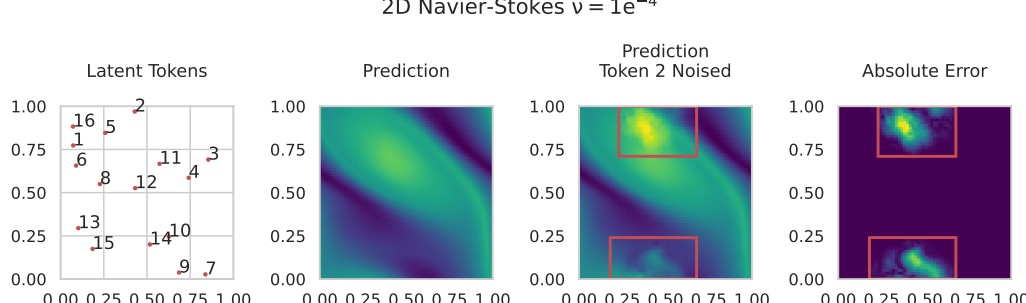

Figure 12: The latent tokens represent local information since adding Gaussian noise to a token (here: Token 2) changes only the solution of the token's surroundings (here: indicated by a red rectangle).

**Visualization of Kernel Weights.** Additionally, we visualize the learned kernel weights. Figure 13 shows the absolute values and Figure 14 the weights with the sign for the first encoder layer of CALM-PDE trained on the 2D Navier-Stokes dataset. The filters have similarities to the filters learned by a discrete CNN and some filters are similar to an edge detection filter (e.g., filter 4 in Figure 14). Figure 15 and Figure 16 show the learned filters for the second encoder layer of CALM-PDE. We only visualize the 64 filters for the first input channel. The filters are less regular and have similarities to attention weights in Transformers. It is important to note that filters complement each other by having one filter that focuses on a subset of tokens and another filter that focuses on the remaining subset of tokens (e.g., filters 31 and 38 in Figure 15).

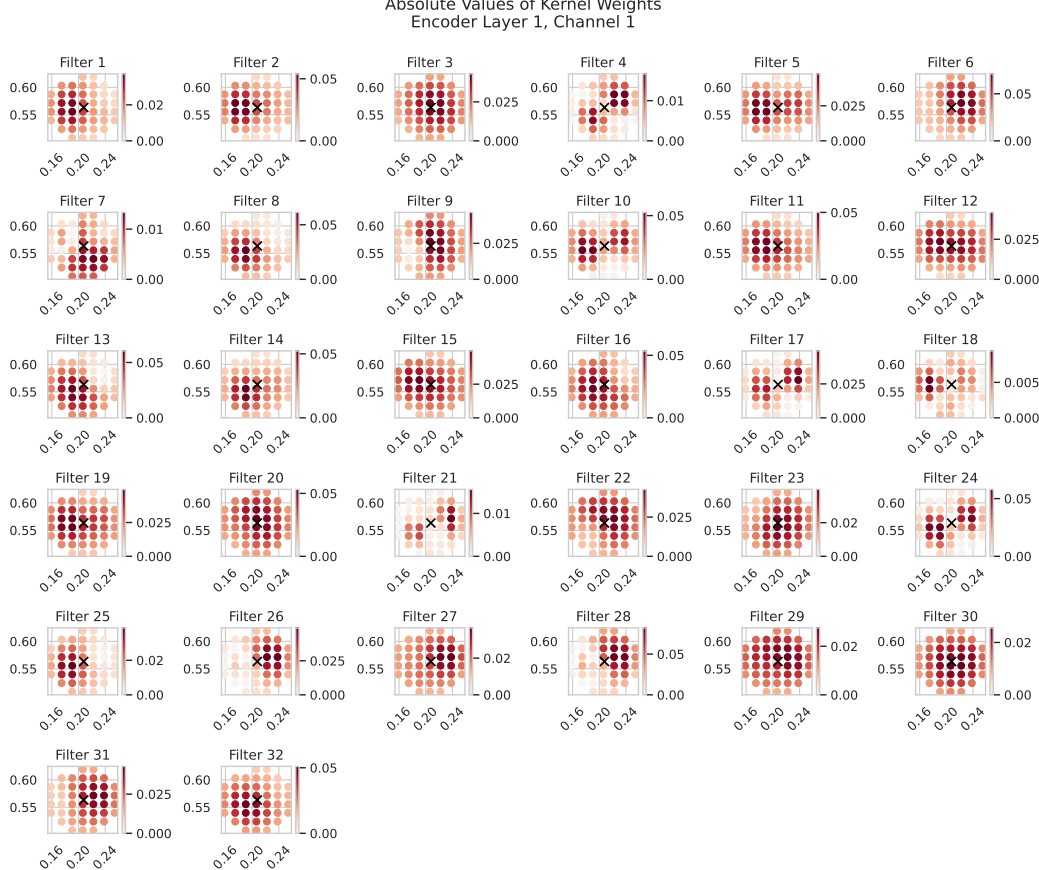

Figure 13: Absolute values of the kernel weights for the first layer of CALM-PDE's encoder trained on 2D Navier-Stokes. The black cross (×) represents the query point.

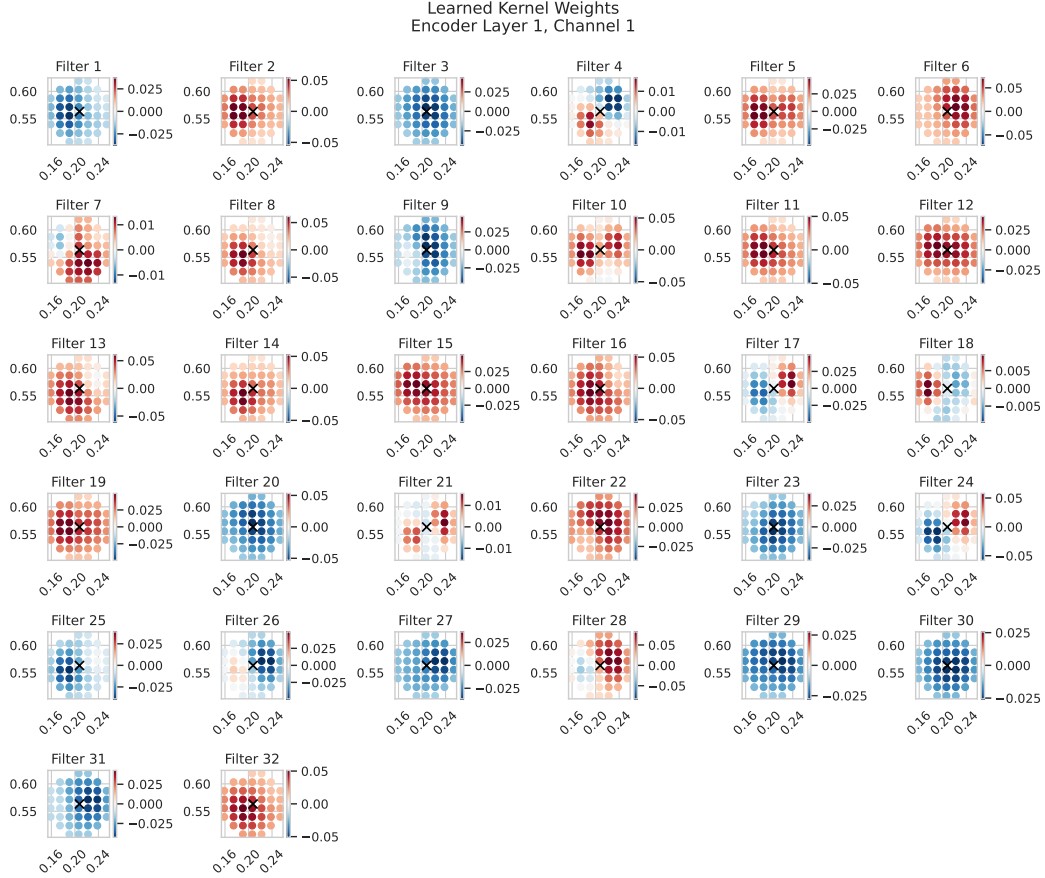

Figure 14: Kernel weights for the first layer of CALM-PDE's encoder trained on 2D Navier-Stokes. The black cross (×) represents the query point.

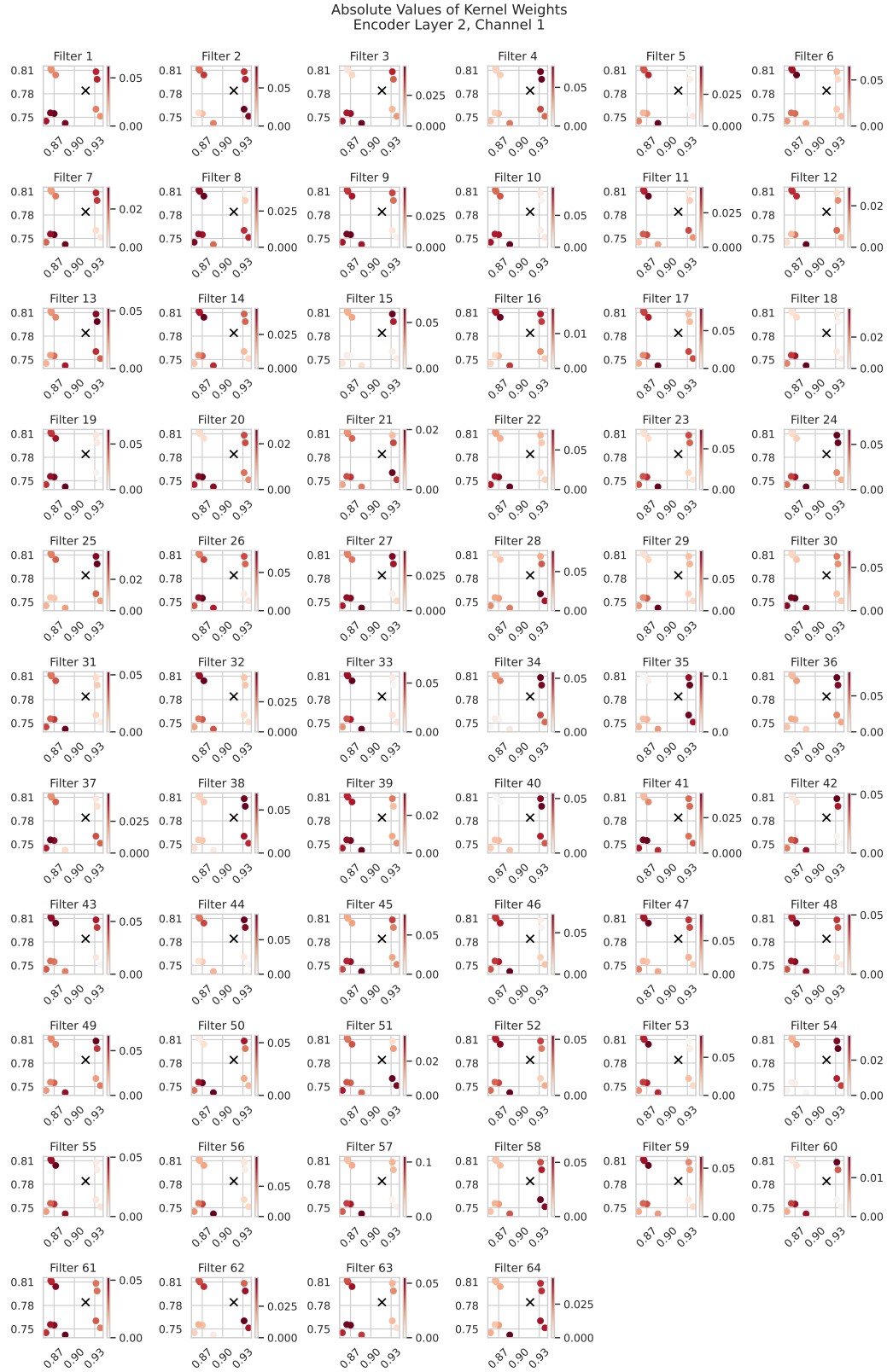

Figure 15: Absolute values of the kernel weights for the second layer of CALM-PDE's encoder trained on 2D Navier-Stokes. The black cross (×) represents the query point.

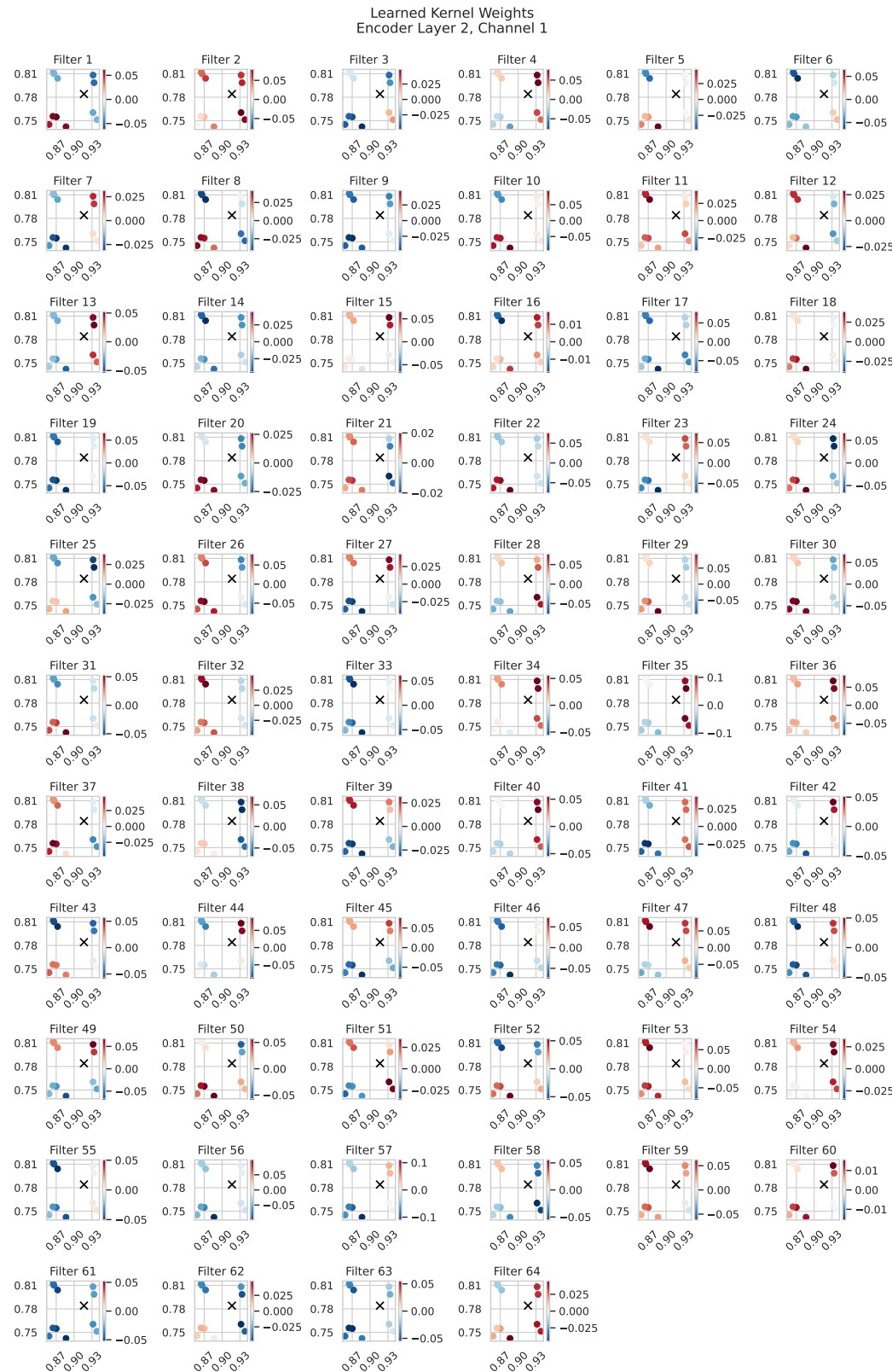

Figure 16: Kernel weights for the second layer of CALM-PDE's encoder trained on 2D Navier-Stokes. The black cross (×) represents the query point.

**Size of Latent Space.** We compare the sizes required for one timestep of a PDE in different representations. We define the representation size as the number of spatial points or tokens times the number of channels per token (e.g., $64 \cdot 64 \cdot 64 \cdot 5 \approx 1,3M$ for a 3D PDE with a spatial resolution of $64 \times 64 \times 64$ and 5 channels). A smaller representation size or latent space can result in reduced model complexity (memory consumption and or inference time). However, if a model has a smaller latent space but the computations are computationally expensive, it might be possible that it is less efficient compared to a model with a larger latent space but simpler computations. Furthermore, a small latent space (i.e., dimensionality reduction) can eliminate redundant features and noise. Our observations show that the baselines often have a larger latent space than the physical space while CALM-PDE consistently performs a compression and has a smaller latent space compared to the representation in the physical space.

Table 15: Representation sizes for one timestep of different PDEs in the physical and model's latent spaces. $^{\dagger}$A larger latent space is infeasible for these models due to memory constraints in the experiments.

| Model | Representation Size ($\downarrow$) | | | | |
|---|---|---|---|---|---|
| | 1D Burgers' | 2D Navier-Stokes | 3D Navier-Stokes | 2D Airfoil | 2D Cylinder |
| Physical Space | 1024 | 4096 | 1.3M | 20,932 | 5655 (avg) |
| FNO | 2048 | 18,816 | 276k | 32,256 | 32,256 |
| OFormer | 98,304 | 786k | 67M$^{\dagger}$ | 670k | 241k (avg) |
| PIT | 65,536 | 65,536 | 131k$^{\dagger}$ | 65,536 | 65,536 |
| LNO | 12,288 | 24,576 | **16,384**$^{\dagger}$ | 24,576 | 22,528 |
| CALM-PDE | **512** | **2048** | **16,384** | **2048** | **2048** |

# M  Ablation Study

We conduct an ablation study to investigate the effect of (i) learnable query positions, (ii) kernel modulation, and (iii) distance weighting. Furthermore, we replace the proposed kernel function with a function that outputs weights only based on the Euclidean distance. We also investigate the influence of different latent dimensions on the model's performance. We test the modified models on the 2D Euler equation airfoil dataset and the 2D Navier-Stokes dataset.

## M.1  Learnable Query Positions

In the first experiment, we investigate the effect of learnable query positions. We conduct the experiment on the 2D Euler equation airfoil dataset with an irregularly sampled spatial domain. We examine three different configurations of CALM-PDE with (a) learnable query points and mesh prior initialization enabled (i.e., we initially sample the query points from the mesh and allow the model to change their positions), (b) no learnable query points but with mesh prior (i.e., we sample from the mesh to obtain the query points and their positions are fix), and (c) no learnable query points and no mesh prior (i.e., the query points are randomly sampled from the entire spatial domain and fixed). Table 16 shows that the model with learnable query points and mesh prior achieves the lowest errors. Soley sampling from the mesh to obtain the query points (mesh prior and no learnable query points) yields a significantly higher error. Using random query points with fixed positions results in the highest errors which is intuitive since the model does not necessarily have query points at important regions and cannot move query points to these regions. Thus, it is forced to learn the underlying mesh only with the kernel function.

## M.2  Kernel Modulation

In this experiment, we disable the query modulation. We conduct the experiment on the 2D Navier-Stokes dataset with a regularly sampled spatial domain and the 2D Euler airfoil dataset with an irregularly sampled spatial domain. We compare the CALM-PDE model with (a) enabled kernel modulation against a model (b) without kernel modulation. Table 17 shows the relative L2 errors for

Table 16: Relative L2 errors of CALM-PDE models trained and tested on the 2D Euler equation airfoil dataset. The values in parentheses indicate the percentage deviation to CALM-PDE with enabled learnable query points and mesh prior. Underlined values indicate the second-best errors.

| | Model Configuration | | Relative L2 Error ($\downarrow$) |
|---|---|---|---|
| | Learnable Query Points | Mesh Prior | 2D Euler Airfoil |
| (a) | ✓ | ✓ | $\mathbf{0.0515}^{\pm 0.0007}$ |
| (b) | ✗ | ✓ | $\underline{0.0709}^{\pm 0.0016}$ (+38%) |
| (c) | ✗ | ✗ | $0.1627^{\pm 0.0046}$ (+216%) |

Table 17: Relative L2 errors of CALM-PDE models trained and tested on the 2D Navier-Stokes and 2D Euler airfoil datasets. The value in parentheses indicates the percentage deviation to CALM-PDE with enabled kernel modulation.

| | Model Configuration | Relative L2 Error ($\downarrow$) | |
|---|---|---|---|
| | Kernel Modulation | 2D Navier-Stokes $\nu = 1e^{-4}$ | 2D Euler Airfoil |
| (a) | ✓ | $\mathbf{0.0301}^{\pm 0.0014}$ | $\mathbf{0.0515}^{\pm 0.0007}$ |
| (b) | ✗ | $0.0358^{\pm 0.0014}$ (+19%) | $0.0632^{\pm 0.0017}$ (+23%) |

both model configurations. The proposed kernel modulation, which allows each query point to have a different kernel function, results in a lower error than the models without the kernel modulation.

## M.3 Distance Weighting

Furthermore, we investigate the effect of the distance weighting term, which includes the Euclidean distance directly into the kernel function to give more weight to closer points. The mechanism ensures that the model puts more emphasis on closer points within the epsilon neighborhood, provides feedback that points on the boundary of the epsilon neighborhood are less important to avoid hard cuts caused by the displacement of query points, and serves as a normalization factor for the Monte Carlo integration. We compare three models with (a) distance weighting, (b) without distance weighting but with a normalization term $1/M = 1/|\mathtt{RF}(a)|$, and (c) without distance weighting and without a normalization term. The kernel function of the model without distance weighting but with a normalization term is given as

$$k_{i,o}(a - \alpha) = \mathtt{MLP}\big(\mathtt{RFF}(a - \alpha)\big)_{i,o} \frac{1}{|\mathtt{RF}(a)|} \tag{24}$$

where $M = |\mathtt{RF}(a)|$ denotes the number of input points in the receptive field $\mathtt{RF}(a)$ of query point $a$. The kernel function of the model without distance weighting and without a normalization term is given as

$$k_{i,o}(a - \alpha) = \mathtt{MLP}\big(\mathtt{RFF}(a - \alpha)\big)_{i,o} \tag{25}$$

where $\mathtt{MLP}$ denotes the modulated two-layer multi-layer perceptron and $\mathtt{RFF}$ random Fourier features. Table 18 shows that the models without distance weighting and without a normalization term diverge during training. Including a normalization term of $1/M$ stabilizes the training but still yields a higher error than the model with distance weighting. Thus, distance weighting, which emphasizes input points closer to the query point and serves as a normalization, is crucial for reducing the error.

## M.4 Distance-based Kernel Function

We replace the proposed kernel function, which computes the weights based on translation vectors and the Euclidean distances, with a kernel function that computes the weights solely based on the Euclidean distances. The temperature in the $\mathtt{softmax}$ function is a learnable parameter, and each

Table 18: Relative L2 errors of CALM-PDE models trained and tested on the 2D Navier-Stokes dataset. [†]The value is unavailable since the models diverge during training.

| | Model Configuration | | Relative L2 Error ($\downarrow$) |
|---|---|---|---|
| | Distance Weighting | Normalization | 2D Navier-Stokes $\nu = 1e^{-4}$ |
| (a) | ✓ | `softmax` | $\mathbf{0.0301}^{\pm 0.0014}$ |
| (b) | ✗ | $1/M$ | $0.0590^{\pm 0.0122}$ (+96%) |
| (c) | ✗ | ✗ | N/A[†] |

Table 19: Relative L2 errors of CALM-PDE models trained and tested on the 2D Navier-Stokes dataset. The value in parentheses indicates the percentage deviation to CALM-PDE with the proposed kernel function.

| | Model Configuration | Relative L2 Error ($\downarrow$) |
|---|---|---|
| | Kernel Function | 2D Navier-Stokes $\nu = 1e^{-4}$ |
| (a) | CALM-PDE | $\mathbf{0.0301}^{\pm 0.0014}$ |
| (b) | Distance-based | $0.1783^{\pm 0.0062}$ (+492%) |

filter has its own temperature parameter. The kernel function is given as

$$k_{i,o}(a - \alpha) = \frac{\exp\left(-\frac{\|a-\alpha\|^2 - \min(a)}{T_{i,o}(\max(a) - \min(a))}\right)}{\sum\limits_{\alpha_j \in \text{RF}(a)} \exp\left(-\frac{\|a-\alpha_j\|^2 - \min(a)}{T_{i,o}(\max(a) - \min(a))}\right)}, \quad \min(a) = \min_{\alpha_j \in \text{RF}(a)} \|a - \alpha_j\|^2 \quad (26)$$

where $T_{i,o}$ represents a learnable temperature parameter and $\text{RF}(\alpha)$ the receptive field for the query point $\alpha$. We compare the CALM-PDE model with (a) the proposed CALM-PDE kernel function against a model with (b) the distance-based kernel function. Table 19 shows that the kernel function proposed by CALM-PDE achieves a lower error than a distance-based kernel function.

## M.5   Latent Dimension

We investigate the effect of different latent dimensions by increasing the number of query points. Starting from model (a), which uses a small number of query points, we incrementally scale up to model (e), characterized by a large number of query points. As shown in Table 20, increasing the latent dimension initially leads to a reduction in test error. However, beyond a certain threshold, further increases of the latent dimension result in an increasing test error, suggesting overfitting or diminishing returns. The table also reports training time, which consistently increases with the number of query points, reflecting the computational cost of a higher latent dimension. The notation [256, 64, 4] denotes 256 query points for the first layer, 64 query points in the 2nd layer, and 4 for the last layer.

Table 20: Relative L2 errors and training times of CALM-PDE models with different numbers of query points trained and tested on the 2D Navier-Stokes dataset.

| | Model Configuration | | Relative L2 Error ($\downarrow$) | Training time ($\downarrow$) |
|---|---|---|---|---|
| | Number of Query Points | | 2D Navier-Stokes | |
| | Encoder | Decoder | $\nu = 1e^{-4}$ | |
| (a) | [256, 64, 4] | [64, 256, -] | 0.0533 | 4:18h |
| (b) | [512, 128, 8] | [128, 512, -] | 0.0425 | 4:40h |
| (c) | [1024, 256, 16] | [256, 1024, -] | 0.0304 | 5:25h |
| (d) | [2048, 512, 32] | [512, 2048, -] | 0.0251 | 6:21h |
| (e) | [4096, 1024, 64] | [1024, 4096, -] | 0.0267 | 8:19h |

# N   Additional Results

## N.1   Quantitative Results

This section provides the full benchmark results with the standard deviations of multiple runs, which we omit for a more compact representation in the tables in the main paper. Table 21 shows the errors for the experiments with regular meshes and Table 22 for the irregular meshes.

Table 21: Relative L2 errors of models trained and tested on regular meshes. The values in parentheses indicate the percentage deviation to CALM-PDE and underlined values indicate the second-best errors.

| Model | Relative L2 Error ($\downarrow$) | | | |
|---|---|---|---|---|
| | 1D Burgers' $\nu = 1e^{-3}$ | 2D Navier-Stokes $\nu = 1e^{-4}$ | 2D Navier-Stokes $\nu = 1e^{-5}$ | 3D Navier-Stokes $\eta = \zeta = 1e^{-8}$ |
| FNO | $0.0358^{\pm 0.0006}$ (+46%) | $0.0811^{\pm 0.0004}$ (+169%) | $0.0912^{\pm 0.0003}$ (-12%) | $0.6898^{\pm 0.0002}$ (+2%) |
| F-FNO | $0.0362^{\pm 0.0001}$ (+47%) | $0.0863^{\pm 0.0012}$ (+187%) | $\underline{0.0844}^{\pm 0.0008}$ (-18%) | $\mathbf{0.6466}^{\pm 0.0044}$ (-4%) |
| OFormer | $0.0575^{\pm 0.0015}$ (+134%) | $\underline{0.0380}^{\pm 0.0009}$ (+26%) | $0.1938^{\pm 0.0106}$ (+88%) | $\underline{0.6719}^{\pm 0.0105}$ (-1%) |
| PIT | $0.1209^{\pm 0.0048}$ (+391%) | $0.0467^{\pm 0.0068}$ (+55%) | $0.1633^{\pm 0.0029}$ (+58%) | $0.7423^{\pm 0.0036}$ (+10%) |
| LNO | $\underline{0.0309}^{\pm 0.0012}$ (+26%) | $0.0384^{\pm 0.0020}$ (+28%) | $\mathbf{0.0789}^{\pm 0.0051}$ (-24%) | $0.7063^{\pm 0.0027}$ (+4%) |
| AROMA | $0.0937^{\pm 0.0143}$ (+281%) | $0.1061^{\pm 0.0455}$ (+252%) | $0.1931^{\pm 0.0161}$ (+87%) | $1.3328^{\pm 0.2210}$ (+97%) |
| CALM-PDE | $\mathbf{0.0246}^{\pm 0.0013}$ | $\mathbf{0.0301}^{\pm 0.0014}$ | $0.1033^{\pm 0.0057}$ | $0.6761^{\pm 0.0020}$ |

Table 22: Relative L2 errors of models trained and tested on the irregular meshes with geometries in the fluid flow. The values in parentheses indicate the percentage deviation to CALM-PDE and underlined values indicate the second-best errors.

| Model | Relative L2 Error ($\downarrow$) | |
|---|---|---|
| | 2D Euler Airfoil | 2D Navier-Stokes Cylinder |
| Geo-FNO | $\underline{0.0388}^{\pm 0.0019}$ (-25%) | $0.1383^{\pm 0.0018}$ (+17%) |
| F-FNO | $0.1081^{\pm 0.0195}$ (+110%) | $0.1490^{\pm 0.0040}$ (+26%) |
| OFormer | $0.0520^{\pm 0.0003}$ (+1%) | $0.2264^{\pm 0.0109}$ (+91%) |
| PIT | $0.0894^{\pm 0.0078}$ (+74%) | $0.1400^{\pm 0.0043}$ (+18%) |
| LNO | $0.0582^{\pm 0.0026}$ (+13%) | $0.1654^{\pm 0.0409}$ (+39%) |
| AROMA | $\mathbf{0.0372}^{\pm 0.0004}$ (-28%) | $\mathbf{0.1139}^{\pm 0.0027}$ (-4%) |
| CALM-PDE | $0.0515^{\pm 0.0007}$ | $\underline{0.1186}^{\pm 0.0020}$ |

## N.2   Qualitative Results

We visualize the predictions and ground truth of randomly selected trajectories for the benchmark problems.

- 1D Burgers' Equation: Figure 17

- 2D Navier-Stokes with Parameters $\nu = 1e^{-4}$ and $\nu = 1e^{-5}$: Figure 18
- 2D Euler Equation with Airfoil Geometry: Figure 19, Figure 20, Figure 22, Figure 21
- 2D Incompressible Navier-Stokes with Cylinder Geometries: Figure 23, Figure 24, Figure 25

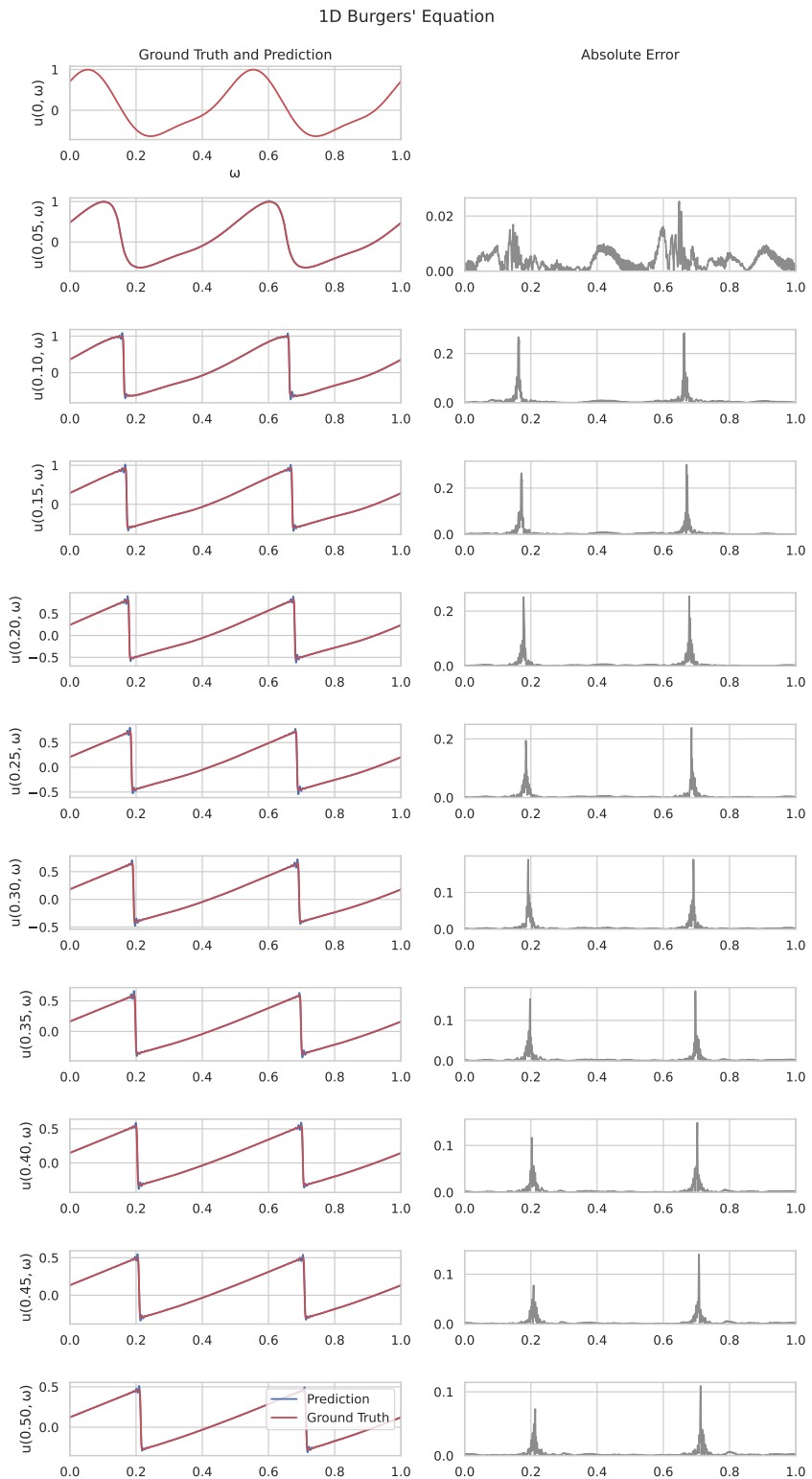

Figure 17: Ground truth, prediction, and absolute error for the first 11 timesteps of a randomly selected 1D Burgers' equation trajectory.

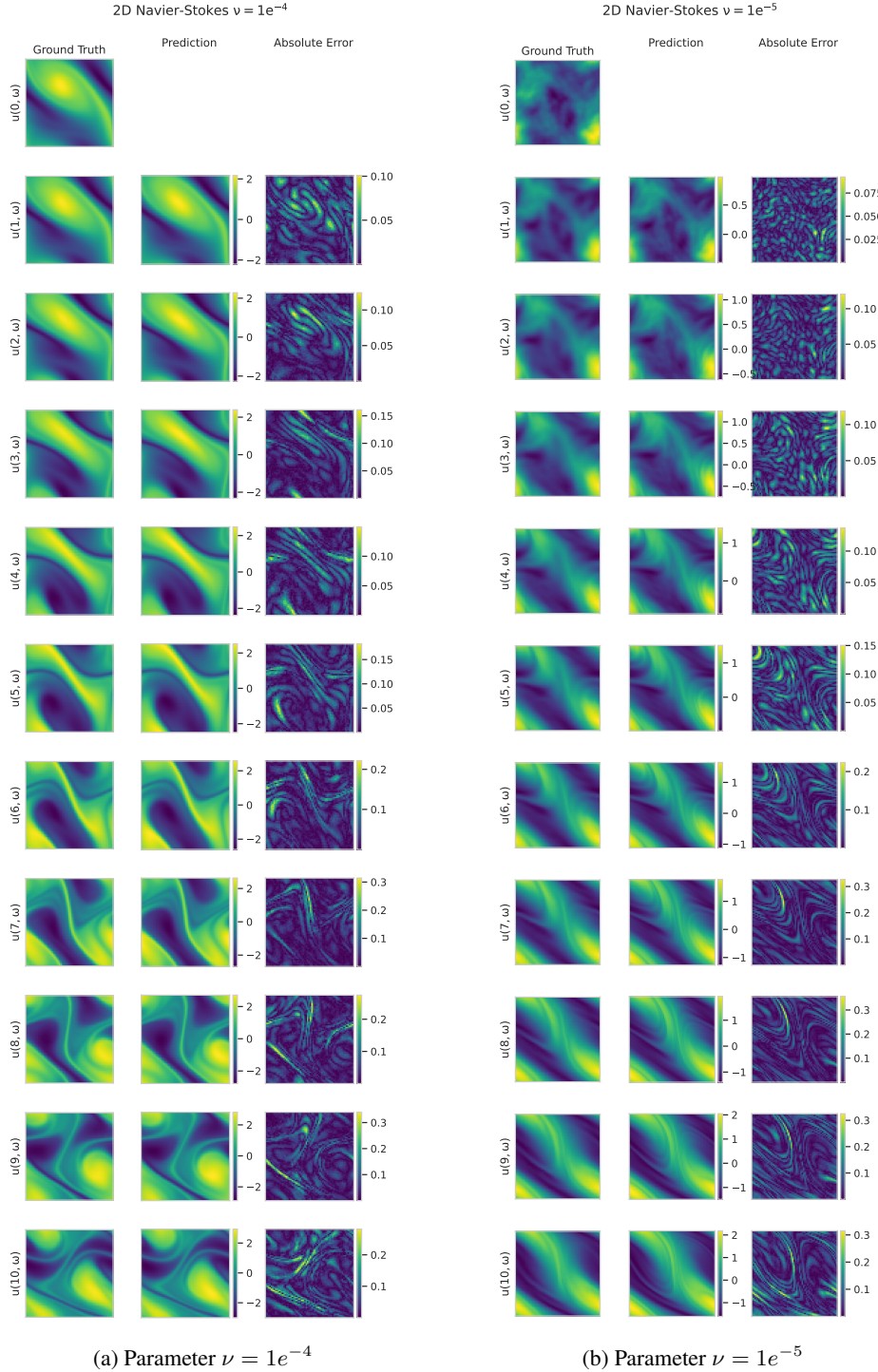

(a) Parameter $\nu = 1e^{-4}$        (b) Parameter $\nu = 1e^{-5}$

Figure 18: Ground truth, prediction, and absolute error for the first 11 timesteps of two randomly selected 2D Navier-Stokes $\nu = 1e^{-4}$ and $\nu = 1e^{-5}$ trajectories.

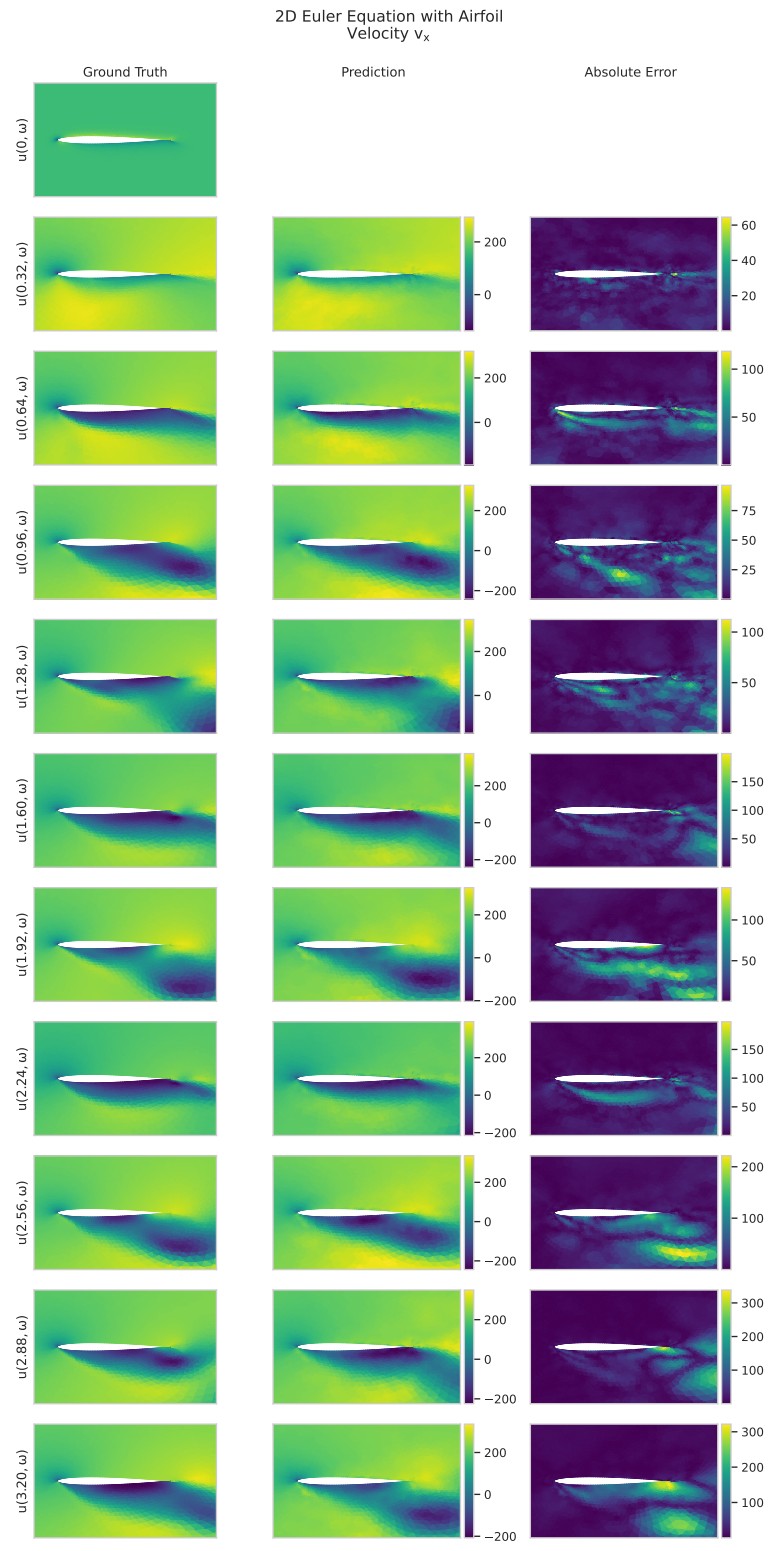

Figure 19: Ground truth, prediction, and absolute error of the velocity $v_x$ for the first 11 timesteps of a randomly selected 2D Euler equation with airfoil geometry trajectory.

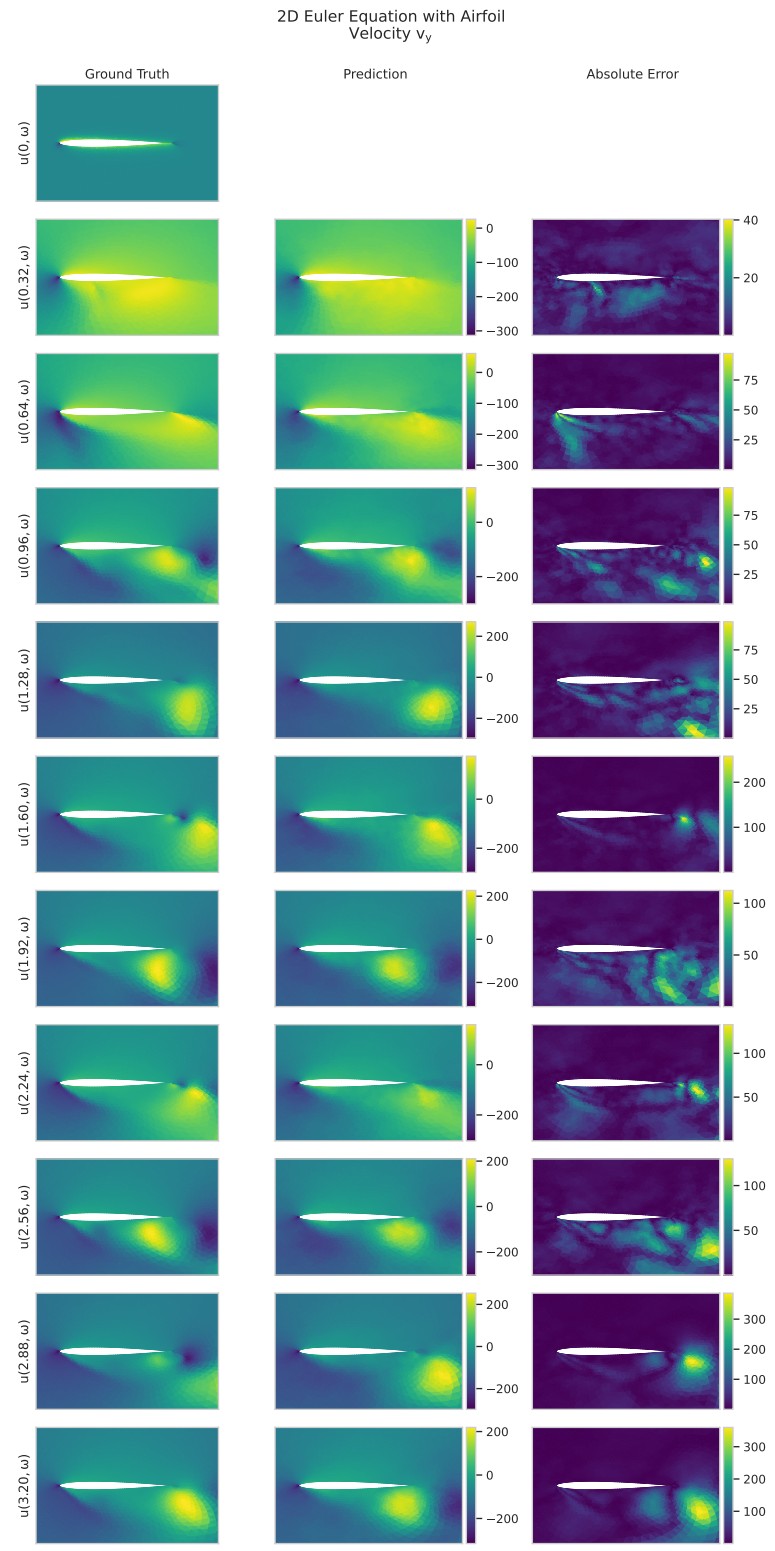

Figure 20: Ground truth, prediction, and absolute error of the velocity $v_y$ for the first 11 timesteps of a randomly selected 2D Euler equation with airfoil geometry trajectory.

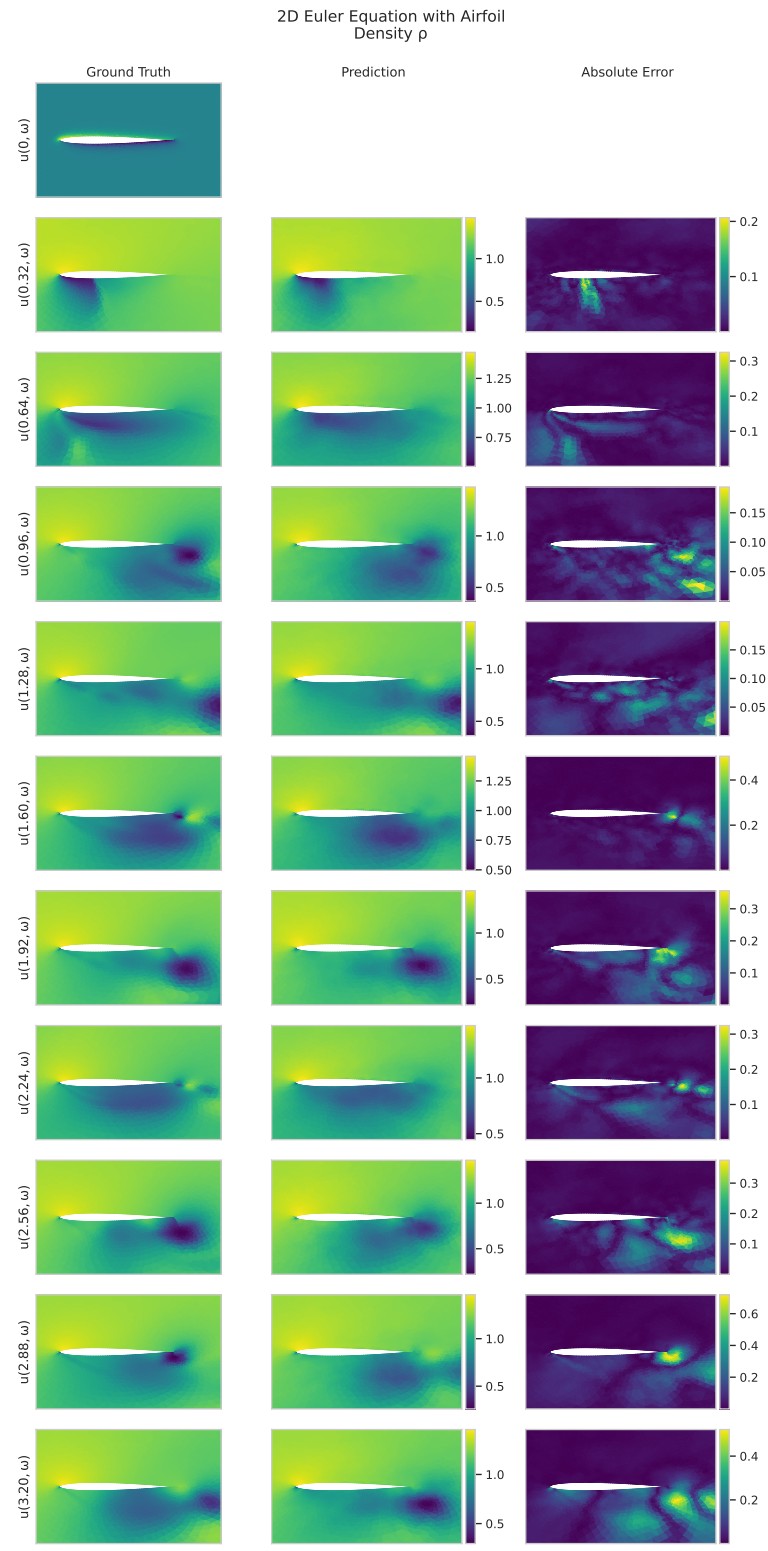

Figure 21: Ground truth, prediction, and absolute error of the density $\rho$ for the first 11 timesteps of a randomly selected 2D Euler equation with airfoil geometry trajectory.

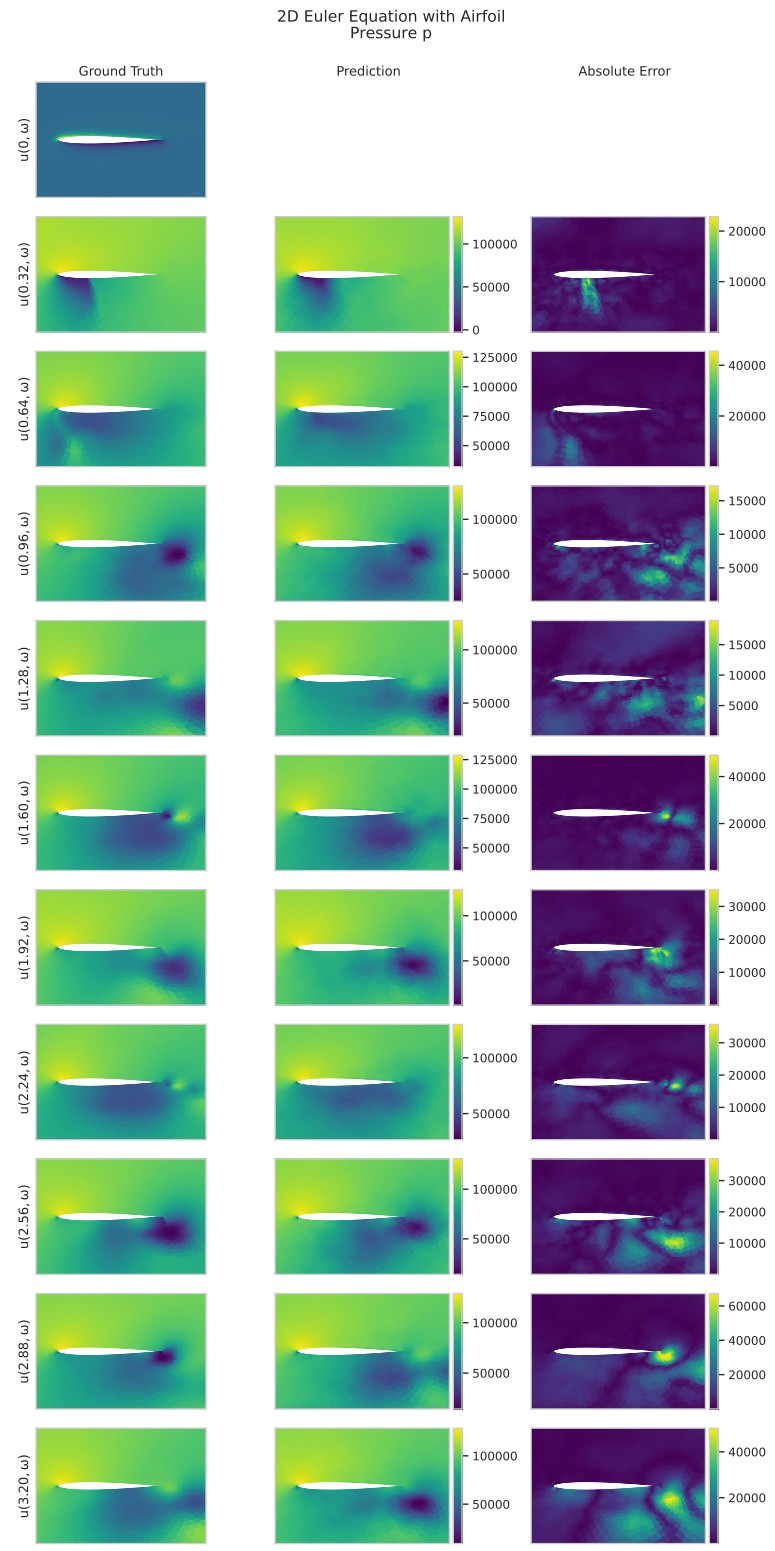

Figure 22: Ground truth, prediction, and absolute error of the pressure $p$ for the first 11 timesteps of a randomly selected 2D Euler equation with airfoil geometry trajectory.

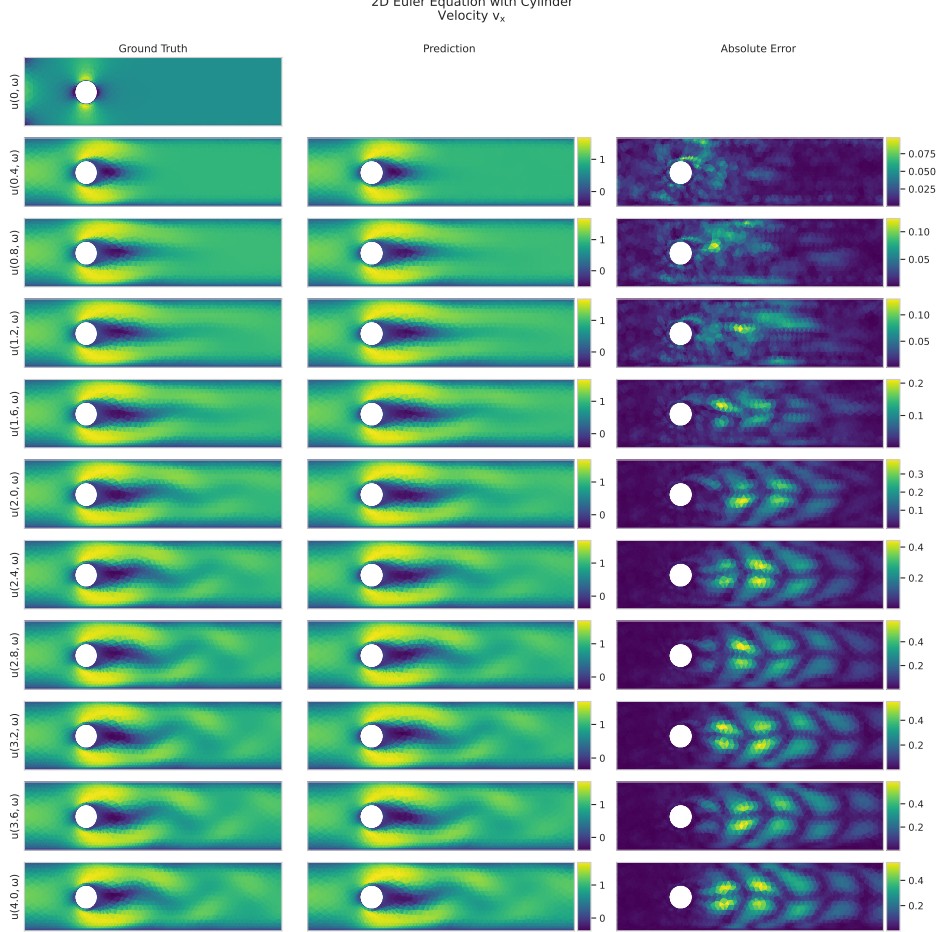

Figure 23: Ground truth, prediction, and absolute error of the velocity $v_x$ for the first 11 timesteps of a randomly selected 2D incompressible Navier-Stokes with cylinder geometries trajectory.

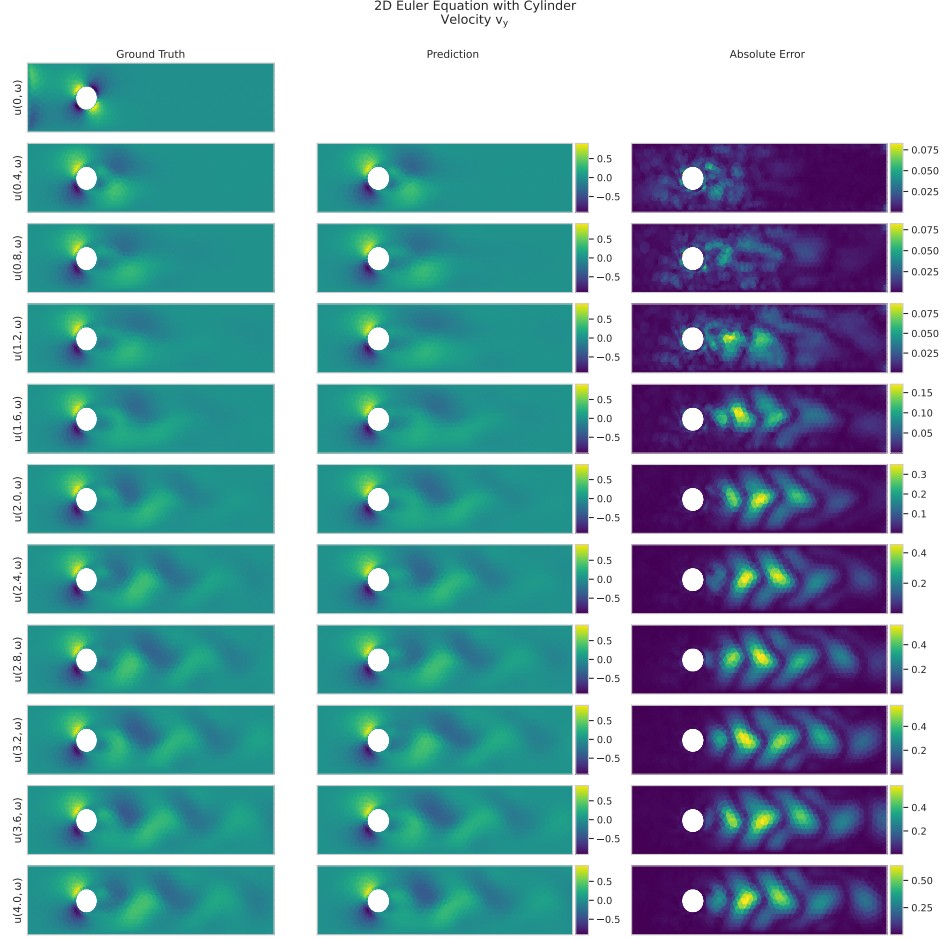

Figure 24: Ground truth, prediction, and absolute error of the velocity $v_y$ for the first 11 timesteps of a randomly selected 2D incompressible Navier-Stokes with cylinder geometries trajectory.

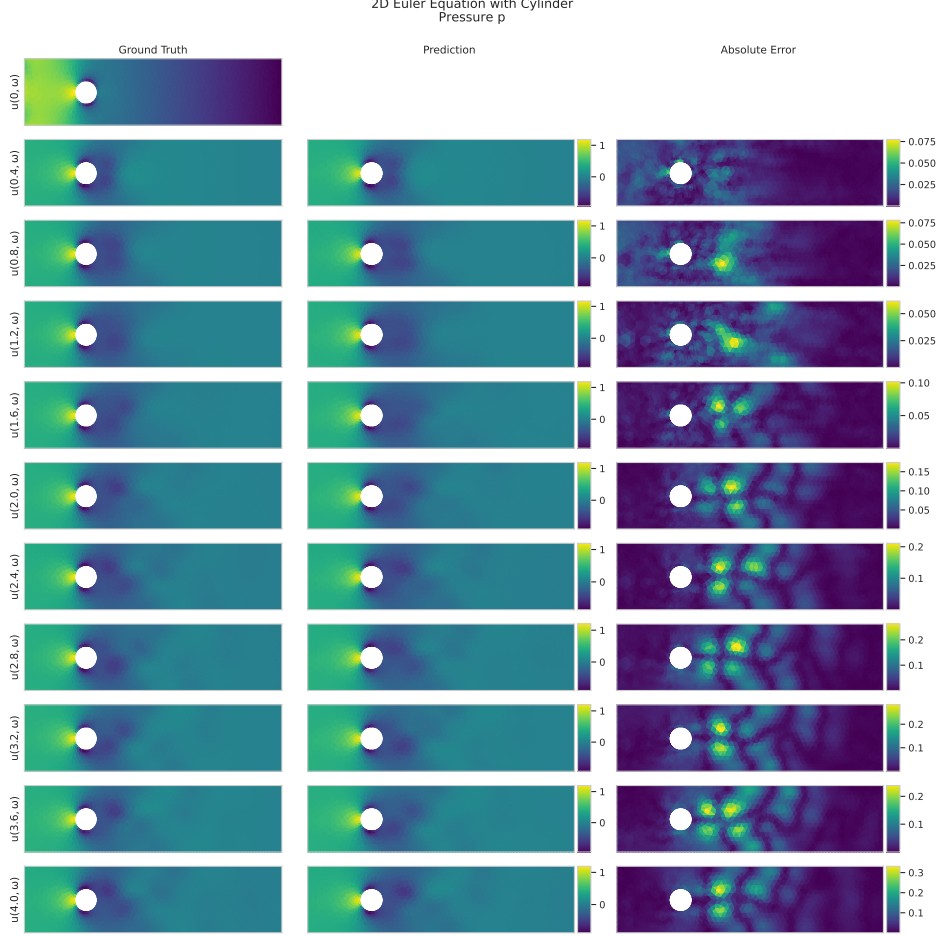

Figure 25: Ground truth, prediction, and absolute error of the pressure $p$ for the first 11 timesteps of a randomly selected 2D incompressible Navier-Stokes with cylinder geometries trajectory.

# O   Additional Experiments

Beyond the main experiments, we explore time-independent PDEs with complex and varying geometries. Additionally, we evaluate an encoder–decoder-only variant of CALM-PDE trained with a self-reconstruction loss to test its encoding and decoding capabilities. Finally, we analyze the scalability of the model with respect to the output query mesh size.

## O.1   Time-independent Problems

We conduct additional experiments on time-independent PDE problems. In particular, we experiment on the time-independent airfoil and elasticity datasets from Geo-FNO (Li et al., 2023c). The task is to map an input geometry (e.g., changing airfoil shape) to a physical quantity. We follow the setup from LNO (Wang & Wang, 2024) and take the errors for the baselines from their work. The baselines include Geo-FNO (Li et al., 2023c), F-FNO (Tran et al., 2023), Galerkin Transformer (Cao, 2021), OFormer (Li et al., 2023a), GNOT (Hao et al., 2023), ONO (Xiao et al., 2024), Transolver (Wu et al., 2024), and LNO (Wang & Wang, 2024). As shown in Table 23, CALM-PDE consistently outperforms 6 out of 8 baselines and ranks among the top models alongside LNO and Transolver. Additionally, the results demonstrate that CALM-PDE can generalize across complex, varying geometries. The hyperparameter for CALM-PDE are denoted in Table 24.

Table 23: Relative L2 errors of models trained and tested on the time-independent airfoil and elasticity datasets. The values in parentheses indicate the percentage deviation to CALM-PDE and underlined values indicate the second-best errors.

| Model | Relative L2 Error $\times 10^{-2}$ ($\downarrow$) | |
|---|---|---|
| | Time-independent Airfoil | Elasticity |
| Geo-FNO | 1.38 (+138%) | 2.29 (+332%) |
| F-FNO | 0.60 (+3%) | 1.85 (+249%) |
| Galerkin Transformer | 1.18 (+103%) | 2.40 (+353%) |
| OFormer | 1.83 (+216%) | 1.83 (+245%) |
| GNOT | 0.75 (+29%) | 0.88 (+66%) |
| ONO | 0.61 (+5%) | 1.18 (+123%) |
| Transolver | **0.47** (-19%) | 0.62 (+17%) |
| LNO | 0.51 (-12%) | **0.52** (-2%) |
| CALM-PDE | 0.58 | 0.53 |

Table 24: Hyperparameters for CALM-PDE used in the time-independent experiments.

| | Parameter | PDE | |
|---|---|---|---|
| | | Time-independent Airfoil | Elasticity |
| Encoder | Layers | 3 | |
| | Channels | [64, 96, 128] | |
| | Query Points | [128, 64, 64] | [1024, 256, 16] |
| | Percentile | [0.25, 0.5, 0.5] | [0.02, 0.02, 0.1] |
| | Temperature | [1.0, 1.0, 1.0] | [1.0, 1.0, 0.1] |
| Decoder | Layers | 3 | |
| | Channels | [96, 64, 1] | |
| | Query Points | [512, 2048, -] | [256, 1024, -] |
| | Percentile | [0.25, 0.01, 0.01] | |
| | Temperature | [1.0, 1.0, 1.0] | [0.1, 1.0, 1.0] |
| | Periodic Boundary | ✗ | |
| | Mesh Prior | ✗ | |
| | Processor Layers | 2 | |
| | Random Starting Points | - | |
| | Learning Rate | 1e-4 | 2e-4 |
| | Batch Size | 4 | |
| | Epochs | 500 | |
| | Parameters | 4,378,273 | 4,353,313 |

## O.2 Encoding and Decoding Capabilities

We train an encoder-decoder-only variant of CALM-PDE with a self-reconstruction loss (i.e., an autoencoder) to investigate the encoding and decoding capabilities of the model. We train CALM-PDE as an autoencoder on the 2D Navier-Stokes $\nu = 1e^{-4}$ dataset and compare it to the self-reconstruction error reported in AROMA (Serrano et al., 2024). The results in Table 25 show that CALM-PDE achieves a significantly lower reconstruction relative L2 error compared to AROMA.

Table 25: Relative L2 errors of AROMA and CALM-PDE models trained and tested on the 2D Navier-Stokes dataset. The value in parentheses indicates the percentage deviation to CALM-PDE.

| Model | Reconstruction Relative L2 Error $\times 10^{-2}$ ($\downarrow$) |
| --- | --- |
| | 2D Navier-Stokes $\nu = 1e^{-4}$ |
| AROMA | 1.049 (+141%) |
| CALM-PDE | **0.435** |

## O.3 Scaling of Output Query Mesh Size

To investigate the scaling behavior of the model for large meshes, we increase the number of output query points during inference. The experiment is conducted on the 2D Navier-Stokes $\nu = 1e^{-4}$ dataset, where the maximum resolution of the ground truth is limited to $64 \times 64$. Thus, we only provide the inference time and GPU memory consumption. The experiment is done on an NVIDIA A100 GPU with a batch size of 32 and predicting one timestep. Table 26 shows the scaling behavior of the model for increased output query mesh sizes.

Table 26: Inference time and memory consumption of CALM-PDE for different output query grid sizes on the 2D Navier-Stokes dataset.

| Qutput Query Mesh Size | 2D Navier-Stokes | |
| --- | --- | --- |
| | Inference Time [ms] | Inference Memory [GB] |
| $64 \times 64$ (4096) | 10.27 | 0.84 |
| $128 \times 128$ (16,384) | 14.83 | 1.80 |
| $256 \times 256$ (65,536) | 33.89 | 5.64 |
| $512 \times 512$ (262,144) | 107.52 | 21.00 |

# P   Notation

**Scalars, Vectors, and Multi-dimensional Arrays.**   We follow the convention and represent scalars with a small letter (e.g., $a$), vectors with a small boldfaced letter (e.g., $\boldsymbol{a}$), and matrices and N-dimensional arrays (with $N \geq 3$) with a capital boldfaced letter (e.g., $\boldsymbol{A}$).

**Partial Derivatives.**   The notation $\partial_{\boldsymbol{\omega}} \boldsymbol{u}, \partial_{\boldsymbol{\omega\omega}} \boldsymbol{u}, \ldots$ is short for the $i^{th}$ order (where $i \in \{1, 2, .., n\}$) partial derivative $\frac{\partial \boldsymbol{u}}{\partial \boldsymbol{\omega}}, \frac{\partial^2 \boldsymbol{u}}{\partial \boldsymbol{\omega}^2}, \ldots, \frac{\partial^n \boldsymbol{u}}{\partial \boldsymbol{\omega}^n}$.

