# OpenReview forum: "CALM-PDE: Continuous and Adaptive Convolutions for Latent Space Modeling of Time-dependent PDEs"
_NeurIPS.cc/2025/Conference — NeurIPS 2025 spotlight_

### Official Review · Reviewer_5iJC · 2025-06-17

**Clarity:** 4
**Significance:** 3
**Originality:** 3
**Rating:** 6
**Confidence:** 4

**Summary:**

The author proposes an encoder-decoder style latent space PDE solver for solving time-dependent PDEs. First, in the encoder, the author sets multiple levels of query points with decreasing quantities, gradually compressing the PDE function information in the spatial domain into a compact latent space through successive convolutions. Then, a Transformer is used to model the mapping of the PDE function. Finally, in the decoder, multiple levels of query points with increasing quantities are set to gradually decompress the PDE function information from the latent space back to the spatial domain. Learnable query points are introduced, enabling the encoding and decoding processes to adapt to the spatial domain.

**Questions:**

1. How is the number of query points in each layer of the encoder and decoder determined? Is there a quantitative method to calculate how many query points are needed?
2. The authors mention that their method does not perform well on PDE problems with many fine details. Did they actually conduct experiments on the Kolmogorov Flow problem? If so, what were the results?

**Ethical Concerns:**

["NO or VERY MINOR ethics concerns only"]

**Final Justification:**

The multi-level learnable query point design achieves a better balance between feature compression and preservation, and combined with continuous convolution, it further enhances the generality of the proposed method. I consider this an outstanding latent PDE solver paper and recommend accepting it.

**Limitations:**

As the authors mentioned, although CALM-PDE makes highly effective use of the latent space, its applicability to real-world scenarios involving a larger number of points is still limited due to the substantial computational cost. However, this is a common limitation shared by all current neural operator methods and should not be considered a drawback specific to this work.

**Quality:**

4

**Strengths And Weaknesses:**

## Strengths
1. The learnable multi-level query point design ensures that patterns of varying complexity across different spatial locations receive appropriate attention, enabling effective compression of spatial domain information into the latent space.
2. The use of continuous convolution allows the model to handle PDE problems on irregular grids and decouples observation and prediction points. Moreover, the neighborhood constraints enable the model to capture local high-frequency information during operator learning, which pure Transformer-based neural operator models generally lack.
3. The proposed model demonstrates excellent performance in both accuracy and efficiency across a wide range of 2D and 3D time-dependent PDE benchmarks.
4. The authors provide a thorough categorization of latent PDE solvers and a well-articulated discussion on the differences and relationships among related works, offering valuable guidance for future research.

## Weaknesses
Except for the experiment on the 2D Navier-Stokes problem (ν = 1e−4), the accuracy and efficiency of CALM-PDE are generally comparable to those of FNO [1] or Geo-FNO[2], while FNO appears to have a simpler architectural design. In this regard, the advantages of CALM-PDE merit further investigation.

[1] Zongyi Li et al. Fourier neural operator for parametric partial differential equations. ICLR 2021.

[2] Zongyi Li et al. Fourier neural operator with learned deformations for pdes on general geometries. JMLR 2023.

---

> ### Author Rebuttal · Authors · 2025-07-29
>
> Thank you for your effort in reviewing our work and your valuable feedback. We appreciate that you recognize the method, featuring the hierarchical query point design and the limited neighborhood, as important to effectively solve PDEs. We are happy to address the mentioned weakness and questions below.
>
> > **(W1)** Except for the experiment on the 2D Navier-Stokes problem (ν = 1e−4), the accuracy and efficiency of CALM-PDE are generally comparable to those of FNO [1] or Geo-FNO[2], while FNO appears to have a simpler architectural design. In this regard, the advantages of CALM-PDE merit further investigation.
>
> We would like to emphasize that we use a curriculum strategy to train the autoregressive models, including FNO and Geo-FNO, which improves the error compared to a fully autoregressive training (see Appendix I, Table 6, line 799). For instance, the curriculum training of FNO improves the error by 39% on the 2D Navier-Stokes $\nu=1e^{-4}$ dataset. Thus, we use strong baselines and incorporate additional techniques to improve them further. In general, CALM-PDE is a viable and competitive alternative to commonly used FNOs and Transformer architectures for solving PDEs. The model demonstrates the effectiveness of convolutional models on irregularly sampled spatial domains and the benefit of sampling the spatial domain via adaptive query points.
>
> > **(Q1)** How is the number of query points in each layer of the encoder and decoder determined? Is there a quantitative method to calculate how many query points are needed?
>
> We determine the number of query points based on the principle that the number of query points, receptive field, and channel dimension must cover the entire spatial domain. Similar to conventional CNNs, a small receptive field with a large stride (e.g., only a few query points) misses information, while a large receptive field with a small stride could have an averaging effect. With this principle, we obtain the number of query points (1024, 256, 16 for the encoder layers), which we use for all 2D experiments. Thus, a selected number of query points works across different problems and does not need individual hyperparameter search.
>
> > **(Q2)** The authors mention that their method does not perform well on PDE problems with many fine details. Did they actually conduct experiments on the Kolmogorov Flow problem? If so, what were the results?
>
> Yes, we have experimented with the Kolmogorov Flow problem from [1] by training only the encoder-decoder model with a self-reconstruction loss (autoencoder) and obtained a relative error of 0.0928 for a simple reconstruction, which is already worse compared to the relative error of 0.8156 for a rollout reported for DilResNet in [1]. Thus, we did not continue the experiment.
>
> **References** \
> [1] Z. Li, D. Shu, and A. B. Farimani, “Scalable Transformer for PDE Surrogate Modeling,” arXiv [cs.LG], 2023.

---

> > ### Comment · Reviewer_5iJC · 2025-08-05
> >
> > I appreciate the authors' detailed clarification, which has addressed all of my concerns. I will keep my original score.

---

> > > ### Author Response · Authors · 2025-08-05
> > >
> > > We appreciate your efforts in reviewing our work and your response. We are happy to hear that we have addressed all of your concerns.

---

### Official Review · Reviewer_27Kq · 2025-06-26

**Clarity:** 3
**Significance:** 3
**Originality:** 2
**Rating:** 4
**Confidence:** 3

**Summary:**

CALM-PDE proposes continuous convolutionnal layers as encoder/decoder components. These layers have the abilities to be used on irregular grids while being based on convolutions. The layer are explained in details and then evaluated on regular and irregular grids a studied through ablations in the appendices.

**Questions:**

-	What are the differences between LNS/LTS and RO discussed in the related work?
-	Can the number of point N differs between samples or must it be fixed ?
-	What happens if there are no points in a considered neighboorhood ?
-	Lines 40-42, « mainly utilises CNNs ... or transformers » but no example are given here, I think this sentence should be illustrated. You could add some references (some of them are already cited in the paper such as AROMA or UPT), this point is only for clarity when reading the paper
- what makes LNO 3x heavier in terms of memory consumption as FNO?
-	How do evolve the performances wrt to query grid size ? in term of computational cost/accuracy?
-	Have you considered other architecture for the process ?
- Why not considering AROMA or UPT as baselines? could CALM-PDE handle parametric PDEs?

**Ethical Concerns:**

["NO or VERY MINOR ethics concerns only"]

**Final Justification:**

CALM-PDE introduces a convolution-based layer designed to operate on irregular geometries. While most existing methods rely on attention mechanisms or graph-based approaches to handle and compress data on irregular grids, CALM-PDE offers a novel alternative for tackling this challenging setting with convolutions.

The authors have responded to my questions and provided new experiments to support their claims. I particularly appreciate their honesty in acknowledging the limitations of their method (ie performances and computational cost), which makes this work a promising basis for future research in this direction.

Although CALM-PDE does not achieve state-of-the-art performance across the presented benchmarks, its results remain competitive and convincing relative to other baselines.

**Limitations:**

Yes

**Quality:**

3

**Strengths And Weaknesses:**

### Strenght :
 -	The paper proposes a precise description of the model
-	The text is well written and I found it easy to follow
-	A lot of additional experiments and details are available in appendices for in depth comprehension
-	I found the idea to be Interesting and it could be useful
### Weaknesses :
-	The contribution of the method seems to be the encoding/decoding since the process is a Neural ODE. I think a deeper study of encoding/decoding performances/effects would help the reader clearly identify the benefits of the model.
-	I would lower the first claim "a new model that solves ... PDEs", because the major contribution relies in the encoding/decoding architectures.
- One of the arguments for the model being the computational cost, I think a deeper comparison on this aspect with other model would strengthen the argumentation. What are the computational cost wrt to existing method (mostly transformer based method such as AROMA/UPT)? How performs CALM-PDE in the encoding tasks wrt to other encoding architectures?

---

> ### Author Rebuttal · Authors · 2025-07-29
>
> Thank you for your feedback and your questions. Your constructive feedback helped us to improve our submission. We appreciate that you consider our method interesting and helpful. Below, we address your concerns and questions in detail.
>
> > **(W1)** The contribution of the method seems to be the encoding/decoding since the process is a Neural ODE. I think a deeper study of encoding/decoding performances/effects would help the reader clearly identify the benefits of the model.
>
> Yes, the main contribution of the method is the novel discretization-agnostic encoder and decoder based on continuous and adaptive convolution. The latent time stepping can be interpreted as solving a neural ODE. However, there are different ways of parametrizing the neural ODE, and previous work relied on point-wise MLPs [1, 2], while we utilize a Transformer with a combination of vanilla and position-attention [3] to parametrize the neural ODE. We added additional experiments that investigate the scaling of the latent query points and the output query points to provide more insights into the encoder-decoder architecture. Additionally, we add an explanation of the computational costs for the encoder. Please see (Q6) and (W3).
>
> > **(W2)** I would lower the first claim "a new model that solves ... PDEs", because the major contribution relies in the encoding/decoding architectures.
>
> As mentioned in the response for (W1), we agree that the main contribution is the novel encoder-decoder architecture and not the paradigm itself. Thus, we’ve updated the paper and changed it from “a new model” to “a model with a new encoder-decoder mechanism” to better highlight the main contribution.
>
> > **(W3)** One of the arguments for the model being the computational cost, I think a deeper comparison on this aspect with other model would strengthen the argumentation. [...] How performs CALM-PDE in the encoding tasks wrt to other encoding architectures?
>
> We also compare our model against OFormer (linear attention) and LNO (vanilla attention), which are encoder-decoder architectures that utilize attention for encoding and decoding. As shown in Figure 4 and Table 3 on page 9, CALM-PDE achieves a lower inference time and memory consumption compared to these models. Furthermore, we added an explanation to the paper to clarify the computational cost of the models. Let N be an arbitrary number of input points. For simplicity, we only consider one CALM encoder layer. The output of the layer are features for N’ output or query points, where N’ is a constant number. We also have to consider the percentile p, which determines the size of the receptive field and the feature or channel dimension d. This yields that $\mathcal{O}(p \cdot N \cdot N’ \cdot d)$ computations are needed. Models such as AROMA or LNO do not consider a receptive field, which requires $\mathcal{O}(N \cdot N’ \cdot d)$ computations for N’ query tokens. Since p is usually small (we use p=0.01), the complexity is reduced by two orders of magnitude compared to models without a receptive field.
>
> > **(Q1)** What are the differences between LNS/LTS and RO discussed in the related work?
>
> We use the term Reduced Order Modeling (RO) to generally refer to models that reduce the model complexity in some way. We further introduce the terms Latent Neural Surrogates (LNS) and Latent Time Stepping (LTS) as concrete ways of reducing the model complexity. For instance, LNS reduces the model complexity by reducing the spatial resolution, and LTS reduces the model complexity by computing the PDE trajectory completely in the latent space. LNS and LTS can be combined (e.g., AROMA), but they can also be used separately (only LNS for LNO and only LTS for OFormer).
>
> > **(Q2)** Can the number of point N differs between samples or must it be fixed ?
>
> Yes, the number of input points N can be different between samples, which is the case in the cylinder dataset. Due to continuous convolution, each query point can take an arbitrary and changing number of input points.
>
> > **(Q3)** What happens if there are no points in a considered neighboorhood ?
>
> This problem can only happen if the epsilon is fixed, which is not the case for our model. We adopt the mechanism from PiT [3], which computes the epsilon dynamically based on the p-th percentile of the Euclidean distances. This simply means that each query point has at least N*p points as input. If the input points are far away, the epsilon will be increased, and vice versa. Please see Appendix E.1, line 574 for details.
>
> > **(Q4)** Lines 40-42, « mainly utilises CNNs ... or transformers » but no example are given here, I think this sentence should be illustrated. You could add some references [...], this point is only for clarity when reading the paper
>
> Thanks for pointing that out. We changed the corresponding lines and added LE-PDE [4] as an example for a model using CNNs as well as AROMA and UPT for transformer-based models.
>
> > **(Q5)** what makes LNO 3x heavier in terms of memory consumption as FNO?
>
> The key difference that makes LNO heavier in terms of memory consumption and time is the use of attention compared to the use of the fast Fourier Transform (FFT). The encoder and decoder of LNO are based on cross-attention, which makes it expensive. In particular, cross-attention between a fixed number of query tokens (queries in attention) and all input points (values, keys in attention) is computed to compress the input into a smaller latent representation. In the 2D Navier-Stokes experiments, LNO uses 128 query tokens and the input has 4096 input points, which requires $\mathcal{O}(128\cdot4096)$ computations. For decoding, this process is reversed by using the latent tokens as key, value in attention, and the query positions as queries. The latent representation is processed with a Transformer that computes self-attention, which requires $\mathcal{O}(128\cdot128)$ computations. In contrast, FNO makes use of the FFT in each layer with a complexity of $\mathcal{O}(N \cdot log(N))$, which yields $\mathcal{O}(4096 \cdot log(4069))$ for the 2D Navier-Stokes dataset, which is more efficient.
>
> > **(Q6)** How do evolve the performances wrt to query grid size ? in term of computational cost/accuracy?
>
> We’ve conducted an experiment on the 2D Navier-Stokes $\nu=1e^{-4}$ dataset (max. resolution of ground truth is 64x64) where we increase the **output** query grid size during inference. The numbers are obtained on an A100 40GB GPU with a batch size of 32 and predicting one timestep.
>
> | Output Query Grid Size | Inference Time (ms) | Inference Memory Consumption (GB)
> | ----------- | ---------------------- | ---------- |
> | 64x64 (4096) | 10.27 | 0.84
> | 128x128 (16.384) | 14.83 | 1.80
> | 256x256 (65.536) | 33.89 | 5.64
> | 512x512 (262.144) | 107.52 | 21.00
>
> Additionally, we’ve conducted an experiment where we change the **latent** query grid size (number of latent points). We measure the relative L2 error on the test set and total training time on the 2D Navier-Stokes dataset with $\nu=1e^{-4}$. The bold row shows the configuration used in the paper. The results show that we can expect a saturation effect in the error.
>
> | Encoder Query Points [Layer 1, Layer 2, Layer 3] | Test Relative L2 Error | Training Time |
> | ------------------------------------------------ | ---------------------- | ------------- |
> | [256, 64, 4]  | 0.0533 | 4:18h |
> | [512, 128, 8] | 0.0425 | 4:40h |
> | **[1024, 256, 16]** | **0.0304** | **5:25h** |
> | [2048, 512, 32] | 0.0251 | 6:21h |
> | [4096, 1024, 64] | 0.0267 | 8:19h |
>
>
> > **(Q7)** Have you considered other architecture for the process ?
>
> Besides a Transformer, we’ve considered a pointwise MLP as the latent processor. However, we decided to use a Transformer instead of an MLP to allow the processor to have a global view, which would not be the case for a pointwise MLP, and to allow latent tokens to dynamically interact with each other with attention.
>
> > **(Q8)** Why not considering AROMA or UPT as baselines?
>
> We've added AROMA as an additional baseline. Please note that we use a different setup (e.g., different rel. L2 error computed on non-normalized data, no in-t/ out-t split, etc.) compared to the AROMA paper.
>
> | Model | 2D Navier-Stokes $\nu=1e^{-4}$ | 2D Navier-Stokes $\nu=1e^{-5}$  | 2D Airfoil | 2D Cylinder |
> | ------- | ----------------- | -------------- |--------|-----------|
> | AROMA | 0.1257 (+318%) | 0.1801 (+74%) | 0.0403 (-22%) | 0.1140 (-4%) |
> | CALM-PDE | 0.0301 | 0.1033 | 0.0515 | 0.1186 |
>
> We've also measured the inference times on the 2D Navier-Stokes $\nu=1e^{-4}$ dataset, and AROMA needs 5.18s, in comparison to CALM-PDE with 0.37s. Reducing the refinement steps to 1 reduces the inference time to 3.81s.
>
> > could CALM-PDE handle parametric PDEs?
>
> We did not experiment with parametric PDEs with varying PDE parameters, but CALM-PDE can be easily adapted to handle such cases. For instance, the PDE parameter could be added as an additional input channel, or a more sophisticated conditioning mechanism could be used to incorporate the PDE parameter [5, 6].
>
>
> **References** \
> [1] Z. Li, K. Meidani, and A. B. Farimani, “Transformer for partial differential equations’ operator learning,” arXiv [cs.LG], 2022. \
> [2] L. Serrano et al., “Operator learning with neural fields: Tackling PDEs on general geometries,” arXiv [cs.LG], 2023. \
> [3] J. Chen and K. Wu, “Positional knowledge is all you need: Position-induced Transformer (PiT) for operator learning,” arXiv [cs.LG], 2024. \
> [4] T. Wu, T. Maruyama, and J. Leskovec, “Learning to accelerate Partial Differential Equations via latent global evolution,” arXiv [cs.LG], 2022. \
> [5] M. Takamoto, F. Alesiani, and M. Niepert, “Learning neural PDE solvers with parameter-guided Channel Attention,” arXiv [cs.LG], 2023. \
> [6] J. Hagnberger, M. Kalimuthu, D. Musekamp, and M. Niepert, “Vectorized Conditional Neural Fields: A framework for solving time-dependent parametric Partial Differential Equations,” arXiv [cs.LG], 2024.

---

> > ### Comment · Reviewer_27Kq · 2025-08-01
> > **Answer to Authors' Rebuttal**
> >
> > I thank the authors for their clarifications, their answers to my questions, and the additional experiments. They have addressed most of my concerns. After reading the rebuttal, I now have the following additional questions and remarks:
> >
> > (W1 & 3) Thank you for your response. My point here was more focused on the encoding/decoding task. How does CALM-PDE perform as an autoencoder? Stated otherwise, with a common process, how would CALM-PDE perfom compared to other auto-encoder architecture ? What does the compressed representation of the physical field look like? What does the model learn in terms of latent representation?  I have seen some illustrations in appendix K about the location of the learned latent positions, which seems to be gathered to the location where the the dynamics are more complex – is this correct ?
> >
> > (W1 & Q7) Regarding the latent time-stepping, I was also thinking about other alternatives, such as the standard FNO in a forward manner (ie without neural ODE). I am not necessarily requesting additional experiments, since the rebuttal period is short, however, I was wondering if other processes had been tested before ? Maybe this could improve the performance of CALM-PDE.
> >
> > (Q6) Thanks for the clear comparison. What part of CALM-PDE layers would be the computational overhead for scaling to real-world application ? My initial guess would be the neighbor search step, which impose to compute a distance matrix between all the points of the sequence, and can be pretty costly, but I am not sure about it.
> > While reviewing the paper (and reading the response), I (re)went through the code and my point about computational efficiency/training time was mainly about batching operation. Does the neighbors search works batches when grid have different size such as in the Cylinder Flow dataset ?

---

> > > ### Author Response · Authors · 2025-08-02
> > > **Reply to Reviewer's Questions and Comments**
> > >
> > > Thank you for taking the time to read our rebuttal and for your response. We're glad to see that most of your concerns have been addressed. Below, you'll find our replies to your additional questions and comments.
> > >
> > > > **(W1 & 3)** Thank you for your response. My point here was more focused on the encoding/decoding task. How does CALM-PDE perform as an autoencoder? Stated otherwise, with a common process, how would CALM-PDE perfom compared to other auto-encoder architecture ?
> > >
> > > Thank you for the clarification. We trained CALM-PDE as an autoencoder on the 2D Navier-Stokes $\nu=1e^{-4}$ dataset. It achieves a reconstruction relative L2 error of $0.4348 \times 10^{-2}$. For comparison, AROMA reports a reconstruction error of $1.049 \times 10^{-2}$, as documented in Appendix C.6 of the AROMA paper.
> > >
> > > > What does the compressed representation of the physical field look like? What does the model learn in terms of latent representation?
> > >
> > > The compressed latent representation is a sequence of latent tokens and their positions (i.e., learned query positions). Adding noise to the latent representations shows that each token encodes information of a small region of the physical field (i.e., the surroundings of its query position). Figure 12 in Appendix K (line 872) illustrates how adding noise to a token only affects the physical field in the surroundings of that token/position. So, the latent tokens essentially partition the physical field.
> > >
> > >
> > > > I have seen some illustrations in appendix K about the location of the learned latent positions, which seems to be gathered to the location where the the dynamics are more complex – is this correct ?
> > >
> > > Yes, that is correct. CALM-PDE samples more query points in locations where the dynamics are more complex and where obstacles are located.
> > >
> > > > **(W1 & Q7)** Regarding the latent time-stepping, I was also thinking about other alternatives, such as the standard FNO in a forward manner (ie without neural ODE). I am not necessarily requesting additional experiments, since the rebuttal period is short, however, I was wondering if other processes had been tested before ? Maybe this could improve the performance of CALM-PDE.
> > >
> > > We did not experiment with different approaches, except for using a pointwise MLP and a Transformer to parametrize the neural ODE. Using different strategies for computing the dynamics, such as combining the strengths of CALM-PDE and FNO, could be an interesting area for future work.
> > >
> > > > **(Q6)** Thanks for the clear comparison. What part of CALM-PDE layers would be the computational overhead for scaling to real-world application ? My initial guess would be the neighbor search step, which impose to compute a distance matrix between all the points of the sequence, and can be pretty costly, but I am not sure about it.
> > >
> > > Exactly, the model requires computing a distance matrix with $N \times N’$, where N is the number of input points and N’ the number of query points. For practical applications, N will be large, which makes it costly in terms of computation and memory.
> > >
> > >
> > > > While reviewing the paper (and reading the response), I (re)went through the code and my point about computational efficiency/training time was mainly about batching operation. Does the neighbors search works batches when grid have different size such as in the Cylinder Flow dataset ?
> > >
> > > Yes, the neighbor search also works for padded batches with different grid sizes. We apply padding to the grid to compute the neighborhood search in a batched fashion. By using a position of $x = y= \infty$ as padding, we ensure that the padding points are not selected as neighbors.

---

> > > > ### Comment · Reviewer_27Kq · 2025-08-04
> > > > **Answer to Authors' Rebuttal (2)**
> > > >
> > > > I thank the authors for their response and the additional explanations. Considering the rebuttal suggestions’ discussed before (ie with the additional experiments and precisions), I will raise my score to 4.

---

> > > > > ### Author Response · Authors · 2025-08-05
> > > > >
> > > > > Thank you for your effort in reviewing our work and your response. We are glad to see that we have addressed most of your concerns.

---

### Official Review · Reviewer_dQvB · 2025-06-30

**Clarity:** 4
**Significance:** 2
**Originality:** 2
**Rating:** 5
**Confidence:** 4

**Summary:**

The paper proposes a novel architecture (CALM-PDE) for reduced-order modeling of arbitrarily discretized PDEs. The building block CALM layer uses continuous convolution and adaptively learn query points. Numerical experiments on initial value problems in fluid dynamics show the effectiveness of the method.

**Questions:**

Q1) What values of positions $P$ of latent tokens in Equation (8) can take? How are they initialized and learned?

Q2) How to determine the appropriate dimensionality of the latent variables? Will we observe a saturation effect or diminishing returns in prediction accuracy as the latent dimension increases? Should we use the same hyperparameters for different latent dimensions?

Q3) Can the model handle cases where the physical domain geometry itself is treated as a random input parameter to the neural operator? That is, can it generalize across varying domain shapes or boundary configurations as part of the input space?

**Ethical Concerns:**

["NO or VERY MINOR ethics concerns only"]

**Final Justification:**

The presentation and experiments in the paper are solid and of high quality. In the rebuttal, the authors address my concern about the generalizability and robustness of the method to different PDE setups. Although its current version does not outperform all baselines in terms of accuracy, I think it is still competitive. The ideas the paper offers are inspiring, together with its fruitful experimental results, could be worthy to the research community.

**Limitations:**

Yes, adequately described in the paper.

**Quality:**

3

**Strengths And Weaknesses:**

[Strengths]

S1) The paper is clearly written and well-motivated (particularly in Section 6, which presents the experiments and evaluation through Q\&A effectively).

S2) The appendix is thorough and of high quality, offering essential details that enhance understanding and reproducibility of the main text. I found the visualizations of the learned positions in Section K quite informative.

S3) Learning parametric PDEs on irregular meshes is challenging but highly meaningful for real-world applications. The proposed use of a discretization-agnostic decoder, combined with adaptive learning of query points, presents a general strategy that could be broadly applicable to other neural operator frameworks.

[Weaknesses]

W1) The results on the 2D Navier--Stokes equation with high Reynolds number seem not strong enough compared to FNO in terms of prediction accuracy, training time (and memory), and inference time.

W2) There could be more hyperparameters (such as temperature $T$, receptive field radius $\varepsilon$, number of query points, etc) to tune compared to other baseline models. And it is unclear how robust the model is to these hyperparameters' change.

W3) There has been a lot of follow-up work on reducing the memory consumption of the transformer-based methods. It is possible that using a more efficient way of attention mechanism, transformer-based methods still perform equally well or even better.

---

> ### Author Rebuttal · Authors · 2025-07-29
>
> Thank you for carefully reviewing our work and asking insightful questions, which helped us to improve our submission. We are grateful that you recognize our approach as a strategy with the potential for wide applicability across various neural operator frameworks. We address your concerns and questions separately as follows.
>
> > **(W1)** The results on the 2D Navier--Stokes equation with high Reynolds number seem not strong enough compared to FNO [...]
>
> We developed the architecture for irregularly sampled spatial domains, which are important for many real-world applications, as you already pointed out in strength 3. We included the benchmark on regular grids to demonstrate that our approach is also directly applicable to such settings and yields competitive errors. Thus, CALM-PDE is not specially optimized for regular grids. Since the grid is uniformly structured, the inference time, training time, and memory consumption could be improved by exploiting the uniform structure, which we consider as interesting future work. However, we specifically target PDEs with irregularly sampled spatial domains with our method.
>
> > **(W2)** There could be more hyperparameters [...] to tune compared to other baseline models. And it is unclear how robust the model is to these hyperparameters' change.
>
> The model introduces the softmax temperature, receptive field radius, channel dimension, and number of query points as hyperparameters. The receptive field radius is equivalent to the kernel size in conventional CNNs and the number of query points to downsampling (e.g., stride and pooling) in CNNs. The channel dimension is equivalent to the channel dimension in CNNs. Alternatively, the number of query points can be compared to the query points used in other models, such as AROMA [1] or LNO [2]. Thus, our architecture does not introduce more hyperparameters.
>
> Selecting good hyperparameters for our model does not require an extensive hyperparameter search. As shown in Table 11 in Appendix J.4 (line 856), we use the same temperatures, receptive field sizes (percentiles), and number of query points for all 2D experiments, which demonstrates that the hyperparameters are robust and work across different problems.
>
> Only the temperature is a new hyperparameter that controls the distance weighting within the receptive field. We also use the same temperatures across all experiments, demonstrating its robustness. However, we’ve conducted a hyperparameter study for the temperature, which shows that it is also robust to changes ($T \in [0.1, 5]$ yields small changes in error). We ran the experiments on the 2D Navier-Stokes $\nu=1e^{-4}$ dataset.
>
> | Temperature T in Encoder and Decoder | Rel. L2 Error |
> | ----------- | ---------------------- |
> | 10 | 0.0398 |
> | 5 | 0.0336 |
> | 2 | 0.0316 |
> | 1 | 0.0304 |
> | 0.1 | 0.0341 |
> | 0.01 | 0.1640 |
>
> The table shows that T=1 yields the lowest errors and that a slightly smaller (e.g., T=0.1) or larger (e.g., T=5) temperature results only in small changes in error. This demonstrates the robustness of the temperature parameter.
>
>
> > **(W3)** There has been a lot of follow-up work on reducing the memory consumption of the transformer-based methods. [...]
>
> In contrast to (efficient) transformer-based methods, CALM-PDE introduces a fundamentally different and efficient architecture grounded in continuous convolution. The model demonstrates that convolution can be applied to irregularly sampled spatial domains, where usually transformer-based methods dominate [1, 2, 3], and that restricting the receptive field inherently reduces the computational and memory complexity. Thus, CALM-PDE is a competitive alternative to (efficient) transformer-based methods. To address concerns around efficiency, we compare our method with OFormer, which leverages linear attention to improve memory and runtime of the attention mechanism (see Figure 4 and Table 3 on page 9). Importantly, the implementation of CALM-PDE is not heavily optimized for GPUs, similar to the implementations of the baselines, which we consider a fair comparison. However, with optimized implementations, the efficiency of CALM-PDE can be improved further, which we aim to investigate as interesting future work.
>
>
> > **(Q1)**  What values of positions $P$ of latent tokens in Equation (8) can take? How are they initialized and learned?
>
> The positions $P$ of the latent tokens are the learned query positions of the last encoder layer. We did not constrain the positions that the latent tokens can take. Thus, they could potentially take an arbitrary position. For periodic domains, this is not an issue because the positions are corrected and will be periodically mapped to $[0,1]^n$. For non-periodic domains, we noticed that the positions do not move too far away and stay close to the spatial domain. We opt for this design to keep the method simple. However, additional regularization to force the positions to stay in $[0,1]^n$ could be tested for future work. The plots with the title “Latent Position” in Appendix K (pages 27 and 28) show the learned positions $P$.
>
> We have two ways of initializing the query positions. a) They are uniformly sampled from $[0,1]^n$ to achieve a uniform cover of the spatial domain. b) They are sampled from a training mesh to have a better initialization and to capture the underlying mesh structure. We use b) only for the airfoil experiment, where we have one irregular mesh for all samples, and a) for the remaining experiments.
>
> The query points are learned similarly to the weights and biases of a neural network. The query points influence how the input features are weighted. Thus, we can compute the gradient of the loss function with respect to the query positions. The gradient is used to optimize the query positions (i.e., move the query points) with a variant of stochastic gradient descent (we use AdamW [4]).
>
> Details about the initialization and learning of the query positions are explained in the paragraph “Learnable Query Points" in line 132 in the paper.
>
>
> > **(Q2)** How to determine the appropriate dimensionality of the latent variables?
>
> The dimensionality of the latent variables is defined by the number of latent variables or query points and their channel dimension. We determine the dimensionality of the latent variables based on the principle that the number of query points, receptive field, and channel dimension must cover the entire spatial domain. Similar to conventional CNNs, a small receptive field with a large stride (e.g., only a few query points) misses information, while a large receptive field with a small stride could have an averaging effect. With this principle, we obtain the number of query points (1024, 256, 16 for the encoder layers), which we use for all 2D experiments. Thus, a fixed number of query points works across different problems.
>
>
> > Will we observe a saturation effect or diminishing returns in prediction accuracy as the latent dimension increases?
>
> We’ve conducted an experiment where we change the latent dimensionality by changing the number of latent points, and the results show that we will observe a saturation effect. We measure the relative L2 error on the test set and total training time on the 2D Navier-Stokes dataset with $\nu=1e^{-4}$. The bold row shows the configuration used in the paper.
>
> | Encoder Query Points [Layer 1, Layer 2, Layer 3] | Test Rel. L2 Error | Training Time |
> | ------------------------------------------------ | ---------------------- | ------------- |
> | [256, 64, 4]  | 0.0533 | 4:18h |
> | [512, 128, 8] | 0.0425 | 4:40h |
> | **[1024, 256, 16]** | **0.0304** | **5:25h** |
> | [2048, 512, 32] | 0.0251 | 6:21h |
> | [4096, 1024, 64] | 0.0267 | 8:19h |
>
>
> > Should we use the same hyperparameters for different latent dimensions?
>
> The number of query points, receptive field, and channel dimension are related to each other. Thus, it is recommended to adapt them jointly. When decreasing the number of latent points, the receptive field and channel dimension have to be increased and vice versa.
>
>
> > **(Q3)** [...] That is, can it generalize across varying domain shapes or boundary configurations as part of the input space?
>
> Yes, the model can also handle varying domain shapes and boundary conditions. The cylinder dataset covers exactly this case. Each sample in the train and test set has a different domain shape and boundary condition (i.e., different diameter and position of the cylinder). The results show that CALM-PDE can also generalize across different domain shapes and boundary conditions.
>
> We have also conducted an additional experiment on the time-independent airfoil and elasticity datasets from Geo-FNO [5]. The task is to map an input geometry (e.g., changing airfoil shape) to a physical quantity. We follow the setup from the LNO [2] paper and take the errors for the baselines from their paper. The results also demonstrate that CALM-PDE can generalize across complex, varying geometries.
>
> | Model | Time-indepdent Airfoil ($\times 10^{-2}$ Rel. L2 Error) | Elasticity ($\times 10^{-2}$ Rel. L2 Error)
> | ----------- | ---------------------- | ---------- |
> | Geo-FNO | 1.38 | 2.29 |
> | F-FNO | 0.60 | 1.85 |
> | Galerkin | 1.18 |  2.40 |
> | OFormer  | 1.83 | 1.83 |
> | GNOT |  0.75 | 0.88 |
> | ONO | 0.61 | 1.18 |
> | Transolver | 0.47 | 0.62 |
> | LNO | 0.51 | 0.52 |
> | CALM-PDE | 0.58 | 0.53 |
>
> The results show that CALM-PDE outperforms 6 out of 8 baselines and that it ranks among the top models alongside LNO [2] and Transolver [6].
>
> **References** \
> [1] AROMA: Preserving spatial structure for latent PDE modeling with local neural fields \
> [2] Latent Neural Operator for solving forward and inverse PDE problems \
> [3] Transformer for partial differential equations’ operator learning \
> [4] Decoupled weight decay regularization \
> [5] Fourier neural operator with learned deformations for PDEs on general geometries \
> [6] Transolver: A fast transformer solver for PDEs on general geometries

---

> > ### Comment · Reviewer_dQvB · 2025-08-04
> >
> > Thank you for your detailed response, which cleared most of my concerns and confusion. I am raising my score accordingly.

---

> > > ### Author Response · Authors · 2025-08-05
> > >
> > > Thank you for taking the time to review our paper and reading our rebuttal. We are glad to hear that we have addressed most of your concerns.

---

### Official Review · Reviewer_2MgG · 2025-06-30

**Clarity:** 3
**Significance:** 2
**Originality:** 2
**Rating:** 4
**Confidence:** 3

**Summary:**

This paper introduces CALM-PDE, a neural surrogate model for PDE modeling which compresses the spatial domain into a latent representation using continuous convolutional layers with adaptive query points. The method handles arbitrarily discretized (irregular) spatial meshes and evolves the solution entirely in latent space, achieving competitive accuracy on several benchmark PDEs while significantly reducing memory usage and inference time compared to various baselines.

**Questions:**

1) Can the authors show a comparison of performance and efficiency to Factorized FNO? it is a variant of FNO which performs competitively to other baselines described in the paper on similar tasks. Given its relevance and efficiency, particularly on similar problems, it seems like an important missing baseline.
2) I also think another recent architecture like Transolver should be discussed and compared to. The best comparisons would be on the irregular grid problems like airfoil, elasticity, plasticity or pipe problems commonly used as baselines for these problems.

**Ethical Concerns:**

["NO or VERY MINOR ethics concerns only"]

**Final Justification:**

The authors have provided some new results. One sticking point is I really think this method will not work that well had the authors used more complex datasets, such as Rayleigh Benard simulations (or more complex MHD simulations in Physics), as I mentioned during discussions.

So I am unsure this architecture will actually become useful in the future. But I appreciate the author's efforts during the rebuttal and some people at this conference might still be interested to hear about this; so I'm fine with increasing my score by 1 point. But I am not going to champion this paper.

**Limitations:**

yes

**Quality:**

3

**Strengths And Weaknesses:**

Strengths:

1) The method seems to work effectively for both regular and irregular grids.
2) It also achieves better inference time scaling wrt recent models like LNO (latent neural operator).
3) The paper is written clearly and is easy to read.

Weaknesses:

1) The accuracy improvements are not universal. In some benchmarks CALM-PDE is outperformed by other models. For example, on one or two regular-grid PDE task the model fell behind FNO and LNO, and on an irregular airfoil flow the Fourier-based Geo-FNO achieved better accuracy. This suggests CALM-PDE’s advantages (for reducing the error) are context-dependent rather than across-the-board. This also suggests that more baselines and datasets are needed for establishing a clearer gain in errors. Other papers in this field compare to various other irregularly spaced datasets such as elasticity, plasticity etc. See the LNO paper for example.

2) Some important baselines such as factorized FNO (FFNO) and transolver, which are quite competitive in this domain are missing.

3) By compressing the solution into a smaller latent space, the method inevitably discards some high-frequency or fine-grained information.

---

> ### Author Rebuttal · Authors · 2025-07-29
>
> Thank you for your insightful feedback and thoughtful questions, which helped us to enhance our submission. We appreciate acknowledging the versatility of our approach in handling both regular and irregular grids effectively. We've carefully reviewed each of your points and provide our responses below.
>
> > **(W1)** The accuracy improvements are not universal. [...] Other papers in this field compare to various other irregularly spaced datasets such as elasticity, plasticity etc. See the LNO paper for example.
>
> As you already pointed out with strength 1, CALM-PDE works effectively for both regular and irregular grids. We would like to emphasize that we do not claim CALM-PDE to be the best model across all problems, as no single model can perform best on all PDE tasks. Rather, our goal is to demonstrate that CALM-PDE is a strong and competitive alternative, particularly in challenging settings such as irregular spatial domains. Indeed, CALM-PDE outperforms all baselines in 3 out of 6 benchmarks and achieves second-best errors in 2 out of 6 benchmarks. Even on the remaining benchmark, CALM-PDE outperforms two transformer-based baselines, reducing the error by 58%. Overall, these results provide evidence that CALM-PDE is a viable and competitive alternative to commonly used FNOs and Transformer architectures for solving PDEs. Notably, the experiments demonstrate the effectiveness of convolutional models on irregularly sampled spatial domains and the benefit of sampling the spatial domain via adaptive query points.
>
> Moreover, we use strong baselines and incorporate additional techniques to further improve them. For instance, we train the autoregressive models (e.g., FNO and Geo-FNO) with a curriculum strategy, which significantly reduces their errors by 39% compared to full autoregressive training (see Table 6 in Appendix I, line 799). Taken together, these results position CALM-PDE as a novel and competitive approach for neural PDE solving.
>
> Following your suggestion, we’ve incorporated the airfoil and elasticity datasets into our analysis. While these datasets are time-independent, our primary focus remains on time-dependent PDEs. We follow the benchmark setup of the LNO paper and provide the baseline errors reported in that work  (see table below).
>
> | Model | Time-independent Airfoil ($\times 10^{-2}$ Rel. L2 Error) | Elasticity ($\times 10^{-2}$ Rel. L2 Error)
> | ----------- | ---------------------- | ---------- |
> | Geo-FNO | 1.38 | 2.29 |
> | F-FNO | 0.60 | 1.85 |
> | Galerkin | 1.18 |  2.40 |
> | OFormer  | 1.83 | 1.83 |
> | GNOT |  0.75 | 0.88 |
> | ONO | 0.61 | 1.18 |
> | Transolver | 0.47 | 0.62 |
> | LNO | 0.51 | 0.52 |
> | CALM-PDE | 0.58 | 0.53 |
>
> The results show that CALM-PDE outperforms 6 out of 8 baselines and ranks among the top models alongside LNO and Transolver, demonstrating strong performance on both time-dependent and time-independent problems. While Geo-FNO underperforms in this setting, it remains among the top models on time-dependent tasks, highlighting that model performance can vary with problem type. We do not claim universal superiority, but CALM-PDE offers a competitive, convolution-based alternative to Fourier and transformer models.
>
> > **(W2)** Some important baselines such as factorized FNO (FFNO) and transolver, which are quite competitive in this domain are missing.
>
> We’ve added F-FNO as an additional baseline and benchmarked it on the 2D Navier-Stokes datasets as well as the irregularly sampled cylinder dataset. Please see the following table. For comparison with Transolver and a further comparison with F-FNO, we added the airfoil and elasticity datasets. Please see the table in our response to (W1).
>
> | Model    | 2D Navier-Stokes $\nu=1e^{-4}$ (Rel. L2 Error) | 2D Navier-Stokes $\nu=1e^{-5}$ (Rel. L2 Error) | 2D Cylinder (Rel. L2 Error) |
> | -------- | --------------------- | --------------------- |  ----------- |
> | F-FNO | 0.0877 (+191%) | 0.08573 (-17%) | 0.1420 (+20%) |
> | CALM-PDE | 0.0301 | 0.1033 | 0.1186 |
>
> > **(W3)** By compressing the solution into a smaller latent space, the method inevitably discards some high-frequency or fine-grained information.
>
> Compressing the solution into a smaller latent space inevitably discards some high-frequency details, a trade-off we accept for improved efficiency. We acknowledge this limitation explicitly in the paper (see line 321 in the Limitations section), and it is a common challenge in reduced-order models.
>
> > **(Q1)** Can the authors show a comparison of performance and efficiency to Factorized FNO? [...] Given its relevance and efficiency, particularly on similar problems, it seems like an important missing baseline.
>
> We added F-FNO as an additional baseline. Please see our response for (W1) and (W2).
>
> > **(Q2)** I also think another recent architecture like Transolver should be discussed and compared to. The best comparisons would be on the irregular grid problems like airfoil, elasticity, plasticity or pipe problems commonly used as baselines for these problems.
>
> We benchmarked our model on the mentioned airfoil and elasticity datasets and included the errors for Transolver. Please see our response for (W1).

---

> > ### Comment · Reviewer_2MgG · 2025-08-02
> >
> > Thank you for the response. I will take it into consideration while deciding my final rating.

---

> > > ### Author Response · Authors · 2025-08-03
> > >
> > > Thank you for reading our rebuttal and your response. We hope we have addressed most of your concerns. If there are any remaining questions or comments, please don’t hesitate to reach out.

---

> > > > ### Comment · Reviewer_2MgG · 2025-08-05
> > > >
> > > > I thank the authors for the new experiments and the detailed rebuttal, which have addressed several of my earlier concerns. However, I still believe the method's limited performance on high-frequency datasets remains a significant drawback and hard to ignore. This is especially relevant given recent findings in the literature that commonly used PDE benchmarks tend to be overly simplistic, exhibiting predominantly low-frequency behavior (see Buitrago et al., ICLR 2025). As such, I remain unconvinced that the proposed method would perform competitively on more challenging benchmarks with richer frequency content, such as the dataset introduced in Buitrago et al..
> > > >
> > > > Secondly, while the paper emphasizes irregular grid and time-dependent problems, it is still unclear how the proposed method compares to state-of-the-art models like FFNO or Transolver in this regime. If I understood correctly, the additional experiments during the rebuttal focused mainly on time-independent + spatially irregular setups, with only a single time-dependent+irregular experiment involving FFNO. While I am not necessarily requesting new experiments since I understand its very difficult to find resources and setup these experiments in the short period, I do believe that broader benchmarking—especially on datasets combining temporal dynamics and spatial irregularity against more recent architectures—is needed to more convincingly demonstrate the method’s advantages in terms of both accuracy and efficiency.
> > > >
> > > > I am therefore inclined to maintain my original rating.

---

> > > > > ### Author Response · Authors · 2025-08-05
> > > > > **Authors' Response Part 1/2**
> > > > >
> > > > > Thank you for your response. We address your comments separately as follows.
> > > > >
> > > > > > **(1)** However, I still believe the method's limited performance on high-frequency datasets [...]
> > > > >
> > > > > We already test our model on two challenging benchmarks listed in the referenced work. In particular, we use the 1D Burgers’ equation from PDEBench [1] and the 2D Navier-Stokes datasets from Li et al. [2]. We use the viscosities $\nu=1e^{-3}$ for the Burgers’ equation as well as $\nu=1e^{-4}$ and $\nu=1e^{-5}$ for Navier-Stokes, similar to the setup in Buitrago et al.. Please see the benchmark results in Table 1, line 223 of our paper. On the 1D Burgers’ equation, CALM-PDE outperforms all baselines. On the $\nu=1e^{-4}$ problem, CALM-PDE also outperforms all models, and on the $\nu=1e^{-5}$ benchmark, it still outperforms two transformer-based baselines. The latter dataset covers a low training data regime and contains only 1000 training samples, while the other dataset contains 9.8k samples for training. Notably, the PDE problems are similar to Buitrago et al., but the concrete setup differs (e.g., $\Delta t$ = 1.0s in our setup vs $\Delta t$ = 0.5s for Navier-Stokes).
> > > > >
> > > > > Nevertheless, as mentioned previously, we do not claim that CALM-PDE is universally better on all PDE problems (e.g., on problems with fine-grained details). This is also not our aim and scientific contribution. Our scientific contribution is an architecture featuring a novel encoder and decoder based on continuous and adaptive convolutions, which enables the model to effectively sample the spatial domain by learning query points. We demonstrate that the novel encoder and decoder can efficiently compress and decompress the PDE solution into/from a small latent space. In contrast, previous models relied on discrete CNNs (e.g., LE-PDE [3]), which limits the model to regular grids, or on Transformers (e.g., LNO and AROMA [4]) for encoding and decoding. Our experiments demonstrate that our new approach, featuring a novel encoder and decoder, is effective on commonly used benchmark problems, as the other reviewers stated correctly.
> > > > >
> > > > > Previous work, such as AROMA [4], which is also a reduced-order model, shows that their model fails on problems with fine-grained details (Appendix C.7 of the AROMA paper). However, solving PDEs with high-frequency content is a fundamental problem of neural PDE solvers and is not limited only to reduced-order models or to AROMA and our approach. Thus, improving models for high-frequency problems is a different research direction and not the scientific contribution or scope of our paper.
> > > > >
> > > > > The referenced framework of Buitrago et al. shows that adding memory to the neural PDE solver reduces the error on high-frequency problems. Other methods, such as PDE-Refiner [5] or NO+DM [6], tackle this issue by proposing frameworks inspired by denoising diffusion. The mentioned methods have in common that they build upon an existing neural PDE solver architecture, which is different from what we aim with our method. CALM-PDE is a new architecture and alternative to architectures such as LNO, OFormer, and AROMA. However, we consider it as interesting future work to explore and improve the performance of CALM-PDE on high-frequency problems by combining it with the previously mentioned methods.
> > > > >
> > > > > > **(2)** [...] If I understood correctly, the additional experiments during the rebuttal focused mainly on time-independent + spatially irregular setups, with only a single time-dependent+irregular experiment involving FFNO.
> > > > >
> > > > > In **(W1)** and **(Q2)**, you suggested a comparison with architectures like Transolver on datasets such as the airfoil and elasticity dataset, which are time-independent problems. Thus, we’ve conducted the experiments on these datasets and included the results in our response for **(W1)**. The results show that CALM-PDE also achieves state-of-the-art performance on time-independent benchmark problems, demonstrating its versatility.
> > > > >
> > > > > Additionally, we added F-FNO as a baseline for the time-dependent problems. We also benchmarked F-FNO on the time-dependent 2D airfoil dataset and included the results. Furthermore, we added AROMA [2] as a baseline, which is a state-of-the-art reduced-order model from NeurIPS 2024. Please see the following table.
> > > > >
> > > > >
> > > > > | Model    | 2D Navier-Stokes 1e-4 | 2D Navier-Stokes 1e-5 | 2D Airfoil | 2D Cylinder |
> > > > > | -------- | --------------------- | --------------------- | -------|  ----------- |
> > > > > | F-FNO | 0.0877 (+191%) | 0.08573 (-17%) | 0.1307 (+154%) | 0.1420 (+20%) |
> > > > > | AROMA | 0.1257 (+318%) | 0.1801 (+74%) | 0.0403 (-22%) | 0.1140 (-4%) |
> > > > > | CALM-PDE | 0.0301 | 0.1033 | 0.0515 | 0.1186 |

---

> > > > > > ### Comment · Reviewer_2MgG · 2025-08-05
> > > > > >
> > > > > > I appreciate the author's response:
> > > > > >
> > > > > > 1) I understand that the paper does not claim to be performing well on high-frequency content datasets. As I can also see from Appendix Figure 18 panel B of the paper where visually I can see smoothening wrt the ground truth. I am only talking about the nu=1e-5 dataset for this example. My concern is more general that given this problem, if the main benchmarks included more complex datasets, such as Rayleigh Benard simulations (or more complex MHD simulations in Physics), which are provided on some recent benchmarks such as the Well dataset, then CALM-PDE in its current form will not do as well. The main reason I brought up Buitrago et al. was to cite an example where it's shown that a lot of widely used benchmarks mostly contain datasets with low frequencies. So to summarize, I understand that CALM-PDE has noted this as a limitation, but to me this limitation is a bit hard to ignore. But I acknowledge that this is not the main focus of this paper, so I will not count this as a negative in making my decision.
> > > > > >
> > > > > > 2) Thank you for adding one more experiment for time-dependent + irregularly sampled dataset. AROMA vs. FFNO comparison looks good to me now. I think adding this to the manuscript will be useful.
> > > > > >
> > > > > > I think with the new table, my main concerns are resolved and I will increase the score by 1 point.

---

> > > > > > > ### Author Response · Authors · 2025-08-05
> > > > > > >
> > > > > > > We appreciate your efforts in reviewing our work, your fast response, and the fruitful discussion. We are glad to see that we have addressed your main concerns.
> > > > > > >
> > > > > > > **(1)** Thank you for the clarification. We acknowledge your concerns regarding high-frequency problems. Given their significance, we explicitly address them in the limitations section of the paper. We understand that adapting CALM-PDE to better handle high-frequency scenarios is an important and valuable direction for future research. Thus, we consider it as future work.
> > > > > > >
> > > > > > > **(2)** Yes, we will include F-FNO and AROMA as additional baselines in the tables of the manuscript. In general, we will add all new experiments that we have conducted during the rebuttal and discussion period to the manuscript.

---

> ### Author Response · Authors · 2025-08-05
> **Authors' Response Part 2/2**
>
> > [...] I do believe that broader benchmarking [...] is needed to more convincingly demonstrate the method’s advantages in terms of both accuracy and efficiency.
>
> We already compare our method against many recent architectures. For instance, we compare our model against recent state-of-the-art methods, including PiT (ICML 2024) [7], LNO (NeurIPS 2024), and AROMA (NeurIPS 2024) [3]. We also include models such as FNO/Geo-FNO (ICLR 2021 and JMLR 2023) and OFormer (TMLR 2023), since they are commonly used and still achieve state-of-the-art performance. With the new experiments, we include a comparison against F-FNO (ICLR 2023) on time-dependent and time-independent problems as well as a comparison with Transolver (ICML 2024) on time-independent problems.
>
> Furthermore, we compare our method on 8 different datasets and problems, including time-dependent and time-independent problems. The used datasets are commonly used in the community to demonstrate the effectiveness of new models (e.g., see AROMA paper).
>
> Thus, we have benchmarked CALM-PDE on a broad set of benchmark problems and compared it against a broad set of state-of-the-art models. The experiments show that the model outperforms or achieves competitive errors on these versatile benchmark problems, which demonstrates the effectiveness of our new approach.
>
> **References** \
> [1] Takamoto, M., Praditia, T., Leiteritz, R., MacKinlay, D., Alesiani, F., Pflüger, D., & Niepert, M. (2022). PDEBENCH: An extensive benchmark for Scientific machine learning. In arXiv [cs.LG]. http://arxiv.org/abs/2210.07182 \
> [2] Li, Z., Kovachki, N., Azizzadenesheli, K., Liu, B., Bhattacharya, K., Stuart, A., & Anandkumar, A. (2020). Fourier neural operator for parametric partial differential equations. In arXiv [cs.LG]. http://arxiv.org/abs/2010.08895 \
> [3] Wu, T., Maruyama, T., & Leskovec, J. (2022). Learning to accelerate Partial Differential Equations via latent global evolution. In arXiv [cs.LG]. http://arxiv.org/abs/2206.07681 \
> [4] Serrano, L., Wang, T. X., Naour, E. L., Vittaut, J.-N., & Gallinari, P. (2024). AROMA: Preserving spatial structure for latent PDE modeling with local neural fields. In arXiv [cs.LG]. http://arxiv.org/abs/2406.02176 \
> [5] Lippe, P., Veeling, B. S., Perdikaris, P., Turner, R. E., & Brandstetter, J. (2023). PDE-Refiner: Achieving accurate long rollouts with neural PDE solvers. In arXiv [cs.LG]. http://arxiv.org/abs/2308.05732 \
> [6] Oommen, V., Bora, A., Zhang, Z., & Karniadakis, G. E. (2024). Integrating neural operators with diffusion models improves spectral representation in turbulence modeling. In arXiv [cs.LG]. http://arxiv.org/abs/2409.08477 \
> [7] Chen, J., & Wu, K. (2024). Positional knowledge is all you need: Position-induced Transformer (PiT) for operator learning. In arXiv [cs.LG]. http://arxiv.org/abs/2405.09285

---

### Note · Authors · 2025-08-12

We sincerely thank the reviewers once again for their valuable time, thoughtful feedback, and fruitful discussions, which helped us to improve our submission. We are glad to hear that we could address the main concerns by providing additional explanations and experiments.

During the rebuttal and discussion phase, we’ve conducted the following experiments and added the following explanations, all of which have been incorporated into the manuscript, thereby enhancing its overall quality.

- We added two more baselines (F-FNO and AROMA) on time-dependent problems.
- We included two datasets on time-independent problems showing that CALM-PDE also works on this kind of problem and can generalize across complex, varying geometries.
- We included a comparison of CALM-PDE’s hyperparameters with discrete CNNs and conducted a hyperparameter study to demonstrate the robustness of the hyperparameters.
- We conducted experiments to investigate the scaling effect of the latent dimension (i.e., number of query points) and the scaling effect of the output query mesh.
- We included an explanation of how to obtain the latent dimension.
- We added an explanation about the computational costs of the CALM layers.
- We included an experiment to show the encoding-decoding capabilities of our method.

Besides the experiments, we’ve improved the contributions listed in the manuscript by emphasizing that the encoder-decoder is the novelty and contribution, and not the paradigm itself. Furthermore, we added clarification that CALM-PDE is not intended to be universally better than the baselines, but an alternative method based on continuous and adaptive convolutions to commonly used FNOs and transformer-based methods that works on both regular and irregular grids. Additionally, we improved the conclusion by emphasizing that our main contribution, a novel encoder and decoder, can efficiently compress and decompress PDE solutions into/from a small latent space. We also added improving CALM-PDE for high-frequency problems to the future work section due to their significance in real-world applications.

---

### Decision · Program_Chairs · 2025-09-17

**Decision:**

Accept (spotlight)

**Comment:**

This paper introduces CALM-PDE, which compresses arbitrarily discretized PDEs with CALM layers (based on continuous convolutions) with multi-level learnable query points, evolves the system in latent space with transformer, and decodes back into original space with CALM based decoder. The method achieves competitive performance compared to strong baselines, and places more query points in regions that are intuitively important.

The reviewers recognizes (which I concur) that the paper addresses the important problem of arbitrarily-discretized domain, which is an important problem in solving PDEs. The reviewers also commend that the paper is well-motivated, well-written and easy to read, the experiments are thorough, and the method achieves competitive performance. During the rebuttal, the authors added additional baselines (e.g., F-FNO, AROMA), added additional experiments showing that the method can generalize across complex, varying geometries, and added additional study into the hyperparameters, etc. These improvements addresses the reviewers' concerns. The authors need to incorporate the additional experiments and discussion into the camera-ready version of the paper. Additionally, for generalization to new geometries, the authors need to detail how the query points' location and parameters are initialized.

Overall, I recommend spotlight.